# Social exposome and brain health outcomes of dementia across Latin America

Joaquin Migeot[1,2,33], Stefanie D. Pina-Escudero[3,4,33], Hernan Hernandez[1,33], Raul Gonzalez-Gomez [1], Agustina Legaz[1,5], Sol Fittipaldi [1,2,3], Elisa de Paula França Resende[3,6], Claudia Duran-Aniotz[1], Jose Alberto Avila-Funes[7], Maria I. Behrens [8,9,10,11], Martin A. Bruno[12], Juan Felipe Cardona[13], Nilton Custodio[14], Adolfo M. García[3,5,15], Maria E. Godoy[1,5], Kun Hu [16], Serggio Lanata[4], Brian Lawlor[2], Francisco Lopera[17], Marcelo Adrian Maito[5], Diana L. Matallana[18,19], Bruce Miller[3,4], J. Jaime Miranda[20], Maira Okada de Oliveira[3,21], Pablo Reyes [22], Hernando Santamaria-Garcia[22,23], Andrea Slachevsky[24,25,26,27], Ana L. Sosa[28], Leonel T. Takada[21], Jacqueline M. Torres[29], Sven Vanneste [2,30,31], Victor Valcour[3,4], Olivia Wen[32], Jennifer S. Yokoyama [3,4,29], Katherine L. Possin[3,4] & Agustin Ibanez[1,2,3,5] ✉

A multidimensional social exposome (MSE)—the combined lifespan measures of education, food insecurity, financial status, access to healthcare, childhood experiences, and more—may shape dementia risk and brain health over the lifespan, particularly in underserved regions like Latin America. However, the MSE effects on brain health and dementia are unknown. We evaluated 2211 individuals (controls, Alzheimer's disease, and frontotemporal lobar degeneration) from a non-representative sample across six Latin American countries. Adverse exposomes associate with poorer cognition in healthy aging. In dementia, more complex exposomes correlate with lower cognitive and functional performance, higher neuropsychiatric symptoms, and brain structural and connectivity alterations in frontal-temporal-limbic and cerebellar regions. Food insecurity, financial resources, subjective socioeconomic status, and access to healthcare emerge as critical predictors. Cumulative exposome measures surpass isolated factors in predicting clinical-cognitive profiles. Multiple sensitivity analyses confirm our results. Findings highlight the need for personalized approaches integrating MSE across the lifespan, emphasizing prevention and interventions targeting social disparities.

Poverty and socioeconomic disparities have multidimensional effects that can harm the brain through multiple pathways. The exposome refers to the totality of environmental, social, and biological exposures accumulated over a lifetime. Within this framework, the social exposome captures the cumulative and multidimensional impact of socially driven exposures that influence health outcomes[1]. These effects may exert a more substantial influence on aging and dementia than isolated risk factors[2–4]. For instance, individuals growing up in poverty may face limited educational opportunities and food insecurity, relying on low-cost, nutrient-poor diets that impair brain development[3]. In adulthood, chronic stress from traumatic events and financial insecurity may add to this burden[5], while in older age, accumulated adverse exposures throughout life contribute to physiological dysregulation, increasing vulnerability to neurodegeneration. The cumulative burden of socially-

related factors[5] (e.g., low educational attainment, adverse childhood experiences, and traumatic events), aggravating factors (e.g., limited access to healthcare[6]), and affected domains (e.g., financial burden and reduced social interaction) may significantly exacerbate dementia phenotypes. These cumulative social burdens constitute the multi-dimensional social exposome (MSE)[1], extending traditional factors such as education level and socioeconomic status (SES). Social determinants of health (SDH), which encompass some proxies of the social exposome, play a crucial role in brain health, healthy aging, and dementia[2,7]. SDH has been linked to the risk and prevalence of dementia, with education and SES being associated with core clinical and neurocognitive measures[8,9]. However, current research fails to characterize the MSE across different life stages[10]. Factors such as food insecurity, access to healthcare[6], childhood experiences, and exposure to traumatic events[11] may significantly contribute to dementia, but they have not been systematically addressed. In particular, the impact of the social exposome on core dimensions of brain health in aging and dementia—cognition, functional ability, neuropsychiatric manifestations, brain atrophy, and connectivity—remains unknown.

Latin America (LA) presents a wide spectrum of social and environmental factors that may influence late-life brain health outcomes. Heterogeneous and distributed factors associated with social and health disparities impact cognition and functional ability across LA[7]. These factors are linked to historical, sociopolitical, and sociocultural diversity[12], as well as political polarization and its potential impact on health. The region has the second-highest estimated global prevalence of dementia[13] and is one of the most unequal regions in the world, with significant disparities in both material and non-material social resources. These include financial assets, healthcare access, education, social support, and infrastructure[12]. Such inequality is reflected by Gini scores frequently surpassing 0.47, indicating considerable wealth inequality[14]. LA is typified by the intersection of unique educational challenges, food insecurity, childhood difficulties, healthcare inequalities, and adverse environments[5,12]. We have recently shown that in LA, inequalities are associated with worsening cognitive and functional phenotypes[7], brain reserve[15], and accelerated brain age[2] in aging and dementia. Other reports suggest stronger effects of SDH than genetic ancestry for dementia risk in LA[16]. Thus, the social exposome in the region seems unique[17]. However, current research is limited to specific factors such as SES, education, or single macrosocial estimates[2,7,14–16] usually investigated in isolation. Considering multidimensional lifespan measures, a cumulative social exposome assessment may be much more informative in capturing the heterogeneity and diversity of underserved populations facing multifocal adversities.

Significant gaps impede an adequate understanding of the MSE. Most evidence is based only on broad measures of education (years of schooling) and SES[10], usually factored out as control variables rather than targeted as relevant phenomena[8]. Although valid, SES measures present multiple limitations[10], such as inconsistent definitions (e.g., varying metrics for education, income, and profession), lack of standardization across contexts, and fragmentation of domains precluding the assessment of cumulative burdens. SDH research has neglected multimodal clinical-cognitive measures and their biological embedding of brain health[10]. The available evidence is biased toward the United States (US) and European data, with an underrepresentation of individuals suffering from larger disparities in countries with more diverse origins. Models based on data from high-income countries fail to capture brain and dementia phenotypes accurately and do not generalize well to more diverse populations, particularly in LA[18]. Though variously defined, brain health is consistently considered a multilevel construct encompassing brain structural and functional integrity, cognitive health, functional ability, and low levels of neuropsychiatric symptoms. There is a need to explore the various aspects of the social exposome and its impact on brain health outcomes in aging and dementia for underrepresented populations. This may help to develop tailored biopsychosocial models of dementia.

We aimed to address these gaps by employing an MSE approach (Fig. 1) to characterize brain health in aging and dementia in terms of cognition, functional ability, neuropsychiatric symptoms, brain volume, and functional connectivity in a non-representative sample of 2211 individuals across 6 LA countries, including healthy controls (HC), patients with Alzheimer's disease (AD) and frontotemporal lobar degeneration (FTLD) (Table 1). These two conditions are among the leading causes of dementia and are critically underrepresented in research from LA. We created a multidimensional score leveraging expert criteria, factorial confirmatory analysis, and structural equation modeling of combined variables in different domains (education, food insecurity, financial status, assets, access to healthcare, childhood labor, subjective SES, childhood experiences, traumatic events, and relationship assessments). Each dimension included a combination of various components across different life stages (from birth to present). For instance, food insecurity assessed whether the participant had to eat less or less healthily due to economic hardship from 0 to 10, 35 to 45 years old, and recently. Access to healthcare is considered an item evaluating challenges in paying for services from 0 to 10, 35 to 45 years old, and recently, including specific barriers such as economic reasons, medical appointment availability, transportation, distance to medical facilities, lack of accompaniment, the COVID-19 pandemic, or socio-political events. Childhood experiences covered the number of children in the household, whether the participant left school to support the family, the frequency of feeling loved and receiving physical affection during childhood, and family conflicts. The MSE also included exposure to alcohol and/or drug abuse in the household, experiences of neglect during childhood, and household organization (see Dimensions in MSE assessment). We anticipated that adverse MSE would have an extensive effect on brain health, including (a) poor cognition, reduced functional ability, and increased neuropsychiatric symptoms; (b) a more substantial impact than individual classical predictors (e.g., education, income); and (c) larger brain atrophy and reduced functional connectivity in dementia-sensitive brain regions for AD and FTLD.

## Results

The MSE was operationalized by grouping items measuring facets of similar dimensions based on expert consensus agreement. Then, we calculated the weighted averages on these items to create dimension scores for education, food insecurity, financial status, assets, healthcare access, childhood labor, subjective SES, childhood experiences, traumatic events, and relationships. These scores were obtained by scaling between the minimum and maximum values for 319 items and then averaging these to obtain a value for each dimension (Fig. 1B). We validated these dimensions via confirmatory analysis. Structural equation modeling was fed with these composite factors to create a global latent MSE score per subject (Fig. 1C). To examine whether the combined MSE effects were stronger than, and not driven by, single factors influencing the global score, we tested the contributions of individual predictors with Lasso models. Then, we compared them with the global MSE score via meta-regressions. The structural and functional brain correlates of the MSE scores were tested in a subsample with whole-brain gray matter volume and ROI-to-ROI functional connectivity (Fig. 1D). Multiple sensitivity analyses were employed to confirm that the results were not biased by individual dimensions or traditional SES factors, demographic, recruitment bias, disease-related factors (age, sex, disease severity, age at diagnosis, years after diagnosis, and FTLD subtype); or scanner effects (scanner type, signal-to-noise ratio, and motion parameters).

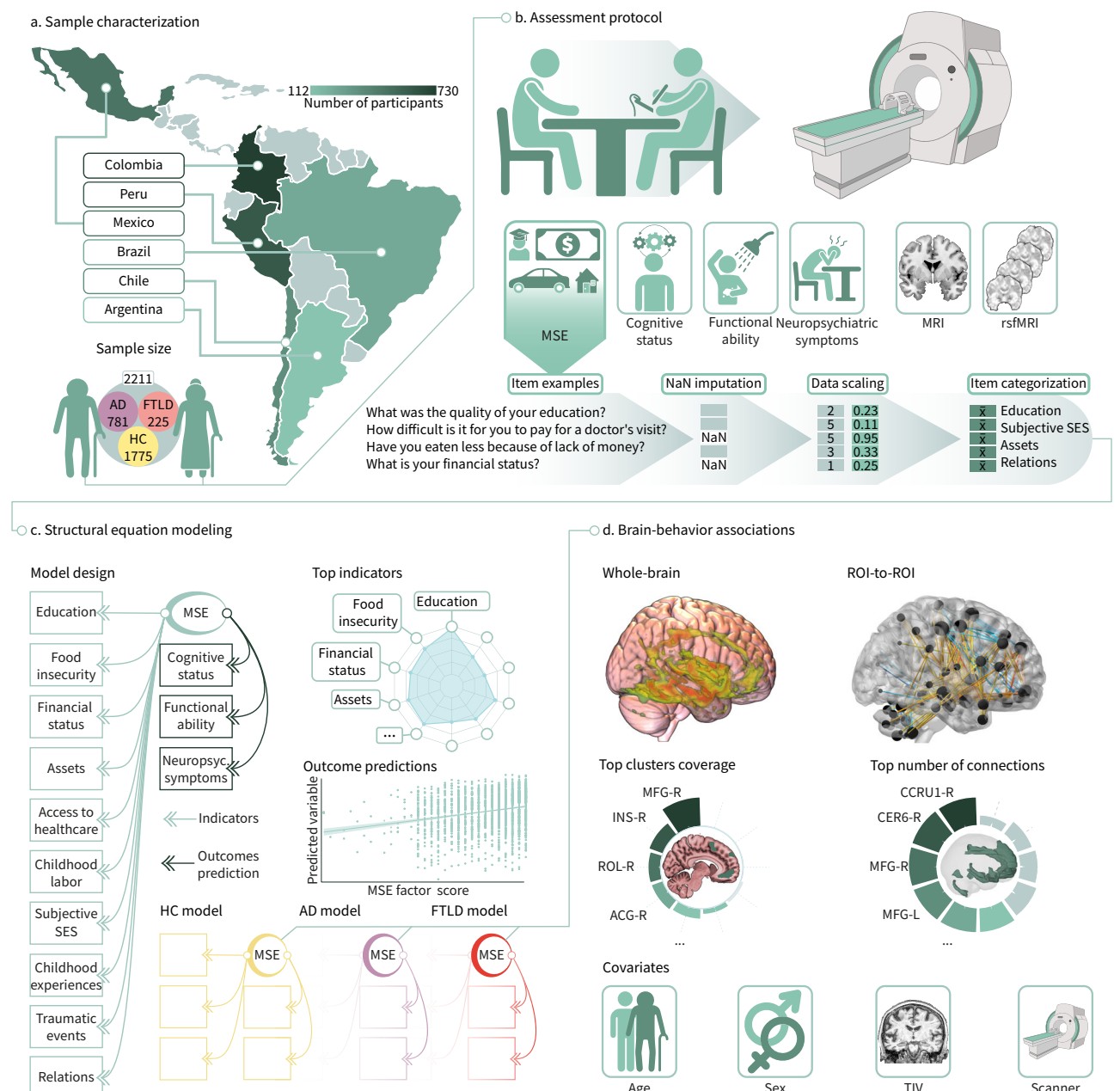

**Fig. 1 | Study design and analysis pipeline. a** Healthy controls (HC) and participants with Alzheimer's disease (AD) and frontotemporal lobar degeneration (FTLD) were recruited from six Latin American countries: Argentina, Brazil, Chile, Colombia, Mexico, and Peru. **b** The assessment protocol included a comprehensive multidimensional social exposome (MSE) evaluation alongside clinical, cognitive, and neuroimaging measures. For MSE quantification, all items underwent imputation for missing values (NaN) and were then min-max scaled between 0 and 1, with higher values indicating lower levels of social adversity. Composite scores for each participant were calculated by averaging variables within specific categories: education, food insecurity, financial status, assets, access to healthcare, childhood labor, subjective socioeconomic status (SES), childhood experiences, traumatic events, and relationships. **c** These domain-specific scores, ranging from 0 to 1, served as inputs in structural equation models (SEM) to predict cognitive function, functional ability, and neuropsychiatric symptoms in the whole sample (green), HC

(yellow), AD (purple), and FTLD (red). **d** The global MSE score, extracted from the latent variable, was employed to predict brain structure and functional connectivity. The coverage of regions within the top clusters identified in whole-brain voxel-based morphometry analyses using magnetic resonance imaging (MRI) was examined. Specific regions may include the middle frontal gyrus (MFG), insula (INS), rolandic operculum (ROL), and anterior cingulate cortex (ACG). In resting-state functional magnetic resonance imaging (rsfMRI) analyses, region of interest (ROI)-to-ROI connectivity was assessed by identifying the top regions with the highest number of connections. For illustration purposes, critical regions such as the cerebellum (CCRU cerebellum crus, CER cerebellum lobule), and middle frontal gyrus (MFG) are highlighted. All analyses were controlled for age, sex, total intracranial volume (TIV, for voxel-based morphometry analyses), and scanner. Created in BioRender. Migeot, J. (2025) https://BioRender.com/rxcmtgq (sample size, assessment protocol, covariate icons and MRI device).

## MSE associations with cognitive and clinical domains

For the entire sample (Fig. 2A), the larger the adverse MSE, the worse the negative outcomes (cognition, functional ability, and neuropsychiatric symptoms): CFI = 0.939, TLI = 0.893, RMSEA = 0.076, and SRMR = 0.049. The global score of adverse MSE was associated with

diminished cognitive function ($\beta = 0.261$ [95% CI = 0.18–0.282], $p < 0.001$), decreased functional ability ($\beta = 0.079$ [95% CI = 0.015–0.11], $p < 0.001$), and heightened neuropsychiatric symptoms ($\beta = 0.105$ [95% CI = 0.033–0.132], $p < 0.001$). Top MSE indicators included education ($\beta = 0.732$ [95% CI = 0.685–0.778], $p < 0.001$),

**Table 1 | Demographic, cognitive, functional, and neuropsychiatric characterization of subjects**

| | | HC | AD | FTLD | Statistics |
|---|---|---|---|---|---|
| Sample size | | 1175 | 781 | 255 | |
| Country | Argentina | 52 | 55 | 5 | |
| | Brazil | 116 | 35 | 21 | |
| | Chile | 78 | 83 | 39 | |
| | Colombia | 250 | 386 | 94 | |
| | Mexico | 227 | 103 | 26 | |
| | Peru | 452 | 119 | 70 | |
| Age | | 60.21 (11.62) | 70.5 (8.4) | 67.02 (7.78) | $F(2, 2208) = 247.24$, $p < 1 \times 10^{-97}$, $\eta^2 = 0.183$, HC < FTLD < AD |
| Sex (F:M) | | 858:317 | 494:287 | 130:125 | $\chi^2 (2, 2211) = 53.85$, $p < 1 \times 10^{-12}$ |
| Cognition (MMSE) | | 27.50 (2.90) | 20.93 (4.41) | 21.25 (6.05) | $F(2, 2208) = 747.01$, $p < 1 \times 10^{-248}$, $\eta^2 = 0.404$, AD < FTLD < HC |
| Functional ability (Pfeffer) | | 0.20 (1.11) | 12.16 (8.11) | 13.69 (9.14) | $F(2, 2208) = 1258.36$, $p < 1 \times 10^{-300}$, $\eta^2 = 0.533$, HC < AD < FTLD |
| Neuropsychiatric symptoms (NPI-Q) | | 1.45 (2.90) | 6.57 (5.71) | 10.48 (7.02) | $F(2, 2208) = 537.53$, $p < 1 \times 10^{-191}$, $\eta^2 = 0.327$, HC < AD < FTLD |

followed by food insecurity ($\beta = 0.702$ [95% CI = 0.658–0.745], $p < 0.001$), financial status ($\beta = 0.694$ [95% CI = 0.659–0.730], $p < 0.001$), assets ($\beta = 0.649$ [95% CI = 0.610–0.687], $p < 0.001$), and access to healthcare ($\beta = 0.506$ [95% CI = 0.456–0.557]) (Supplementary Table 1).

In healthy aging, adverse MSE was linked to lower cognitive function ($\beta = 0.610$ [95% CI = 0.463–0.737], $p < 0.001$) but did not predict functional ability ($\beta = -0.047$ [95% CI = −0.026 to 0.073], $p = 0.101$) nor neuropsychiatric symptoms ($\beta = -0.032$ [95% CI = −0.114 to −0.002], $p = 0.259$) (Fig. 2B). The model showed lower fit indices with CFI = 0.859, TLI = 0.756, RMSEA = 0.105, and SRMR = 0.092. Key MSE indicators encompassed assets ($\beta = 0.924$ [95% CI = 0.880–0.967], $p < 0.001$), education ($\beta = 0.921$ [95% CI = 0.865–0.976], $p < 0.001$), financial status ($\beta = 0.526$ [95% CI = 0.477–0.574], $p < 0.001$), childhood labor ($\beta = 0.440$ [95% CI = 0.388–0.491], $p < 0.001$), and food insecurity ($\beta = 0.373$ [95% CI = 0.302–0.444], $p < 0.001$) (Fig. 2B, Supplementary Table 1).

In individuals with AD, adverse MSE was associated with lower cognitive function ($\beta = 0.229$ [95% CI = 0.157–0.309], $p < 0.001$), reduced functional ability ($\beta = 0.079$ [95% CI = 0.004–0.152], $p = 0.038$), and an increase in neuropsychiatric symptoms ($\beta = 0.104$ [95% CI = 0.013–0.193], $p = 0.007$) (Fig. 2C). The model obtained excellent fit indices with CFI = 0.955, TLI = 0.922, RMSEA = 0.058, and SRMR = 0.041. The leading MSE indicators were education ($\beta = 0.800$ [95% CI = 0.730–0.869], $p < 0.001$), food insecurity ($\beta = 0.716$ [95% CI = 0.650–0.782], $p < 0.001$), assets ($\beta = 0.683$ [95% CI = 0.621–0.745], $p < 0.001$), financial status ($\beta = 0.643$ [95% CI = 0.583–0.703], $p < 0.001$), and subjective SES ($\beta = 0.534$ [95% CI = 0.476–0.593], $p < 0.001$) (Fig. 2C, Supplementary Table 1).

For those with FTLD, adverse MSE predicted decreased cognitive function ($\beta = 0.165$ [95% CI = 0.037–0.289], $p = 0.017$), reduced functional ability ($\beta = 0.140$ [95% CI = 0.015–0.288], $p = 0.040$), and increased neuropsychiatric symptoms ($\beta = 0.141$ [95% CI = 0.019–0.281], $p = 0.040$) (Fig. 2D). The model showed excellent fit indices with CFI = 0.985, TLI = 0.973, RMSEA = 0.036, and SRMR = 0.041. The most critical MSE indicators comprised education ($\beta = 0.796$ [95% CI = 0.698–0.895], $p < 0.001$), financial status ($\beta = 0.691$ [95% CI = 0.598–0.784], $p < 0.001$), subjective SES ($\beta = 0.675$ [95% CI = 0.589–0.762], $p < 0.001$), assets ($\beta = 0.666$ [95% CI = 0.569–0.763], $p < 0.001$), and access to healthcare ($\beta = 0.626$ [95% CI = 0.511–0.742], $p < 0.001$) (Fig. 2D, Supplementary Table 1).

In summary, adverse MSE was related to impairments in cognitive, functional, and neuropsychiatric domains. The most significant adverse MSE predictors in healthy aging were assets, education, and financial status, associated with lower cognitive function. In dementia, the importance of MSE predictors was more distributed across the different domains.

## Single dimensions compared to the global MSE performance

When individual predictors were assessed for the entire sample using Lasso regression, a reduced effect across outcomes was observed compared to the global MSE score (all $p < 0.01$, Table 2). For cognitive function ($R^2 = 0.11$, $f^2 = 0.13$, MnSqErr = 0.02, MAE = 0.12), the top predictors were education (coef = 0.23), assets (coef = 0.09), and childhood experiences (coef = 0.02). Similarly, a reduced effect was noted for functional ability ($R^2 = 0.04$, $f^2 = 0.04$, MnSqErr = 0.05, MAE = 0.18), with subjective SES (coef = 0.16), education (coef = 0.12), and access to healthcare (coef = 0.11) emerging as key predictors. For neuropsychiatric symptoms ($R^2 = 0.03$, $f^2 = 0.03$, MnSqErr = 0.03, MAE = 0.13), access to healthcare (coef = 0.14), childhood experiences (coef = 0.09), and subjective SES (coef = 0.05) were identified as leading predictors (Supplementary Table 2).

This pattern of reduced effects of individual predictors compared to the global MSE score was consistent between groups (all $p < 0.01$, Table 2). For healthy aging, reduced effects for cognition were observed ($R^2 = 0.43$, $f^2 = 0.75$, MnSqErr = 0.01, MAE = 0.05), with education (coef = 0.14) and assets (coef = 0.26) being prominent contributors. Among individuals with AD, a reduced effect ($R^2 = 0.06$, $f^2 = 0.07$, MnSqErr = 0.02, MAE = 0.10) highlighted education (coef = 0.18), food insecurity (coef = 0.03), and access to healthcare (coef = 0.02) as key predictors. For those with FTLD, a reduced effect ($R^2 = 0.04$, $f^2 = 0.04$, MnSqErr = 0.03, MAE = 0.14) was linked to childhood experiences (coef = 0.08), childhood labor (coef = 0.03), and education (coef = 0.04) (Supplementary Table 2).

Regarding functional ability in AD ($R^2 = 0.01$, $f^2 = 0.01$, MnSqErr = 0.05, MAE = 0.18), subjective SES (coef = 0.08) was the primary predictor. For FTLD ($R^2 = 0.05$, $f^2 = 0.05$, MnSqErr = 0.06, MAE = 0.20), financial status (coef = 0.17), relations (coef = 0.15), and childhood labor (coef = 0.08) were the top contributors. In the context of neuropsychiatric symptoms, AD showed a reduced effect ($R^2 = 0.02$, $f^2 = 0.02$, MnSqErr = 0.03, MAE = 0.13), with food insecurity (coef = 0.05), childhood experiences (coef = 0.05), and access to healthcare (coef = 0.01) as leading predictors. For FTLD ($R^2 = 0.05$, $f^2 = 0.05$, MnSqErr = 0.04, MAE = 0.16), traumatic events (coef = 0.29), financial status (coef = 0.13), and childhood labor (coef = 0.07) were identified as top predictors (Supplementary Table 2). Thus, the cumulative MSE effects were neither driven by nor weaker than any independent effects.

## Adverse MSE and brain volume in dementia

In individuals with AD, more adverse MSE was linked to reduced gray matter volume in frontal and cerebellar regions, with the highest cluster peak values observed in the superior, middle, and inferior frontal gyrus, vermis (lobules IV–VI), cerebellum (crus I and II, lobule VIIb and VIII), and insula (Fig. 3A, Table 3). A closer inspection of the

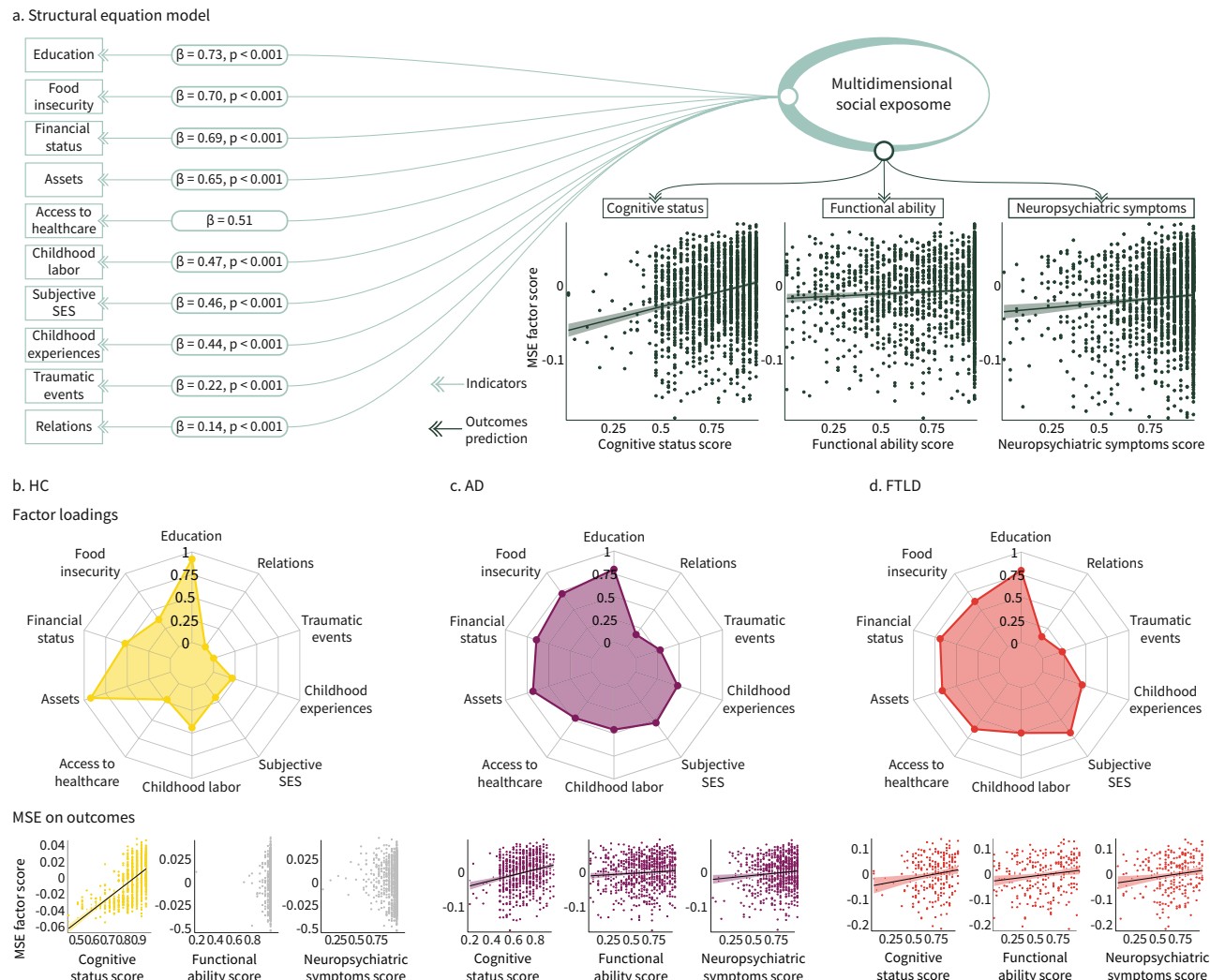

**Fig. 2 | Structural equation modeling (SEM) of multidimensional social exposome and clinical-cognitive phenotypes.** SEM between the multidimensional social exposome (MSE) and cognitive status, functional ability and neuropsychiatric symptoms. Top factor loadings (exact *p*-values: education = $1 \times 10^{-308}$; food insecurity = $1 \times 10^{-308}$; financial status = $1 \times 10^{-308}$; assets = $1 \times 10^{-308}$; childhood labor = $1 \times 10^{-308}$; subjective SES = $1 \times 10^{-308}$; childhood experiences = $1 \times 10^{-308}$; traumatic events = $1 \times 10^{-15}$; relations = $1 \times 10^{-8}$) and distribution of association with outcomes for **a** the whole sample (green), **b** healthy controls (HC, yellow), **c** persons with Alzheimer's disease (AD, purple) and **d** frontotemporal lobar degeneration

(FTLD, red). No adjustments were made for multiple comparisons. The SEM was implemented as a two-sided model by default. Functional ability and neuropsychiatric symptoms score were inverted to facilitate the interpretability of associations relative to the cognitive status score, so that the higher the value, the higher the functional capacity and the lower the neuropsychiatric symptoms. In the scatter plots, each point represents an individual observation. The black line indicates the linear regression fit, and the shaded area represents the 95% confidence interval of the fit. Source data are provided as a Source Data file.

regions covered in the top clusters revealed the Rolandic operculum, gyrus rectus, orbitofrontal cortex, anterior cingulate gyrus, and the putamen, which showed a stable association between less adverse MSE and higher gray matter volumes of each top cluster (Fig. 3A).

For those with FTLD, larger adverse MSE was associated with decreased gray matter volume in fronto-temporo-cerebellar regions. The highest cluster peaks were found in the vermis (lobule IV, V, and VII), superior temporal gyrus, superior and middle frontal gyrus, and anterior cingulate cortex (Fig. 3B, Table 4). Coverage of top cluster regions identified the cerebellum (crus I and II, lobule IV, V, VI, VIII, and IX), Rolandic operculum, insula, Heschl's gyrus, and postcentral gyrus. A consistent link was found between fewer adverse MSE and increased gray matter volumes in each top cluster (Fig. 3B).

**Adverse MSE and altered brain connectivity in dementia**

In individuals with AD, more adverse MSE was associated with lower frontotemporal connectivity between the anterior cingulate cortex,

hippocampus, parahippocampal gyrus, striatum, insula, and cerebellum (lobule III and vermis lobule III). Conversely, greater MSE adversity was associated with higher connectivity between frontal (superior, middle, inferior), temporal (middle), occipital (middle, cuneus), and cingulate regions (Fig. 4A, Supplementary Table 3). The middle temporal, cingulate, and superior frontal gyrus had the most connections (Fig. 4A).

In FTLD, more adverse MSE was associated with lower fronto-temporal connectivity primarily between frontal (superior, middle, and inferior, precentral gyrus), temporal (superior, parahippocampal gyrus), limbic (amygdala, hippocampus, and cingulate cortex), and cerebellar regions (crus I, lobule III, IV-V, IX, and X). Greater MSE adversity was associated with higher connectivity in frontal (superior, middle, inferior), temporal (superior, middle, inferior, and parahippocampal gyrus), hippocampus, and cerebellar regions (crus I, lobule IV-V, VI, and X) was observed (Fig. 4B, Supplementary Table 4). The regions with the highest number of connections were the

**Table 2 | Meta-regression comparison between the global MSE score and the effect of individual MSE dimensions**

| Variable | Model type | Common effect size | Models difference | |
|---|---|---|---|---|
| | | | $\chi^2$ | p-Value |
| **All subjects** | | | | |
| Cognition | Global MSE score | 0.23 (0.23–0.24) | 3498.68 | $<1 \times 10^{-768}$ |
| | Individual MSE dimensions | 0.12 (0.12–0.12) | | |
| Functional ability | Global MSE score | 0.06 (0.06–0.06) | 130.98 | $<1 \times 10^{-30}$ |
| | Individual MSE dimensions | 0.05 (0.05–0.05) | | |
| Neuropsychiatric symptoms | Global MSE score | 0.08 (0.08–0.08) | 803.71 | $<1 \times 10^{-176}$ |
| | Individual MSE dimensions | 0.03 (0.03–0.03) | | |
| *HC* | | | | |
| Cognition | Global MSE score | 0.60 (0.60–0.60) | 7133.11 | $<1 \times 10^{-308}$ |
| | Individual MSE dimensions | 0.43 (0.43–0.44) | | |
| *AD* | | | | |
| Cognition | Global MSE score | 0.23 (0.23–0.24) | 3078.06 | $<1 \times 10^{-675}$ |
| | Individual MSE dimensions | 0.08 (0.08–0.08) | | |
| Functional ability | Global MSE score | 0.08 (0.08–0.09) | 369.65 | $<1 \times 10^{-82}$ |
| | Individual MSE dimensions | 0.04 (0.03–0.04) | | |
| Neuropsychiatric symptoms | Global MSE score | 0.11 (0.10–0.11) | 502.61 | $<1 \times 10^{-111}$ |
| | Individual MSE dimensions | 0.05 (0.04–0.05) | | |
| *FTLD* | | | | |
| Cognition | Global MSE score | 0.17 (0.16–0.18) | 375.32 | $<1 \times 10^{-83}$ |
| | Individual MSE dimensions | 0.08 (0.07–0.08) | | |
| Functional ability | Global MSE score | 0.15 (0.14–0.16) | 174.28 | $<1 \times 10^{-39}$ |
| | Individual MSE dimensions | 0.08 (0.08–0.09) | | |
| Neuropsychiatric symptoms | Global MSE score | 0.15 (0.15–0.16) | 144.27 | $<1 \times 10^{-33}$ |
| | Individual MSE dimensions | 0.10 (0.09–0.10) | | |

Meta-regression models were implemented as two-sided. No adjustments were made for multiple comparisons.

hippocampus, inferior and superior frontal gyrus, and cingulate gyrus (Fig. 4B).

**Sensitivity analyses**

Multiple effects were tested to confirm the robustness of our analysis. The association between adverse MSE with reduced cognition and functional ability and increased neuropsychiatric symptoms was preserved (all $p < 0.01$) after controlling for the effect of demographics, potential recruitment bias, and dementia-related covariates (age, sex, disease severity, age at diagnosis, years after diagnosis, and FTLD subtype; Supplementary Table 5). Despite an unbalanced distribution of age and sex across groups (Table 1), all effects remained consistent after controlling for these covariates. Moreover, none of these covariates were associated with the MSE score (all $p > 0.10$), confirming that potential confounders did not drive the main effects (Supplementary Table 5). The associations between adverse MSE with reduced cognition and functional ability and increased neuropsychiatric symptoms were preserved (all $p < 0.01$) after controlling for country level (Supplementary Information 1). Additional models were run to rule out potential bias from single factors linked to worse clinical-cognitive phenotypes (i.e., education and SES). Excluding education from the global MSE score confirmed that the association between adverse MSE and reduced cognition remained significant (Supplementary Information 2). Furthermore, models considering only objective or subjective SES showed lower fit indices and weaker associations than MSE in predicting cognition, functional ability, and neuropsychiatric symptoms (Supplementary Information 3). In summary, the robustness of our findings was supported through multiple approaches, including the validation of MSE dimensions, controlling for key covariates, and assessing cross-country consistency. MSE models demonstrated better fit and predictive power than SES-only

models, and the effects persisted even when education was excluded from the MSE or missing values were not imputed.

To control the effects of different scanners, scanner type was included as a dummy covariate of no interest among the predictors in whole-brain gray matter volume and ROI-to-ROI functional connectivity analyses[14], thus corroborating that scanner type did not explain findings (Tables 3 and 4, Supplementary Tables 3 and 4). The observed effects in whole-brain gray matter volume and ROI-to-ROI functional connectivity analyses were robust to imbalances in age and sex distribution across groups. Consistent results were observed while controlling for these covariates in the regression models (Tables 3 and 4, Supplementary Tables 3 and 4). We also explored the association between MSE and f/MRI data quality metrics (Methods, Sensitivity analysis). Across all subjects, MSE was not associated with either spatial (MRI: $r = 0.002$, $p = 0.958$; fMRI: $r = -0.063$, $p = 0.183$) or temporal ($r = 0.005$, $p = 0.912$) signal-to-noise ratio nor motion parameters ($r = -0.076$, $p = 0.088$). The same null effects were observed across groups (all $p > 0.113$) (Supplementary Table 6). Further, we tested separate models excluding participants with less than 70% of artifact-free frames[19]. Results replicated the main functional connectivity effects (Supplementary Tables 7 and 8). In addition to the primary analyses, we ran additional voxel-based morphometry models employing a harmonization strategy that normalized the voxel intensity values for MRI scans[2]. The results of this harmonization closely matched those from the main results, as the associations with the core regions were preserved (Supplementary Tables 9 and 10). To provide a framework for replication with previous evidence on functional connectivity in LA populations[2,14,15,20–24], we ran additional models employing the automated anatomical labeling (AAL) atlas. Results matched the connectivity patterns described in the main results (Supplementary Information 4, Supplementary Tables 11 and 12). All

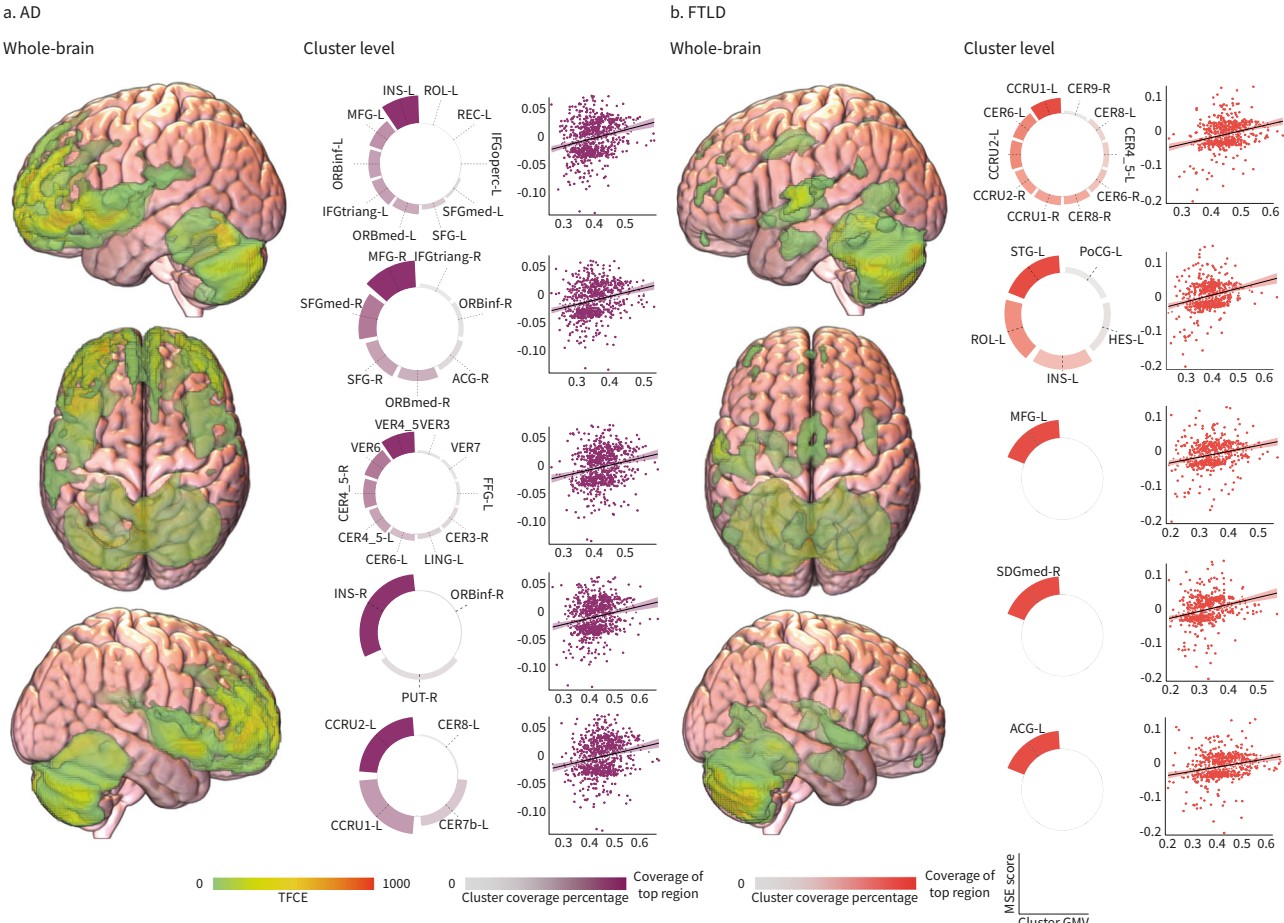

**Fig. 3 | Association between multidimensional social exposome and brain structure. a** Whole-brain analysis for persons with Alzheimer's disease (AD, purple) shows regions with significant associations between the multidimensional social exposome (MSE) and gray matter volume (GMV), highlighted by threshold-free cluster enhancement (TFCE) values. Circular barplot charts are included for the top cluster, indicating the percentage coverage of the top region within the cluster. Scatterplots illustrate the relationship between MSE scores and clusters' GMV. **b** Whole-brain analysis for persons with frontotemporal lobar degeneration (FTLD, red) highlights regions significantly associated with MSE, with TFCE values similarly visualized. Corresponding scatterplots depict the associations between MSE scores and GMV for each identified cluster. In the scatter plots, each point represents an individual observation. The black line indicates the linear regression fit, and the shaded area represents the 95% confidence interval of the fit. ACG anterior cingulate gyrus, CCRU cerebellum crus, CER cerebellum lobule, FFG fusiform gyrus, IFGoperc inferior frontal gyrus, opercular part, IFGtriang inferior frontal gyrus, triangular part, INS insula, LING lingual gyrus, MFG middle frontal gyrus, ORBinf inferior frontal gyrus, orbital part, ORBmed medial orbital gyrus, PUT putamen, REC rectus gyrus, ROL rolandic operculum, SFG superior frontal gyrus, SFGdor superior frontal gyrus, dorsolateral, SFGmed medial superior frontal gyrus, VER vermis lobule. Source data are provided as a Source Data file.

analyses were controlled by age, sex, and total intracranial volume (for voxel-based morphometry analyses). Multiple comparisons were false discovery rate (FDR)-corrected.

## Discussion

This study aimed to identify the association of MSE and brain health outcomes in aging and dementia in LA. We found that the more adverse the MSE, the larger the cognitive, functional, and neuropsychiatric impairment. The combined effects of MSE components demonstrated stronger associations, even after excluding education. The stronger model fit and greater magnitude of associations for MSE compared to SES alone underscored a cumulative multidimensional burden. The top predictors in healthy aging were composite measures of education and assets, whereas, in dementia, more multidimensional exposomes emerged, such as food insecurity, financial status, subjective SES, and access to healthcare. Adverse MSE were linked to the structure and functional connectivity of dementia-specific regions in AD and FTLD, predominantly fronto-temporo-limbic and cerebellar regions. Potential compensatory functional hyperconnectivity, characterized by higher functional connectivity associated with higher MSE adversity, was observed primarily in regions with reduced volume that were associated with more adverse MSE. Variations in participants' country of origin, demographic and dementia-related factors, image acquisition methods, or signal quality did not account for the results. Cross-country and cross-site effects did not influence the link between MSE and clinical, cognitive, and neuropsychiatric outcomes. Results demonstrate that the MSE, as a combined measure, exhibits stronger effects than individual factors and composite SES measures, providing a low-dimensional representation of the cumulative social exposome. The findings highlight how diverse data and tailored modeling can capture precise brain health outcomes of aging and dementia, crucial for efficient prevention and multicomponent interventions addressing both individual- and societal-level determinants.

While some studies have assessed the influence of SES[7], nutrition, relationships[7], access to healthcare, and traumatic experiences on functional ability and neuropsychiatric symptoms in aging, we evaluated their complex cumulative effects across the lifespan, ranging from birth to the present time. We expanded the classical assessments of SDH[25] by including food insecurity, access to healthcare, subjective SES, and a weighted and lifespan evaluation of these factors. For instance, most previous studies[8,9] have used years of education, and we expanded these effects with a combined measure that incorporates

**Table 3 | MSE on voxel-based morphometry for AD**

| Region | Coordinates | | | $K_E$ | TFCE | Peak $P_{FDR}$ |
|---|---|---|---|---|---|---|
| | X | Y | Z | | | |
| Left superior frontal gyrus | −25.5 | 60 | 13.5 | 4241 | 893.734 | 0.011 |
| Left middle frontal gyrus | −34.5 | 55.5 | 7.5 | | 812.536 | 0.011 |
| Left superior frontal gyrus | −22.5 | 55.5 | 24 | | 807.055 | 0.011 |
| Right inferior frontal gyrus (orbital part) | 45 | 46.5 | −4.5 | 2746 | 697.559 | 0.012 |
| Right middle frontal gyrus | 33 | 55.5 | 7.5 | | 628.300 | 0.012 |
| Right superior frontal gyrus | 25.5 | 61.5 | 7.5 | | 619.292 | 0.014 |
| Vermis (lobules IV and V) | −3 | −60 | −12 | 2198 | 672.708 | 0.014 |
| Vermis (lobule VI) | 1.5 | −69 | −21 | | 618.504 | 0.018 |
| Left cerebellum (lobules IV and V) | −6 | −51 | −10.5 | | 567.011 | 0.018 |
| Left cerebellum (crus II) | −13.5 | −81 | −37.5 | 3004 | 671.336 | 0.018 |
| Left cerebellum (crus I) | −30 | −85.5 | −30 | | 632.271 | 0.017 |
| Left cerebellum (crus I) | −34.5 | −76.5 | −33 | | 623.019 | 0.019 |
| Right insula | 34.5 | 13.5 | −4.5 | 820 | 589.028 | 0.020 |
| Right cerebellum (lobule VIII) | 27 | −73.5 | −55.5 | 937 | 339.418 | 0.030 |
| Right cerebellum (crus II) | 19.5 | −78 | −39 | | 304.794 | 0.045 |
| Right cerebellum (lobule VIIb) | 31.5 | −70.5 | −46.5 | | 293.710 | 0.038 |
| Right cerebellum (crus II) | 45 | −63 | −39 | 399 | 297.271 | 0.040 |
| Right cerebellum (crus I) | 40.5 | −70.5 | −36 | | 280.771 | 0.045 |
| Right cerebellum (crus I) | 34.5 | −76.5 | −33 | | 275.386 | 0.046 |
| Left superior frontal gyrus | −27 | 61.5 | 6 | 32 | 258.641 | 0.011 |
| Left superior frontal gyrus | −21 | 46.5 | 39 | 41 | 109.647 | 0.011 |
| Left superior frontal gyrus (medial part) | −9 | 54 | 33 | 98 | 108.794 | 0.047 |
| Left superior frontal gyrus (medial part) | −9 | 42 | 51 | | 104.690 | 0.034 |
| Left superior frontal gyrus (medial part) | −7.5 | 51 | 45 | | 102.827 | 0.041 |
| Left inferior frontal gyrus (triangular part) | −51 | 21 | 27 | 21 | 59.985 | 0.011 |
| Left inferior frontal gyrus (triangular part) | −49.5 | 30 | 21 | | 29.292 | 0.018 |
| Left middle frontal gyrus | −34.5 | 31.5 | 43.5 | 29 | 36.023 | 0.025 |
| Left precentral gyrus | −52.5 | 0 | 42 | 11 | 16.939 | 0.029 |

$P_{FDR} < 0.05$, TFCE-corrected, covariates: age, sex, TIV, and recording site. Regions are presented on the MNI space using the AAL atlas.

**Table 4 | MSE on voxel-based morphometry for FTLD**

| Region | Coordinates | | | $K_E$ | TFCE | Peak $P_{FDR}$ |
|---|---|---|---|---|---|---|
| | X | Y | Z | | | |
| Vermis (lobule VII) | −4.5 | −67.5 | −24 | 19680 | 1024.724 | 0.011 |
| Vermis (lobule VII) | 4.5 | −69 | −24 | | 1018.217 | 0.011 |
| Vermis (lobules IV and V) | −3 | −60 | −12 | | 979.880 | 0.011 |
| Left superior temporal gyrus | −63 | −7.5 | 7.5 | 655 | 321.480 | 0.011 |
| Left superior temporal gyrus | −54 | −15 | 6 | | 250.572 | 0.011 |
| Left superior temporal gyrus | −57 | −3 | −1.5 | | 209.457 | 0.011 |
| Left middle frontal gyrus | −34.5 | 55.5 | 7.5 | 87 | 77.526 | 0.011 |
| Left middle frontal gyrus | −43.5 | 48 | 16.5 | | 42.374 | 0.013 |
| Left anterior cingulate cortex | −6 | 43.5 | 12 | 30 | 45.076 | 0.012 |
| Right superior frontal gyrus (medial part) | 7.5 | 54 | 39 | 15 | 30.507 | 0.015 |
| Left middle frontal gyrus | −37.5 | 42 | 30 | 23 | 17.195 | 0.033 |

$P_{FDR} < 0.05$, TFCE-corrected, covariates: age, sex, TIV, and recording site. Regions are presented on the MNI space using the AAL atlas.

multiple domains (school type and location, perceived education quality throughout life, the number of books at home during childhood, parental education levels, and years of formal schooling). Financial status captured the difficulty in covering basic needs, the sufficiency of income at the end of the month, the diversity of income sources, the proportion of income spent on housing, and the participant's financial resilience in the event of lost income from 0 to 10, 35 to 45 years old, and recently (see Dimensions in MSE assessment). This approach allowed us to address complex cumulative effects. In healthy aging, cognition was mainly modulated by traditional factors such as education, assets, and financial status[7,26] as assessed with complex measures. Our approach considered the combined influence of traditional (e.g., educational achievement and financial status) and non-traditional factors for each dimension, such as parental education,

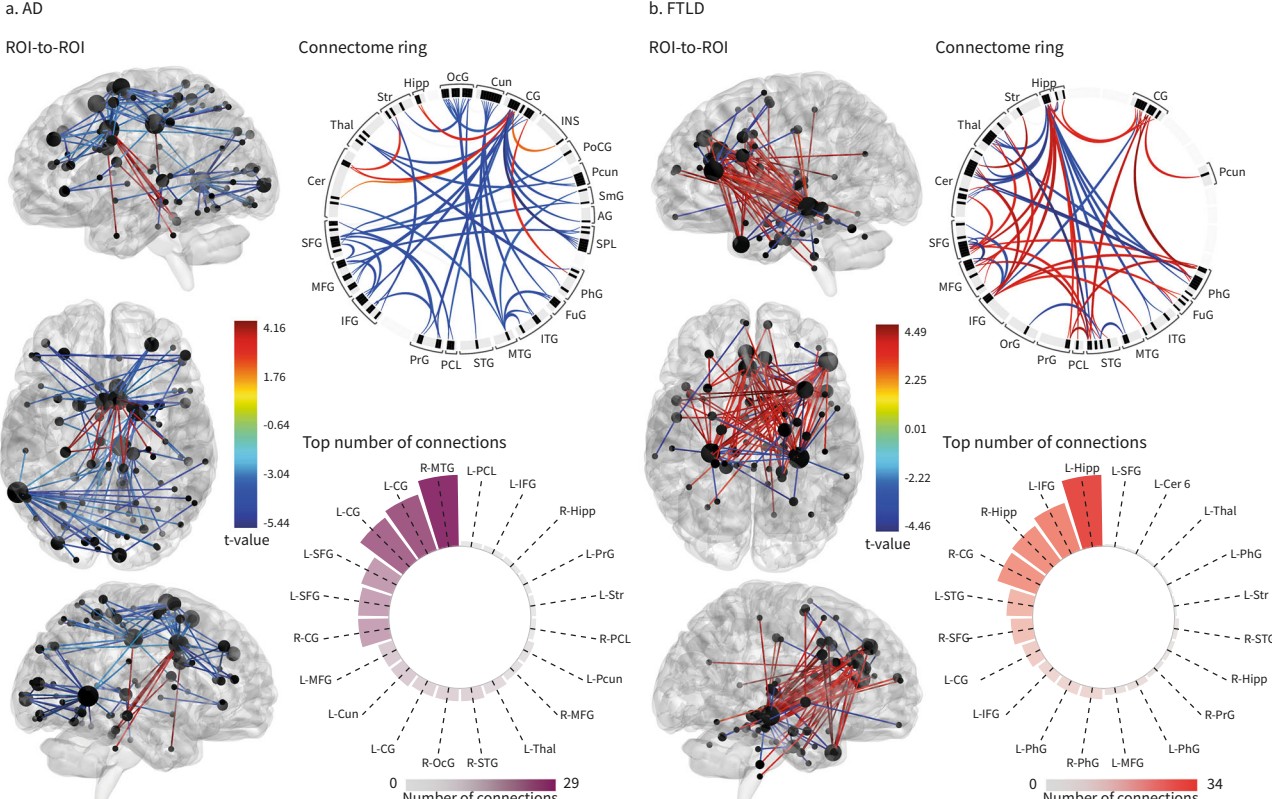

**Fig. 4 | Associations between multidimensional social exposome and brain connectivity. a** Functional connectivity in persons with Alzheimer's disease (AD) displayed through region of interest (ROI)-to-ROI connectivity maps, where color-coded edges represent significant *t*-values for decreased (red) and increased (blue) connectivity associated with MSE. The connectome ring provides a simplified visualization of interconnections between key regions. The circular barplot chart highlights the top regions with the highest number of connections in this group (purple). **b** In persons with frontotemporal lobar degeneration (FTLD), ROI-to-ROI connectivity maps similarly provide visualizations of significant connections associated with MSE, with the connectome ring displaying interconnections between key regions and a circular barplot chart showing key regions with the highest number of connections in this group (red). AG angular gyrus, Cer cerebellum, CG cingulate gyrus, Cun Cuneus, FuG fusiform gyrus, Hipp hippocampus, IFG inferior frontal gyrus, INS insula, ITG inferior temporal gyrus, MFG middle frontal gyrus, MTG middle temporal gyrus, OcG occipital gyrus, PCL paracentral lobule, Pcun precuneus, PhG parahippocampal gyrus, PoCG postcentral gyrus, PrG precentral gyrus, SFG superior frontal gyrus, SmG supramarginal gyrus, SPL superior parietal lobule, STG superior temporal gyrus, Str striatum, Thal thalamus. Source data are provided as a Source Data file.

rurality, education quality, financial stress, food insecurity, difficulties in accessing healthcare[6], childhood experiences, exposure to traumatic events[11], and relationships[7]. Each factor has been studied individually, but not combined into a composite exposome measure to assess its association with brain health. Thus, our approach allowed us to examine the cumulative effects of multiple dimensions via the MSE. The impact on functional ability and neuropsychiatric symptoms in aging was null, probably due to the floor effect of these measures (see below). While factors like food insecurity, access to healthcare, childhood experiences, and subjective SES were significant predictors, the stronger contribution of classical factors probably reflects regional inequalities. Formal education is the main modifiable risk factor for dementia in LA[27], associated with marked inequalities between the US and low-middle income countries, and critically differentiates aging and dementia brain outcomes in LA vs. the US[15]. SES is the second most burdensome disparity-related factor across LA regarding cognition[7], probably by reducing cognitive reserve in older adults[26]. Current results support the role of education in preventing cognitive decline and dementia, this time involving intergenerational and quality-based measures. Moreover, results call for brain capital policy initiatives[1]; without systematically decreasing the structural inequalities[14], the adverse individual-level effects on brain and cognitive health in the region cannot be properly addressed.

Different exposomes impact clinical, cognitive, and brain outcomes in healthy aging[28] and dementia[4]. Physical exposome

factors—such as lead exposure, microplastics, heatwaves, and air pollution—affect cognitive performance and brain structure. Lifestyle exposomes, including physical activity, smoking, alcohol consumption, and cognitive engagement in late life, are associated with dementia risk[3]. Specific components related to social exposome—including education, social isolation, socioeconomic status (SES), and structural inequality—have been linked to cognition, functional ability[7], and alterations in brain structure and functional connectivity[2,14,15]. The complex social exposome in AD[29] and FTLD[4] suggests a lifespan cumulative influence[3,30–32]. These effects can lead to accumulated dysregulation across different life stages, predisposing individuals to neurodegeneration in aging. Components of adverse social exposome in AD are associated with dysregulated stress responses, linked with reduced cognition, impaired functional ability, increased neuropsychiatric symptoms, and brain burden[33]. Elevated plasma cortisol levels, indicative of HPA axis dysregulation, have been associated with reduced total brain volume, decreased glucose metabolism in the frontal cortex, and increased beta-amyloid load in AD-vulnerable regions[34,35]. Emerging evidence in FTLD links altered stress responses with neurodegenerative processes via multisystemic dysregulations[31,32]. Overall, our MSE approach offers a combined and granular characterization of the social exposome in dementia. This work supports current efforts to define the broader dementia exposome[4], enabling more precise risk profiling.

The brain correlates of adverse MSE in dementia align with disease-specific environmental explanations for pathophysiological mechanisms in AD. These suggest lower resistance to p-tau and Aβ burden across frontotemporal regions associated with lifetime stressful events[33], stress-mediated impaired insulin signaling and cerebellar vulnerability[30], and inflammatory models of clinical progression[33]. Regarding brain connectivity, our results showed similar associations with those reported with physical exposome[2], education[15], and structural inequalities[14] in dementia. There is little evidence on environmental stress and FTLD phenotypes, suggesting a substantial genetic contribution to this condition[17]. However, FTLD brain health outcomes were linked with the social exposome in this study. Evidence of pathophysiological mechanisms suggests that the brain burden observed in FTLD may be driven by the accumulation of physiological stressors over time. Cerebellar gray matter patterns with posterior involvement are associated with educational disparities[15] and structural inequality[14]. Evidence on pathophysiological mechanisms links the brain burden to the accumulation of physiological stressors (i.e., allostatic overload) in FTLD[31,32] and also relates it to SDH[2,14,15], targeting both disease-specific and non-specific regions. Environmental stressors can impact distributed brain regions while accelerating specific pathophysiological processes related to dementia[2,14,15,21]. In AD, progressive amyloid and tau pathology affect frontal and parietal structures, potentially amplifying the effects of MSE disrupting executive and integrative processes related to cognitive, functional, and neuropsychiatric impairments[33–35]. In FTLD, early degeneration targets fronto-temporo-limbic networks, leading to connectivity disruptions underlying socioemotional dysregulation, behavioral changes, and similar clinical outcomes. MSE may primarily exacerbate AD structural vulnerability through cumulative stress and resource depletion, whereas in FTLD, they predominantly could aggravate functional connectivity dysregulation. These neurobiological pathways may suggest that MSE can be linked with partially comparable clinical outcomes despite differing structural and functional correlates. Conversely, greater functional connectivity is associated with greater MSE adversity is aligned with evidence on compensatory effects in AD[36] and FTLD[37], as well as aversity-related factors such as socioeconomic status[38], financial stress[39], and discrimination[40]. This may reflect a compensatory process triggered by brain burdens associated with (a) neurodegenerative processes[36,37], (b) disease-specific environmental factors contributing to pathophysiological mechanisms[30–32], and (c) their interactions. In line with the prolonged accumulation of amyloid and tau pathology and with environmental exposures linked to the later onset of AD[33–35], stronger compensatory functional connectivity associated with greater MSE adversity was observed in this group compared to FTLD. This may reflect the longer-lasting impact of degenerative and environmental burdens on brain structural signatures (atrophy) in AD, whereas in FTLD, more transient compensatory changes may emerge earlier in the disease course due to its typically younger onset. In addition to the disease-sensitive associations in each condition, we found additional cerebellar involvement in structural correlations. Although sometimes neglected in the literature[41], the cerebellum is a core structure in atrophy related to dementia[41] and environmental stressors such as low-SES[9], structural inequality[14], and poverty[42]. This may also explain why the cerebellum is part of the allostatic interoceptive network, which is impaired in individuals with dementia[21,31,43,44] and those with social disparities[32,38]. Thus, allostatic interoception provides a framework for understanding the dual impact of dementia pathophysiology and social adversity. Overall, our findings deepen the association between social exposome and brain health outcomes in dementia in LA.

Our study has several strengths. We provided comprehensive coverage of social exposome dimensions and extensive brain health phenotyping of aging and dementia beyond current approaches that usually only consider traditional and unidimensional measures (e.g., years of education, basic SES scores), specific life periods (i.e., aging), or unimodal phenotypes. The cumulative lifespan of social exposome provides an understanding of the long-term impacts on brain health and dementia. We leveraged a large and geographically diverse sample across LA encompassing six countries with varying characteristics, surpassing the size of those typically found in current research in a region with marked inequalities[14,16]. Our protocol was harmonized across settings, and multiple sensitivity analyses confirmed robust results against heterogeneities in demographics (age and sex), country of origin, disease-related factors and recruitment bias (disease severity, age at diagnosis, years after diagnosis, FTLD subtype), and scanner effects (scanner type, signal-to-noise ratio, and motion parameters). Recruitment bias—factors that may influence the distribution of MSE scores during participant recruitment—could influence the results. However, several reasons suggest this is not the case, as we performed multiple control analyses. These included: (a) accounting for variations in disease severity, age at diagnosis, and time elapsed since diagnosis to mitigate potential overrepresentation of participants from disadvantaged groups in more advanced disease stages; (b) eliminating circularity in the assessment of MSE by excluding variables that may confound socioeconomic disparities with dementia outcomes (e.g., access to specific services such as cell phones could indicate either socioeconomic disparities or the consequences of cognitive decline); and (c) conducting within-group analyses to identify MSE associations through a dimensional approach rather than presuming uniform patterns across groups. However, future studies should incorporate other probabilistic or stratified random sampling strategies in the designs. The findings demonstrate the need for diverse data and tailored modeling to capture brain health outcomes of aging and dementia, crucial for more efficient prevention and interventions.

These strengths are accompanied by limitations that invite further research. While our MSE assessment considers different life stages, it relies on participant self-report, which could introduce bias. This is a limitation of the broader field of dementia assessment through self-report methods. We partially addressed this issue by omitting items from the evaluation that can be biased by dementia symptomatology (e.g., smartphone use, occupational situation, Supplementary Information 5) and corroborating participant self-reports with information provided by caregivers. This is consistent with standard practices for questionnaire-based assessments in individuals with dementia, such as evaluating depressive symptoms[45], quality of life[46], and activities of daily living[47]. Additionally, we controlled the impact of disease severity on these predictors to minimize confounding effects. Longitudinal designs and objective retrospective records may confirm and expand the current results. While our MSE assessment considers different life stages, items were limited to early, middle, and late life (0–10, 35–45 years old, and recently), leaving out other age periods. This approach was taken to ensure a more detailed assessment than traditional SES approaches while avoiding excessive items that could increase participant burden and compromise data quality. However, future studies should incorporate additional life stages to enable a more comprehensive analysis across the lifespan. Factors other than MSE, beyond the social domain, have cumulative effects on brain health[48]. Potentially modifiable risk factors for dementia may be influenced by disparities[3]. LA has one of the highest population-attributable fractions for dementia risk factors in the world[27], associated with high levels of adversity, inequality[5,12], and dementia prevalence[12,13]. Complex interactions between social exposome, genetics, physical exposome (pollution, pesticides, heavy metals)[2,17], and modifiable risk factors are beyond the scope of this work and require additional research. Future studies should assess gene-environmental interactions. Like previous studies employing social determinants measures[15,16], our data were measured at the individual level across countries. Our results do not comprise aggregate-level data. Future studies should incorporate nested designs with aggregated-level analyses and geocoded data relevant to brain health in LA, such as structural inequality[14], air pollution[2], and access to green spaces[49]. A further step in future

research may involve adjusting individual metrics by nested analysis of spatial buffers at different scales (i.e., block [0–100 m], neighborhood [100–500 m], district/commune [500–5000 m], city [5–50 km]). Despite being unavailable in the current neuroscience literature, this would improve the ecological significance of the approach. Our study did not account for (ethnic or racial) discrimination and ancestry/ genetic effects in a region with marked ethnic diversity[16,17].

As other works in the region[2,14,15], a limitation of this study is the reliance on clinical criteria for diagnosis without biomarkers like amyloid-β and tau, assessed via PET or plasma. PET is costly, globally inaccessible, and lacks validation in diverse populations such as LA[50]. Similarly, the feasibility and accuracy of blood-based biomarkers in LA have not yet been developed to validate their use combined with clinical criteria. Sample sizes are relatively modest within each country. ReDLat[51] participants are derived from clinical samples and are not representative of the general population. However, the ReDLat cohort uses a well-distributed LA clinical sample that includes diverse populations[51,52]. While we acknowledge that LA is not a homogenous construct, this does not preclude the use of harmonized datasets across diverse countries[2,7,14,15,20,53], as has been done in other large-scale initiatives such as ADNI and European cohorts, which also feature significant socio-cultural differences. Future work should expand these findings by incorporating community-representative samples to enhance the generalizability of results. Although our recruitment was aligned with current guidelines[54], many of the recommended strategies are designed with US research infrastructure in mind. They may not entirely apply to the general LA context, where limited resources and larger socio-economic disparities often hinder research. Further, our protocol adheres to specific methodological considerations and recommended best practices for cross-national comparisons[55]: (a) The MSE questionnaire was developed and harmonized by a team of experts from various countries where it was subsequently applied, validated with a diverse sample of Latino Spanish-speaking participants[56], and tested to represent the same construct across countries via psychometric validation. (b) The covariables included in structural equation model (SEM) analyses are available and collected under a harmonized protocol across all countries. (c) We used a pooled analysis approach, incorporating country-level effects in sensitivity analysis. Our study addresses under-represented populations from LA, which present unique patterns of brain health[2,7] compared to more homogenous populations (e.g., UK Biobank). The assessments of the dimensions related to social exposome are designed with different objectives and utilize distinct methodologies[10]. Future studies should develop more global and inclusive participation and evaluation globally. The functional ability and neuropsychiatric measures were not sensitive to healthy aging, likely due to a disease-sensitive floor effect of these measures (i.e., healthy controls present floor effects with no impairments). Future studies should consider other assessments tailored to healthy aging. Lastly, our neuroimaging analyses encompassed different scanners and recording parameters. However, sensitivity analyses confirmed that inter-scanner differences and data quality metrics did not compromise the results.

In conclusion, this study evidenced an association of the multidimensional social exposome on brain health outcomes in aging and dementia across LA underserved populations. These results call for the development of tailored models incorporating the impact of social environments in characterizing dementia. Prevention initiatives and multicomponent interventions targeting structural inequality should consider lifespan cumulative effects to foster healthier aging trajectories and improve brain health in LA.

## Methods
### Participants
The study included 2211 participants with a mean age of 64.63 years (SD = 11.26), of whom 67.03% were women. Sex information was determined by self-report. The sample was comprised of individuals with probable AD ($n = 781$), FTLD ($n = 255$), and HC ($n = 1175$) (Table 1). Participants were recruited from the Multi-Partner Consortium to Expand Dementia Research in Latin America (ReDLat)[51,57], with recruitment conducted across six LA countries: Argentina ($n = 112$, HC:AD/FTLD = 52:60), Brazil ($n = 172$, HC:AD/FTLD = 116:56), Chile ($n = 200$, HC:AD/FTLD = 78:122), Colombia ($n = 730$, HC:AD/FTLD = 250:480), Mexico ($n = 356$, HC:AD/FTLD = 227:129), Peru ($n = 641$, HC:AD/FTLD = 452:189). ReDLat collects data from multiple centers across these countries, using a standardized data framework and harmonized diagnostic protocols[7,20,53,58,59]. Participants were recruited from extensive networks including (a) clinical networks, involving memory clinics, neurology departments, and affiliated hospitals; (b) academic collaborations, leveraging partnerships with universities and research institutions; (c) community outreach programs, engaging with local communities through informational sessions, and culturally tailored materials to encourage participation from rural and urban populations with diverse socioeconomic backgrounds; and (d) public health initiatives and local organizations, integrating recruitment efforts with public health campaigns and community groups to raise awareness and facilitate participation. These efforts allowed us to include individuals from rural and urban settings, focusing on under-represented groups, as our ReDLat cohort is marked by socio-economic inequality[14] and educational disparities[15]. Strategies to improve access and recruitment for these groups involve field screenings, community engagement efforts, and the use of mobile units. No compensation was provided to participants. Exclusion criteria included participants with conditions other than AD or FTLD or those with impairments preventing task completion. Diagnoses were determined by consensus among expert healthcare providers at each site, based on cognitive and neurological exams, clinical interviews, and MRI[57]. The diagnoses are based on the clinical criteria established by the National Institute of Neurological and Communicative Disorders and Stroke and the Alzheimer's Disease and Related Disorders Association (NINCDS-ADRDA) for AD, as well as on the clinical criteria specified for FTLD. Healthy Controls had preserved cognition and no history of neurological or psychiatric conditions. A standardized battery was used to capture clinician evaluations. Clinical and cognitive assessments across ReDLat sites were harmonized, normalized, and validated[57]. ReDLat aligns with current guidelines to increase the representativeness of samples enrolled in Alzheimer's disease research centers[54] by implementing inclusive recruitment strategies and strengthened community engagement, geographic decentralization, and unified inclusion and exclusion criteria. All clinicians underwent training and certification by a specialized team and adhered to a quality control protocol[57]. The study received approval from the ReDLat consortium through multiple institutional review boards: FWA00028264, FWA00001035, FWA00028864, FWA00001113, FWA00010121, FWA00014416, FWA00008475, FWA00029236, FWA00029089, and FWA00000068. Data collection and analysis presented no risks related to stigmatization, incrimination, discrimination, animal welfare, environmental or health concerns, safety, security or personal privacy. Additionally, no transfer of biological materials, cultural artifacts or traditional knowledge occurred. All participants signed informed consent in accordance with the 2013 Declaration of Helsinki.

### MSE assessment
**Questionnaire.** An expert team designed a questionnaire that reviewed existing tools and their validity data on various social, economic, familial, and environmental factors that can influence health across the lifespan[56]. This questionnaire was created by different experts working with Latino participants from Mexico, Central America, and South America. It was co-designed by the ReDLat consortium and the UCSF-MAC, following the NIA Health Disparities Research Framework and the Institute of Medicine reports. Matching guidelines

to increase sample representativeness in Alzheimer's disease research[54] and recommended practices for cross-national comparisons[55], validity testing was performed with a diverse sample of Latino Spanish-speaking participants, clinicians, and researchers to iteratively improve the questionnaire and standardize the collection of sociodemographic factors. Unlike typical questionnaires that focus on single domains, such as education, childhood experiences, or traumatic experiences, this questionnaire aimed to evaluate multiple domains simultaneously to understand how life experiences and conditions from childhood to the present impact health. The questionnaire builds on existing tools like the Protocol for Responding to and Assessing Patient Assets, Risk, and Experiences (PRAPARE) and the Latin American and Caribbean Food Security Scale (ELCSA). Questionnaire construction aligns with current harmonization guidelines[60], as (a) items were selected after careful inspection from clinicians and researchers from different cultures (at least one from each country) and comparability across countries; (b) development of psychometric procedures to test if dimension composite score represents a coherent and consistent latent construct of interrelated variables and captures the intended dimension; and (c) transparency and openness by providing the full version of the questionnaire in a repository (https://osf.io/78ng6/).

**Validation.** The protocol has been standardized, validated, and subjected to psychometric standardization samples from different countries, including healthy individuals and those with neurocognitive disorders, ensuring its applicability to diverse populations[56,61]. The questionnaire was administered to 137 participant-caregiver dyads in a more detailed validation. After multivariate assumptions such as normality, linearity, homogeneity, and homoscedasticity, adequacy tests were conducted to ensure data quality (Bartlett's test of sphericity: $\chi^2 = 2191.611$, $p < 0.001$; Kaiser−Meyer−Olkin measure: MSA = 0.75). Lastly, a three-factor SEM was created, including SES, challenging life experiences, and educational environment, which showed good fit indices (TLI = 0.764, RMSEA = 0.062, SRMR = 0.06, CFI = 0.884)[56]. We reviewed the questionnaire to remove items that might create circular associations with dementia-related outcomes. We excluded dimensions that could be directly influenced by dementia symptoms, thereby risking the measurement of those symptoms instead of the intended MSE construct. Specifically, we removed items related to technology access (smartphone use), employment status (work engagement), and income sources (current salary) (Supplementary Information 5).

**Dimensions.** The MSE questionnaire systematically evaluates a comprehensive range of topics to ensure a thorough assessment of the participant's background and experiences.

Education is evaluated through the number of years of formal education, the type of school attended (public or private), the geographical location of the school (rural or urban), and the overall quality of education received (from very bad to excellent), number of books at home during the first 10 years of life (from 0 to more than 200), and the educational achievement of the mother and father (from none to doctoral degree).

Food insecurity captures if the participant had to eat less (yes/no) or less healthy (yes/no) due to economic hardship across the lifespan (from 0 to 10, 35 to 45 years old, and recently), capturing the quality and consistency of food insecurity experienced by the participant across the lifespan.

Financial status assesses the participant's financial stability and ability to meet basic needs over different periods of their life (from 0 to 10, 35 to 45 years old, and recently). It measures the difficulty in covering basic needs, the sufficiency of income at the end of the month, the diversity of income sources, and the proportion of income spent on housing. Additionally, it gauges the participant's financial resilience

in the event of lost income and health insurance (public, private, or none), offering a comprehensive view of their economic situation and security throughout their lifespan.

Assets evaluate the participant's access to essential household utilities and goods over different periods (from 0 to 10, 35 to 45 years old, and recently). It assesses whether the participant had access to electricity, radio, television, refrigerator, washing machine, landline telephone, water heater, indoor bathroom, running water, automobile, computer, internet, sound system, smartphone, and private room. Additionally, it measures the quantity of certain items like televisions, bathrooms, cars, computers, and private rooms the participant had during these periods. This assessment provides a detailed view of the participant's material living conditions and how they have evolved throughout their lifespan.

Access to healthcare assesses the participant's difficulty in affording medical care across different life stages (from 0 to 10, 35 to 45 years old, and recently). It evaluates whether the participant experienced difficulty paying for medical services and explores specific barriers that may have prevented them from seeing a doctor (from not hard to very hard). These barriers include financial constraints, appointment availability, transportation issues, distance, lack of accompaniment, the COVID-19 pandemic, or socio-political events (yes/no). This assessment provides insights into the participant's access to healthcare and the factors influencing it throughout their lifespan.

Childhood labor investigates whether the participant worked before the age of 18 (yes/no) and the specific age range during which they started working (from 5 to 18 years old). It also delves into the reasons for working at a young age, including whether to meet personal needs, support family needs, save money, or learn a new skill (yes/no). This assessment provides insight into the participants' early exposure to work and the SES factors that may have influenced their decision to work during childhood or adolescence.

Subjective SES is evaluated by asking participants to reflect on their family's ability to meet basic needs and their financial situation from 0 to 10, 35 to 45 years old, and recently (from level 0 to 10).

Childhood experiences comprise the participant's early childhood environment, focusing on family structure, emotional support, and potential adverse experiences. It includes questions about the number of children in the household during the first 10 years, whether the participant had to leave school to support the household (yes/no), and the frequency of feeling loved by parents and receiving physical affection. Additionally, it explores conflicts within the family, such as fights between parents, parents-participant, and siblings. The questions also inquire about exposure to alcohol abuse, the organization of the household, and experiences of neglect during childhood (from never to frequently). This assessment provides a comprehensive view of the participant's early family life and its potential impact on their development.

Traumatic events evaluate the participant's exposure to traumatic events, violence, and maltreatment from 0 to 10, 11 to 24, 25 to 34, 35 to 45, 46 to 65 years old, less than a year, and recently. It includes questions about the loss of siblings or children before the age of 18, experiences of political violence and repression, and exposure to theft or robbery both inside and outside the home. It also examines the participants' involvement in or witness to accidents, situations where they feared for their lives, and encounters with dead or injured individuals. Furthermore, the questions explore experiences of physical violence, being attacked with a weapon, humiliation, restrictions on personal freedom, neglect, and financial control. Additionally, the set assesses instances where the participant was pressured to leave their property or coerced into sexual activity (yes/no). These experiences are considered in relation to different individuals (family members, friends, caregivers, strangers) (yes/no). This comprehensive assessment aims to capture the depth and breadth of the participant's

exposure to adverse and potentially traumatic experiences throughout their life.

Relationships assess the participant's frequency of contact with loved ones and experiences of being treated with less respect across different stages of life and in recent times. It measures how often the participant interacts with close family or friends, ranging from less than once a week to more than five times a week. Additionally, it explores the frequency of disrespect or mistreatment experienced during various life stages (0 to 10, 11 to 24, 25 to 34, 35 to 45, 46 to 65 years old, less than a year, and recently). These experiences are categorized by how often they occur, ranging from almost daily to less than once a year or never. This assessment provides insights into the participant's social connections and the prevalence of negative interpersonal experiences throughout their life.

**Scoring.** The questionnaire employs various response formats, including Likert scales, multiple-choice questions (Yes/No), and free text fields. Privacy is prioritized, with the most sensitive questions, such as those about experiences of violence, abuse, and discrimination, answered individually by the participant (without the presence of the caregiver). Following previous reports[62], we scored the questionnaire by creating composite variables by grouping items based on expert consensus on the similarity of the assessed domains. After excluding items (Supplementary Information 5), each variable was min–max scaled between 0 and 1, with 0 and 1 indicating lower and higher levels of adversity of social exposome. Then, the dimension scores for each participant were obtained by averaging the value of the total number of variables per category (education, food insecurity, financial status, assets, access to healthcare, childhood labor, subjective SES, childhood experiences, traumatic events, and relations) to produce a score between 0 and 1. These values for each domain were used as inputs in the SEM. Confirmatory factor analyses showed a good model fit for the dimensions (Supplementary Table 13). These results indicate that the dimension composite scores accurately represent the latent construct and consistently capture the intended dimensions.

### Cognition, functional ability, and neuropsychiatric symptoms

The mini-mental state examination (MMSE) is a widely used cognitive screening tool that evaluates various cognitive functions, including arithmetic, memory, and orientation. It assesses five key domains: orientation, registration, attention and calculation, recall, and language. The MMSE is scored from 0 to 30, with higher scores indicating better cognitive function. A score of 24 or above is considered normal, while scores below 24 suggest cognitive impairment. Specifically, scores between 19 and 23 indicate mild cognitive impairment, scores between 10 and 18 indicate moderate cognitive impairment, and scores below 10 suggest severe cognitive impairment. The MMSE has demonstrated moderate to high reliability and is frequently used across LA populations.

The Pfeffer Functional Activities Questionnaire (PFAQ) is a tool designed to assess the functional ability of older adults, focusing on instrumental activities of daily living. The PFAQ evaluates the capacity to perform activities such as using the telephone, managing finances, preparing meals, and managing medications. Scoring of the PFAQ ranges from 0 to 30, with higher scores indicating greater levels of dependence or functional impairment. Each item is rated on a scale from 0 (independent) to 3 (dependent), and the total score provides an overview of the individual's functional status. The PFAQ is reliable and employed in studies involving LA populations.

The Neuropsychiatric Inventory Questionnaire (NPI-Q) is a streamlined version of the original Neuropsychiatric Inventory (NPI), designed to assess a wide range of neuropsychiatric symptoms commonly found in patients with dementia. The NPI-Q evaluates 12 key domains: delusions, hallucinations, agitation/aggression, depression/dysphoria, anxiety, euphoria/elation, apathy/indifference,

disinhibition, irritability/lability, aberrant motor behavior, nighttime behavioral disturbances, and appetite/eating abnormalities. Scoring of the NPI-Q ranges from 0 to 36, with two components for each domain: the frequency and severity of symptoms. Frequency is rated from 1 (occasionally) to 4 (very frequently), and severity is rated from 1 (mild) to 3 (severe). The total score sums these ratings across all domains, providing a comprehensive measure of the patient's neuropsychiatric profile. Higher scores indicate more severe and frequent symptoms. The NPI-Q has adequate test-retest reliability and convergent validity and is commonly applied in research involving LA populations.

Scores of functional ability and neuropsychiatric symptoms were inverted to facilitate the interpretability of the associations in the same direction as cognitive scores. The higher the value, the higher the functional capacity and the lower the neuropsychiatric symptoms.

### Neuroimaging acquisition and preprocessing

**MRI preprocessing.** 3D T1-weighted images were collected for 875 individuals with a mean age of 65.13 years (SD = 10.77), of whom 65.94% were women. Sex information was determined by self-report. 310 individuals with AD, 123 with FTLD, and 442 HC were included (Argentina [$n = 84$, HC:AD/FTLD = 38:46], Brazil [$n = 92$, HC:AD/FTLD = 69:23], Chile [$n = 137$, HC:AD/FTLD = 53:84], Colombia [$n = 203$, HC:AD/FTLD = 25:178], Mexico [$n = 53$, HC:AD/FTLD = 29:24], Peru [$n = 306$, HC:AD/FTLD = 228:78], Supplementary Table 14). Preprocessing and analysis were employed using VBM with the Computational Anatomy (CAT12) toolbox and in Statistical Parametric Mapping software (SPM 12; Wellcome Center for Human Neuroimaging; www.fil.ion.ucl.ac.uk/spm/software/spm12/) in Matlab R2021a. The standard pipeline included bias-field correction, noise reduction, skull stripping, segmentation, and normalization to the Montreal Neurological Institute (MNI) space at a 1.5 mm isotropic resolution. CAT12 also performed intra-subject harmonization by normalizing data to the mean global intensity for each subject, followed by smoothing gray matter images with a $6 \times 6 \times 6$ mm Gaussian kernel. The homogeneity and orthogonality of the images were verified. Scanner effects were controlled through two approaches: by including scanner type as a covariate in the VBM regression models and standardizing between the minimum and maximum intensity values of each voxel for all subjects evaluated by each scanner type[2].

### Resting-state fMRI preprocessing

Resting-state sequences were collected for 500 individuals with a mean age of 65.27 years (SD = 12.02), of whom 61.80% were women. Sex information was determined by self-report. 232 individuals with AD, 85 with FTLD, and 183 HC were included (Argentina [$n = 51$, HC:AD/FTLD = 27:24], Brazil [$n = 86$, HC:AD/FTLD = 65:21], Chile [$n = 127$, HC:AD/FTLD = 47:80], Colombia [$n = 202$, HC:AD/FTLD = 24:178], Mexico [$n = 34$, HC:AD/FTLD = 20:14], Supplementary Table 15). Image preprocessing was employed using the fmriprep (version 22.0.2) standard pipeline, encompassing head motion artifacts, slice timing, susceptibility distortion correction, co-registration to the anatomical image, and normalization to standard space, with additional steps in the CONN22.a toolbox[63]. This involved smoothing with a $6 \times 6 \times 6$ mm Gaussian kernel and denoising through linear regression with nine nuisance regressors applied in a single step: six motion parameters (translation and rotation), white matter, cerebrospinal fluid signals, and scrubbing regressors for high-motion time points, and applying a band-pass filter (0.008-0.09 Hz. A motion correction technique was applied by rigidly aligning fMRI volumes to T1-weighted images, ensuring that the impact of motion artifacts was minimized. Motion scrubbing was then applied using framewise displacement >0.2 mm and temporal derivative variance across apace> 5%, which are stricter than conventional thresholds, to flag and remove high-motion frames. The mean proportion of artifact-free to rejected frames was 0.915 (SD = 0.126), with values ranging from 30% to 100%. This was

implemented using the artifact detection tools within the CONN toolbox[63], using a conservative setting (FD = 0.2 mm, global signal $Z = 5$) to remove motion artifacts while preserving the biological signal. Pearson correlation coefficients were computed between the average BOLD time series of each pair of regions of interest (ROIs) from the Brainnetome atlas, a structural and functional connectivity-based parcellation atlas that captures both cortical and subcortical regions, better suited to functional connectivity analysis. AAL atlas cerebellar regions were added, generating a total of 272 × 272 ROIs correlation matrix for each participant. These correlation matrices were Fisher z-transformed to normalize the distribution of the correlation coefficients. Scanner variability was controlled by including scanner type as a covariate in the ROI-to-ROI connectivity regression models.

## Statistical analyses

**Structural equation model.** SEM is a statistical method that facilitates the examination of relationships among observed and latent variables, particularly useful for measuring the effect of each indicator and its cumulative effect through a latent variable by integrating aspects of regression analysis, factor analysis, and simultaneous equation models[64]. The model's structure was based on ten composite variables derived from the MSE questionnaire, which included education, food insecurity, financial status, assets, access to healthcare (randomly set as the fixed indicator to identify the values for the latent factor), childhood labor, subjective SES, childhood experiences, traumatic events, and relationships. These variables served as indicators of a single MSE latent variable, which was used to predict cognition, functional ability, and neuropsychiatric symptoms. The functional ability and neuropsychiatric symptoms scores were inverted to align their interpretation with other measures. Covariances between variables were theoretically guided by the top modification index score. Model fit was assessed using the fit indices comparative fit index (CFI ≥ 0.9), Tucker–Lewis index (TLI ≥ 0.9), root mean square error of approximation (RMSEA ≤ 0.08), and standardized root mean square residual (SRMR ≤ 0.08). We did not consider $\chi^2$ based on its high sensitivity to sample size; as the sample size increases, $\chi^2$ values rise, resulting in lower $p$-values, thus skewing the interpretation of the model's fit.

Separate models were run for each group to examine the specific associations with cognition, functional ability, and neuropsychiatric symptoms. In addition to testing group-specific effects, this approach allows us to isolate the effects within each group. This approach abolishes the confounding factors observed in group comparisons due to age and sex differences. The value of the MSE latent variable for each subject was extracted from these models for subsequent analysis of associations with gray matter and resting-state functional connectivity. Even the smallest sample size (FTLD) met the N ratio of 10:1 for statistical accuracy and power, with 225 participants exceeding the number of estimated parameters (q = 13). Missing data (maximum 19.7%) were imputed across the sample using predictive mean matching through the multivariate imputation by chained equations (MICE) package (3.16.0) in R version 4.3.0. Additional sensitivity analyses were performed to check for bias due to imputation. The main SEM models were run without imputation for the entire sample and each group separately. Results showed consistent associations across all models, indicating that imputation did not introduce bias. (Supplementary Table 16)

**Lasso regression.** Lasso regression was employed to examine the individual contributions of MSE dimensions while controlling for multicollinearity. This method was chosen for its ability to perform variable selection and regularization, shrinking less relevant coefficients toward zero and effectively excluding them from the model. The standardized beta coefficients from the SEM models linking the global MSE score to cognition, functionality, and neuropsychiatric symptoms were compared to the $R^2$ values from the Lasso models that included individual MSE dimensions via meta-regressions. This comparison assessed the global MSE score's explanatory power versus individual dimensions, highlighting the benefits of using an aggregate social exposure metric for understanding multidimensional outcomes.

### Meta-regression comparing SEM and Lasso regression models

Bootstrap resampling ($n = 400$) was applied to estimate the standardized beta coefficients for the paths from the MSE factor to each outcome, capturing the effect sizes and their variability across the bootstrap samples. In parallel, Lasso regression was applied to each bootstrap sample to identify the model's predictive performance using $R^2$, also capturing effect size variability across the bootstrap samples. After obtaining the bootstrap estimates for both SEM and Lasso regression, the results were compared via meta-regressions for each group across outcomes.

### Gray matter associations with MSE

We conducted regression analyses using parametric tests to examine the associations between MSE and brain volume in SPM12, with age, sex, total intracranial volume (TIV), and scanner effects (see Supplementary Table 17 for acquisition parameters) included as covariates of no interest. To enhance behavioral variance and statistical power, HC were merged with AD patients to run VBM models for the AD group, and the process was repeated by pairing HC with FTLD patients for the FTLD model. Multiple comparisons were corrected using the TFCE method, applied via the TFCE toolbox (http://www.neuro.uni-jena.de/tfce). This method integrates voxel and cluster thresholds, eliminating the need for arbitrary cluster formation thresholds and enhancing sensitivity to both focal and diffuse effects, thus optimizing the balance between the FDR and replicability[65]. Statistical significance was assessed through 5000 permutations and set at $P_{FDR} < 0.05$. Extent and height cluster parameters were set to 0.5 and 2, respectively, following standard approaches[66]. Brain imaging results were visualized using MRIcroGL (v1.2.20220720).

### Resting-state functional connectivity associations with MSE

We conducted regression analyses via parametric tests with the MSE score as a predictor of the whole-brain ROI-to-ROI connectivity. The number of connections of each ROI was calculated to assess its contribution to the broader connectivity patterns associated with MSE. Following alternate methods of calculating degree centrality[67,68], we considered edges that were positively or negatively weighted and statistically significant (FDR-corrected). These correspond to values different from zero in the t-statistic matrix, reflecting meaningful ROI-to-ROI relationships. These edges were binarized—assigned a value of one—so that each significant, weighted edge contributes a count of 1 to a node's degree centrality, representing the number of connections each ROI exhibits in relation to MSE. Higher values indicate that a given ROI's connectivity demonstrates more widespread associations with MSE. As for gray matter associations, age, sex, and scanner effect (see Supplementary Table 18 for acquisition parameters) were included as covariates of no interest, and each patient group was analyzed in tandem with HC. All results were $P_{FDR}$ corrected at the connection level for the size of the ROI-to-ROI matrix for each ROI, as implemented in CONN toolbox[63]. Visualization was performed using BrainNet Viewer.

### Sensitivity analysis

To control for the effect of confounding variables, we ran a separate SEM model for the entire sample that included age and sex as predictors of the latent MSE factor, as well as cognition, functional ability, and neuropsychiatric symptoms. To control for disease-related variables, we ran a model with AD and FTLD participants that included disease severity (Clinical Dementia Rating (CDR) total score and CDR-FTLD scale-modified), age at onset, and years since diagnosis, in

addition to age and sex as predictors of MSE, as well as cognition, functional ability, and neuropsychiatric symptoms. Additionally, to examine the potential effect of the FTLD subtype, we implemented a model with FTLD participants, including a binary subtype variable (behavioral variant frontotemporal lobar degeneration *vs.* any other subtype) as a predictor of MSE, as well as cognition, functional ability, and neuropsychiatric symptoms.

To investigate whether MSE was influenced by data quality of structural and functional MRI, we conducted sensitivity analyses examining the relationships between MSE and spatial signal-to-noise ratio (SNR) for both MRI and fMRI, temporal SNR for fMRI, and head motion parameters for fMRI. To evaluate the quality of MRI and fMRI data, we employed the ODQ metric[69]. For MRI data, the assessment focused on the signal-to-noise ratio (SNR) across each slice. The SNR was determined by calculating the mean signal of brain voxels and dividing it by the standard deviation of the signal. For fMRI data, the quality evaluation involved segmenting each time series into 20 repetition times (TR) intervals to compute the temporal signal-to-noise ratio (tSNR)[70]. The tSNR was defined as the mean fMRI signal within each segment divided by its standard deviation. To further test the effect of motion correction, we applied two procedures: rigid realignment of fMRI volumes to T1-weighted images and scrubbing of high-motion frames. Scrubbing was based on framewise displacement >0.2 mm and global signal change >5% (Methods, Neuroimaging Acquisition and Preprocessing). Participants with less than 70% of artifact-free frames were excluded (Supplementary Tables 7-8). Scanner scaling approaches were applied to MRI data to reduce inter-scanner variability[2] (Supplementary Tables 9 and 10). The minimum and maximum voxel intensity values were extracted from all subjects belonging to each scanner type, which were then scaled between the minimum and maximum intensity values specific to each scanner type. Then, voxel-based morphometry analyses were replicated with the scaled images. This method accounts for differences in location and scale (e.g., mean and variance) between sites, reducing scanner-related variability. This process helps to minimize the variability introduced by different scanners[2,20].

**Reporting summary**

Further information on research design is available in the Nature Portfolio Reporting Summary linked to this article.

## Data availability

The preprocessed behavioral, MRI, and fMRI data generated in this study have been deposited in the OSF database under accession code https://osf.io/78ng6/. The raw data are available under restricted access due to ethical and regulatory constraints. Access can be obtained after IRB approval of the formal data-sharing agreement in a process that can last up to 12 weeks. For questions related to the data request and usage, contact Agustín Ibáñez at agustin.ibanez@gbhi.org. Source data are provided with this paper.

## Code availability

The code used to preprocess and analyze the data of this work is available in an Open Science Foundation repository at the following address: https://osf.io/78ng6/

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

## Acknowledgements

Data in this manuscript were collected by the MULTI-PARTNER CON-SORTIUM TO EXPAND DEMENTIA RESEARCH IN LATIN AMERICA (ReDLat), supported by Fogarty International Center (FIC), National Institutes of Health, National Institutes of Aging (R01 AG057234, R01 AG075775, R01 AG21051, R01 AG083799, CARDS-NIH 75N95022C00031), Alzheimer's Association (SG-20-725707), Rainwater Charitable Foundation – The Bluefield project to cure FTD, and Global Brain Health Institute)]; AI is supported by ANID/FONDECYT Regular (1250091, 1210176, and 1220995) and ANID/FONDAP/15150012. CDA is supported by ANID/FONDECYT Regular 1210622. CDA and AI are supported by grant ANID/PIA/ANILLO ACT210096. The contents of this publication are solely the author's responsibility and do not represent the official views of these institutions.

## Author contributions

Joaquin Migeot: Conceptualization, Methodology, Software, Formal analysis, Writing— Original Draft, Visualization. Stefanie D. Pina-Escudero: Conceptualization—Writing secondary Draft. Hernan Hernandez: Writing—Review & Editing, Validation, Formal analysis, Methodology, Software. Raul Gonzalez-Gomez: Data Curation, Writing—Review & Editing, Formal analysis, Software. Agustina Legaz: Writing—Review & Editing, Software. Sol Fittipaldi: Writing - Review & Editing. Elisa de Paula França Resende: Writing—Review & Editing. Claudia Duran-Aniotz: Writing - Review & Editing. Jose Alberto Avila-Funes: Writing—Review & Editing, Resources. Maria I. Behrens: Writing—Review & Editing, Resources. Martin A. Bruno: Writing—Review & Editing, Resources. Juan Felipe Cardona: Writing—Review & Editing, Resources. Nilton Custodio: Writing—Review & Editing, Resources. Adolfo M. García: Writing—Review & Editing, Resources. Maria E. Godoy: Writing—Review & Editing, Project administration. Kun Hu: Writing—Review & Editing, Resources. Serggio Lanata: Writing—Review & Editing. Brian Lawlor: Writing—Review & Editing. Francisco Lopera: Writing—Review & Editing, Resources. Marcelo Adrian Maito: Writing—Review & Editing, Data Curation. Diana L. Matallana: Writing—Review & Editing, Resources. Bruce Miller: Writing—Review & Editing, Funding acquisition, Project administration. J. Jaime Miranda: Writing—Review & Editing. Maira Okada de Oliveira: Writing—Review & Editing. Pablo Reyes: Writing—Review & Editing. Hernando Santamaria-Garcia: Writing—Review & Editing, Resources. Andrea Slachevsky: Writing—Review & Editing, Resources. Ana L. Sosa: Writing—Review & Editing, Resources. Leonel T. Takada: Writing—Review & Editing, Resources. Jacqueline M. Torres: Writing—Review & Editing. Sven Vanneste: Writing—Review & Editing. Victor Valcour: Writing—Review & Editing. Olivia Wen: Writing—Review & Editing. Jennifer S. Yokoyama: Writing—Review & Editing. Katherine L. Possin: Writing—Review & Editing, Supervision. Agustin Ibanez: Conceptualization, Methodology, Writing—Original Draft, Visualization Review & Editing, Supervision, Project administration, Funding acquisition.

## Competing interests

The authors declare no competing interests.

## Additional information

¹Latin American Brain Health Institute (BrainLat), Universidad Adolfo Ibáñez, Santiago de Chile, Metropolitan Region of Santiago, Santiago 7910075, Chile. ²Global Brain Health Institute (GBHI), Trinity College Dublin, Dublin, Dublin 2, Ireland. ³Global Brain Health Institute, University of California, San Francisco, CA 94158, USA. ⁴Memory and Aging Center, Department of Neurology, University of California, San Francisco, CA 94158, USA. ⁵Cognitive Neuroscience Center, Universidad de San Andrés, Ciudad Autónoma de Buenos Aires, Buenos Aires 1644, Argentina. ⁶Universidade Federal de Minas Gerais, Belo Horizonte, Minas Gerais 31270-901, Brazil. ⁷Dirección de Enseñanza, Instituto Nacional de Ciencias Médicas y Nutrición, Salvador Zubirán, Ciudad de México, Mexico City 14000, México. ⁸Departamento de Neurociencia, Faculty of Medicine, University of Chile, Santiago de Chile, Metropolitan Region of Santiago, Santiago 8380453, Chile. ⁹Centro de Investigación Clínica Avanzada (CICA), Universidad de Chile, Santiago de Chile, Metropolitan Region of Santiago, Santiago 8380453, Chile. ¹⁰Departamento de Neurología y Psiquiatría, Clínica Alemana-Universidad del Desarrollo, Santiago de Chile, Metropolitan Region of Santiago, Santiago 7550000, Chile. ¹¹Departamento de Neurología y Neurocirugia, Hospital Clínico Universidad de Chile, Santiago, Chile. ¹²Instituto de Ciencias Biomédicas, Universidad Católica de Cuyo, San Juan J5400, Argentina. ¹³Facultad de Psicología, Universidad del Valle, Cali, Valle del Cauca 760032, Colombia. ¹⁴Unit Cognitive Impairment and Dementia Prevention, Peruvian Institute of Neurosciences, Lima 15046, Peru. ¹⁵Departamento de Lingüística y Literatura, Facultad de Humanidades, Universidad de Santiago de Chile, Santiago de Chile, Metropolitan Region of Santiago, Santiago 9170124, Chile.

[16]Department of Anesthesia, Critical Care and Pain Medicine, Massachusetts General Hospital, Harvard Medical School, Boston, MA 02114, USA. [17]Neuroscience Research Group (GNA), Universidad de Antioquia, Medellín, Antioquia 050010, Colombia. [18]Departamento de Salud Mental, Hospital Universitario Fundación Santa Fe, Bogotá, Colombia. [19]Centro de Memoria y Cognición Hospital Universitario San Ignacio, Bogotá, Colombia. [20]Sydney School of Public Health, Faculty of Medicine and Health, University of Sydney, Sydney, Australia. [21]Cognitive Neurology and Behavioral Unit (GNCC), University of São Paulo, São Paulo 05508-000, Brazil. [22]Pontificia Universidad Javeriana, Bogotá, D.C 110311, Colombia. [23]Hospital Universitario San Ignacio, Center for Memory and Cognition, Intellectus, Bogotá, D.C 110231, Colombia. [24]Geroscience Center for Brain Health and Metabolism (GERO), Santiago de Chile, Metropolitan Region of Santiago, Santiago 8331150, Chile. [25]Memory and Neuropsychiatric Center (CMYN), Neurology Department, Hospital del Salvador & Faculty of Medicine, University of Chile, Santiago de Chile, Metropolitan Region of Santiago, Santiago 7500921, Chile. [26]Neuropsychology and Clinical Neuroscience Laboratory (LANNEC), Physiopathology Program – Institute of Biomedical Sciences, Neuroscience and East Neuroscience Departments, Faculty of Medicine, University of Chile, Santiago de Chile, Metropolitan Region of Santiago, Santiago 8380453, Chile. [27]Servicio de Neurología, Departamento de Medicina, Clínica Alemana-Universidad del Desarrollo, Santiago de Chile, Metropolitan Region of Santiago, Santiago 7550000, Chile. [28]Instituto Nacional de Neurología y Neurocirugía Manuel Velasco Suarez, Secretaría de Salud de México, Ciudad de México, Mexico City 14269, México. [29]Department of Epidemiology & Biostatistics, University of California, San Francisco 94143, USACA. [30]Lab for Clinical and Integrative Neuroscience, Trinity College Institute for Neuroscience, Trinity College Dublin, Dublin D02 PN40, Ireland. [31]School of Psychology, Trinity College Dublin, Dublin D02 PN40, Ireland. [32]Department of Psychology and Neuroscience, Boston College, Chestnut Hill, MA 02467, USA. [33]These authors contributed equally: Joaquin Migeot, Stefanie D. Pina-Escudero, Hernan Hernandez ✉e-mail: agustin.ibanez@gbhi.org

