## [Transparent Peer review file · Nature Communications]

Social exposome and brain health outcomes of dementia across Latin America

Corresponding Author: Dr Agustín Ibáñez

A version of this paper was originally rejected for publication by Nature Communications, however that decision was reconsidered after appeal by the authors.

Version 0:

Reviewer comments:

Reviewer #1

(Remarks to the Author)

Thanks for the opportunity to review this paper.

This is an interesting study about the effect of social exposome on brain health outcomes of dementia in some countries in Latin America. The authors use a series of variables measuring some risk factors for dementia and socioeconomic variables to predict cognition, functionality and neuropsychiatric outcomes in healthy individuals as well as patients with AD and FTD. However the study is quite interesting and methodological well executed, I have some comments and suggestions.

1) Fundamentally, the authors want to resume in a unique variable, called multidimensional social exposome (MSE), several factors that may influence the cognition in individuals with and without dementia. However, it is hard to understand this final MSE variable given that the variables from which it is formed are so different from each other and in fact represent different constructs. For example, education, traumatic events, childhood experiences and assets can be understandable as dementia risk factors; however, nutrition suffers from an obvious reverse causality in patients with dementia, that is, weight loss as dementia begins, or even before dementia as an effect of the incipient neurodegenerative process. Others, for instance, access to health care or subjective SES can represent a factor related to worsening of dementia symptoms, not just a risk factor.

I think the authors should clarify these complexities.

2) The effect of education on cognition in healthy individuals can be explained simply because the effect of education on MMSE is quite high, reason why a different cutoff is commonly used to understand the effect of this scale. Maybe the authors can make a sensitivity analysis using the MMSE as a categorical variable, as a way to remove the effect of education, which unsurprisingly has the biggest effect on this analysis.

3) However the variables that authors evaluated in this work are interesting, several other risk factors that may have a higher impact on cognition and functionality are missed. For example, what are the effects of hypertension, diabetes, social interactions and all the well-known dementia risk factors? The MSE variable would still be able to predict the outcomes if those variables were included in the model. I suggest, to make the analysis still more robust, the authors include all risk factors available in a sensitivity analysis.

4) Specifically in AD patients, one of the weaknesses is the lack of AD biomarkers to make a biological diagnosis. Considering that around 30% of individuals can be misdiagnosed solely on clinical grounds, it is a weakness that deserves mention.

5) About the neuroimaging analysis. It is interesting that cerebellar areas are so correlated with the MSE in this sample. The authors should discuss this.

6) About the figures. However the objective seems to make them comprehensive, they sometimes can be hard to understand

and are quite polluted. I suggest simplify them. Additionally, all the abbreviations should be explained in the figures.

Reviewer #2

(Remarks to the Author)

The authors report on the relationship between multi-dimensional social exposome and cognitive and brain function, which they term "brain health," in a multi-site cohort of healthy and demented subjects from Latin America. The authors describe a new questionnaire called the "multidimensional social exposome" (MSE) assessment which they use to evaluate social and environmental inequities for cognitively intact and demented adults. Cross-sectional analyses were grouped into two themes, 1) MSE associations with cognitive, neuropsychiatric, and functional ability measures, and 2) MSE associations with structure and functional brain measures.

This study is staged to address a potential gap in our understanding of the impact of the social exposome on cognitive and brain health in an older adult sample from Latin America. However, the manuscript has several considerable conceptual and methodological limitations. Relatedly, the description lacks sufficient detail to allow replication.

While the authors focus on their own prior work, several studies have described the effects of social factors on cognitive and structural and functional brain health in aging (Avila et al., *Alzheimer & Dementia* 2021; Binnewies et al., *Brain Research Bulletin* 2023; Chan et al., *Nature Aging*, 2021; Soh et al., *JAMA Neurology* 2023; Walhovd et al., *Cerebral Cortex* 2022). It is not clear how the MSE approach builds on the limitations of prior work which have used summary measures of exposures (e.g., income, education). I agree with the value of examining multi-dimensional exposures and appreciate the focus on participants from Latin America, which follows several related papers from this group of authors. However, the current work lacks in demonstrating information over and above traditional measures such as SES. As it rests, the paper also ultimately constitutes a large listing of MSE-brain/cognition relationships. It is not clear what exactly is learned beyond what has been demonstrated in work examining social determinants of brain health disparities in older adults, and how these specific sets of relationships should be considered.

Additional methodological/reporting issues:

- a. Distribution of healthy and diseased older adult subjects pooled across all six sites for each analysis family is not described in detail. For an effort of this size, greater details about demographics and within sample distinctions should have been included.
- b. The justification and procedure for calculating degree centrality and subsequent analyses of degree centrality with MSE to understand the results reported on p12-13 and Figure 4 are not provided. This seems like a very opaque way of examining functional correlations given the dependence on thresholding edges.
- c. The ten-factor structure of MSE used in this study is not clearly described. Two sources are cited, the first source - cited as 85, Pina-Escudero et al., 2023 is an abstract reporting only three latent MSE factors, not ten, are valid with caregiver input. The second source - citation 86, is a manual about research practices and does not appear to be specific to MSE.
- d. The procedure to discard (p22) some MSE items is not described.
- e. Steps for data harmonization between sites mentioned on p28 and separately on p32 are not clear. For example, it is stated (p28) that further modeling of site was used in the LASSO regression for structural brain measures (not SEM?). Is this because site effects remained? How much variance did site explain in your models before and after adjustment? Along these lines it is not clear that re-scaling MRI measures using a min and max within scanner value (p31-32) procedure adequately achieves the goal of harmonizing data across sites.
- f. T1w MRI processing is not clearly described on p28. For example, authors say " a 6 x 6 x 6 mm Gaussian kernel was used to smooth grey matter segmentations" are you referring to just grey matter or some kind of grey matter atlas?
- g. Rest-state nuisance regression is not described in sufficient detail for replication on p28. For example, were eight or nine regressors used in a single step? Is "motion scrubbing" referring to a spike regressor or the removal of frames that meet a certain threshold after nuisance regression?
- h. Why was the AAL atlas chosen for functional connectivity analysis (p28)? AAL do not correspond to functional areas and violates known biological principles of functional organization. This makes the resting-state analyses uninterpretable.
- i. The imputation procedure was not well described on p30-31:
 - a. Was it within disease state, within site, or across sample?
 - b. How was 20% imputation, above the 5-10% recommended threshold, justified (Jakobsen et al., *BMC Medical Research Methodology* 2017)?
- j. Multiple comparison correction for resting state MRI functional connectivity is described with threshold free cluster estimated (TFCE) voxel-based correction p32. However, TFCE relies on spatial information not found in a functional connectivity matrix.
- k. Many of the tables and figures lack clarity in their representation of methodology, analysis, and results. For example:
 - a. Table 2 results cover a SEM path analysis and LASSO regression for "Global Score" and "individual predictors." Yet, there is neither a table legend or explicit in-text description of what "Global Score" and "individual predictors" are. Moreover, it is not clear why "individual predictors" are a single value in Table 2 when p9-10 describe results for different individual predictors that differ between analysis groups (i.e., all subjects, healthy, and demented).
 - b. Figure 3 does not provide x and y-axis labels for regression plots so authors must assume x-axis is some individual brain structure's volume regressed overall MSE composite. However, a toy representative plot in the bottom of Figure 3 may be a legend for regression plots except they say "MSDH" on y-axis relating to Cluster GMV on x-axis.

Reviewer #3

(Remarks to the Author)

This study investigates the influence of the social exposome – an aggregate of exposures/experiences across the lifespan,

like nutrition, education, finances, and healthcare access – on cognition and brain health among healthy controls and people diagnosed with Alzheimer’s dementia or frontotemporal lobar degeneration residing in various countries in Latin America. While we commend the authors for these large cross-national efforts and the richness of the data collected (i.e., social factors, diagnostic, neuroimaging), there are significant limitations in their approach that obscure any potential meaningful contribution this study can offer to improving our understanding of global cognitive and brain health.

Major Comments:

First, the authors group various diverse Latin American countries together to represent Latin America as a whole, when Latin America is far from a homogenous construct. How do the findings reported in the manuscript provide any additional help to further our understanding of cognitive health among these Latin American countries? All countries listed here have very diverse, historical and current, sociopolitical and sociocultural contexts that likely would influence unique and shared pathways that impact late-life cognitive health. A more appropriate and informative approach would be to leverage this large cohort for cross-national comparisons. For instance, are the results reported here the same across each of the countries? For instance, educational and healthcare policies are different between these countries, would that not suggest differences in the strength of the associations reported? Would that not provide more valuable information to the aging communities in the individual Latin American countries and their respective policy makers to effect change for the benefit of their communities? I recommend the current article in guidance on best approaches for cross-national analyses <https://doi.org/10.1002/alz.13694>.

Second, sample sizes are relatively modest within each country, yet they are described as if they represent the given country and thus represents Latin America. What efforts have been made to demonstrate that these samples are representative of their country? If they are not meant to be representative, that is fine, but then interpretation and messaging of findings should be modified slightly as to make sure to not indicate that these findings represent Latin America as their introduction and discussion states. I recommend the following article regarding issues related to representativeness <https://doi.org/10.1002/dad2.12450>.

Third, to further enhance the contribution and differences between each country, information regarding recruitment approaches by country should be described and provide information whether the distribution of the MSE scores were similar across country or not. For instance, authors describe that they ran sensitivity analyses to address recruitment bias, yet they do not define what is meant by recruitment bias and no such analysis is reported. Similarly, it would be helpful to know whether participants are recruited from one particular city in each country or multiple cities? Primarily urban settings or rural settings as well? What are the sociodemographic characteristics of the different cohorts? Along those same lines, it would be informative to know about differences in the MSE characteristics by country.

Fourth, the construct of the social exposome is not clearly defined nor supported. In the introduction, the authors do not define the social exposome. It would be helpful to provide support for why this composite of various social factors is more informative than evaluating each factor jointly? To gain better understanding on how and where to intervene, we need to understand the contribution of individual social factors on health. In fact, the authors do provide in their results information on which MSE indicators are driving their findings, then, why use the MSE composite score at all? For instance, a top indicator was education, but education comprised of several different aspects from years of schooling, self-reported quality of education, parental years of education, so what about education might be driving this result? How can one intervene in terms of education then? And would the intervention be the same across all of Latin America?

Fifth, the development and validity of the MSE is unclear. The authors reported that they created composite factors, but that wording suggests they were derived through a factor analysis when they were not. These composite scores were derived through consensus agreement and summing of each item. Why not employ psychometric techniques to determine the structure of the proposed measure such as using EFA and CFA techniques to determine that the theoretically proposed groupings do hang together psychometrically? And are these factors measuring truly the same thing between countries? Greater information regarding their statistical harmonization approaches is warranted. I recommend the following readings as helpful guidelines <https://doi.org/10.1037/neu0000816> and [10.1016/S2666-7568\(23\)00170-8](https://doi.org/10.1016/S2666-7568(23)00170-8)

Minor Comments:

In Figure 1 for their Outcomes predictions figure shouldn’t the labels be flipped and MSE factor score be on the X axis? It is my understanding that they are using MSE to predict the outcomes of cognitive status, functional ability, and neuropsychiatric symptoms, right? If not, then please clarify.

Across their assessments of lifecourse experiences there is a huge age gap between age 10 and 35. Unclear why they do not assess for experiences during that large and formative age period.

There are several sentences that require additional information to help clarify what was being meant:

Authors state in their introduction that LA is one of the most unequal regions in the world, but unequal in what way? And are those inequalities the same across the various countries in Latin America? These are very large overgeneralizing sweeping comments that obscure the heterogeneity within Latin America.

“Other reports suggest stronger effects of SDH than ancestry for dementia risk in LA” authors should specify that this study measured genetic ancestry.

Sentence: "Thus, incorporating the multiple dimensions of social exposome and assessing their potential impact on brain health outcomes of aging and dementia in underrepresented populations is critically needed to develop robust tailored models." Tailored models of what?

Reviewer #4

(Remarks to the Author)

In this multi-centre study Migeot and coauthors collect in six Latin American countries a rich set of "social exposome" variables using a fit-for-purpose questionnaire alongside demographic, clinical and brain imaging data to address the question about their association with two types of dementia – Alzheimer's disease (AD) and frontotemporal lobar degeneration (FTLD). Using structural equation modeling, the obtained "multidimensional social exposome (MSE)" construct showed association with poorer cognition in healthy aging, mainly modulated by variables related to different measures of education and SES, while in AD and FTLD it also demonstrated associations with functional abilities and neuropsychiatric symptoms. The brain anatomical and functional associations were confined to frontal and cerebellar regions in AD, and fronto-temporo-limbic and cerebellar regions in FTLD.

This is a potentially interesting study, shedding light on parts of the world that are clearly underrepresented in current epidemiological research. In my view, a "head-to-head" comparison with similar open access data (e.g. UK Biobank, ADNI) would help convincing the readership on some of the critical aspects outlined below. The study strongly relies on subjective data with a major part of events dating back many decades before the current assessment. Given that the focus is on dementia – Alzheimer's and FTLD, I remain skeptical about the validity of the obtained information given a strong disease-specific bias. Similarly, the diagnosis of dementia is denoted with "probable". The absence of fluid biomarkers results, together with the missing information about the presence of small vessel disease and the marked age differences between cohorts raise questions about the validity of the obtained results.

1. There is a significant age difference between the cohorts of healthy controls, FTLD and AD, which mounts up to a decade.
2. The expected higher ratio of women in the AD cohort established in epidemiological studies on AD does not substantiate here. Could you please expand on this?
3. In patients with dementia, one would expect difficulties correctly recalling items from the distant and near past. Similarly, individuals with FTLD would tend to differentially recall traumatic events from the past given predominant emotional regulation deficits compared to the other two cohorts. In my view, this represents a significant bias. What measures were taken to mitigate these effects?
4. Could the authors provide a matrix with correlations between the individual variables of the social exposome. Given that the observed effects of nutrition, financial status, subjective SES, and access to healthcare are tightly linked to socio-economic status, I wonder about the amount of shared variance between them.
5. "To enhance behavioral variance and statistical power, we analyzed 720 each patient group (AD and FTLD) in tandem with HCs." Could you please clarify the design matrix for VBM analysis? Did it include all 875 individuals categorized in three different groups according to their disease label or did you create two identical designs with healthy controls and either AD or FTLD?
6. The brain anatomical patterns associated with MSE in both clinical cohorts are unexpected. If MSE in both AD and FTLD is closely associated with cognitive function, functional ability, and neuropsychiatric symptoms, what might be the explanation for the differential anatomical pattern?
7. What is the rationale behind the use of TFCE correction in a well-powered cohort of more than 870 individuals?
8. Were the results of the sensitivity analyses also corrected for multiple comparisons? If yes, please state at which threshold and if FWE, FDR or other method.
9. How was "disease severity" (line 340) measured? If based on MMSE for AD, what was the metric used in patients with FTLD?
10. Could you please elaborate how exactly «scanner type» was included as a covariate of no interest in the brain imaging analyses?
11. What criteria were applied for Quality Assessment of structural and functional MRI data?
12. I wonder about the added value of subjective SES compared to the "objective" SES. Could you please elaborate on this?

Minor

The manuscript needs serious language editing. There are also numerous missed prepositions, tautologies and orthographic errors in both text and figures.

Version 1:

Reviewer comments:

Reviewer #1

(Remarks to the Author)

The authors responded appropriately to the reviewers' comments and in my view the manuscript improved substantially. I don't have any more questions or suggestions to make.

(Remarks on code availability)

Reviewer #2

(Remarks to the Author)

The authors present a revised manuscript detailing an examination of the relationship between the multi-dimensional social exposome and cognitive function and brain structure/function, which they term "brain health," in a multi-site cohort of healthy and demented subjects from Latin America.

The authors have provided responses to several reviewer requests that address some of the identified concerns. However, they have decided to not address previously identified issues related to parcel definition, which is critical for interpreting the brain's functional connectivity measures. In addition, justification for several data processing decisions reveal that the procedures are not appropriate. Based on this I do not accept this manuscript's conclusions; I primarily focus on these issues in my comments below.

1. I agree with authors that no single field standard on atlases exist. However, their rationale does not preclude ensuring the biological relevance of the atlas being used to examine correlations in activity among brain areas. It is known that brain areas do not often follow morphometric boundaries (as used in AAL) which predicated the need for the development of a new generation of atlases for purposes of localization and interrogation in neuroimaging, roughly 15 years ago. The authors respond that "no single parcellation method consistently outperforms others across metrics". However, while prior work has convincingly demonstrated that functional atlases can vary across metrics of validation, they still outperform the AAL atlas which is what the current conclusions rest on (e.g., Arslan, Neuroimage, 2018). The authors go on to indicate that "[they] selected the AAL atlas to ensure comparability with existing literature. This choice also allowed us to use the same atlas for structural and functional imaging analyses, aligning our study with previous works in dementia." The vast majority of the literature has moved away from this outdated atlas; further, the alternate atlases do not preclude examining the same regions for structural and functional imaging analyses as the authors contend. In summary, the functional connectivity results contain a known problem in their estimation, due to the use of inappropriate nodes. This issue is compounded by additional methodological issues which are described further below.

2. The authors have provided some data comparing MSE to individual metrics alongside information on how the MSE improves on prior approaches that incorporate summary measures of SES. However, the authors do not elaborate on what prior work has told us about the exposome and brain structure, function, cognition, and dementia relations, and how their work fits in with these findings, which is a critical part of scholarship.

3. Additional methodological issues:

a) Authors chose to point to the CONN toolbox manual for reviewers to learn the TFCE procedure instead of providing an explicit explanation. CONN TFCE expects a group level clustered T or F statistical matrix with a user selected h parameter to smooth over matrix data for later permutation testing at a given FDR threshold. Authors chose to not describe their cluster arrangement, analysis performed to generate each statistical matrix, or the necessary h parameter selected. The lack of detail prevents readers from replicating their atypical correction for multiple comparisons procedure.

b) The procedure for calculating degree centrality for later testing with MSE is still poorly justified and described. For instance, did authors use their TFCE corrected test statistic matrix as a threshold, such that only significant "clusters" within AAL atlas lobes were summed in degree centrality calculation (assuming the matrix was spatially organized somehow, which seems impossible to achieve)? Similarly, was degree centrality calculated with a weighted (i.e., t-value) or unweighted (i.e., binary) matrix? Too few details are given to support future replication and the absence of adequate explanation of these critical aspects of analysis diminishes confidence in the results.

c) Authors clarify that their "motion scrubbing" procedure refers to spike regression of frames flagged at .05mm FD or 3 standard deviations away from mean global signal - which is a very lenient motion scrubbing (i.e., removal of frames) procedure. Given that older adults express higher rates of motion, the threshold is far too liberal to accept the results as reported (see Ciric et al., 2017). As a result, it is likely that the dataset contains many erroneous correlations that are being attributed to biological signal. Future efforts should work to ensure that resting-state data are adequately cleaned using appropriate data processing parameters.

(Remarks on code availability)

Reviewer #3

(Remarks to the Author)

Dear authors,

I greatly appreciate the time and effort you all put in to address my comments as well as those of other reviewers. Excellent job.

I just have very minor lingering comments.

When you described the model fit for your sensitivity analysis for country effects in your supplementary information, you described that "The model showed good fit: CFI = 0.868, TLI = 0.767, RMSEA = 0.088, SRMR = 0.061". However, CFI/TLI are in the poor range and RMSEA/SRMR are bordering acceptable.

Lastly, when describing the recruitment efforts, authors mentioned that participants were recruited "from extensive networks". A bit more information regarding what is meant by extensive networks, like what kind of networks? would be helpful for future investigators seeking to work in LA.

(Remarks on code availability)

Reviewer #4

(Remarks to the Author)

This is the revised version of the manuscript that I had the chance to review previously. While seemingly exhaustive, the answers to my queries and points of criticism are mostly vague. Most importantly, the way addressing the socio-geographical aspect of the study with focus on Latin America drawing up individual-level conclusions from group level geo-spatial data falls under the description "ecological fallacy". If the goal is to address risk exposures adequately, then we need a true geo-spatially referenced analysis where within- and across-countries differences and similarities will be explicitly tested on a geo-coded reference basis.

One of my suggestions to compare the data at hand with other large-scale open access epidemiological data - e.g. the UK Biobank, was not followed suit. Similarly, there are still many orthographic mistakes - see "finanTial status" in Fig.2 and counterintuitive brain morphometry results (Fig. 3) that show implausible association patterns of the multi-dimensional social exposome with brain structure in patients with AD and FTLD.

(Remarks on code availability)

Version 2:

Reviewer comments:

Reviewer #2

(Remarks to the Author)

The authors present a second revised manuscript detailing an examination of the relationship between the multi-dimensional social exposome and cognitive function and brain structure/function, which they term "brain health," in a multi-site cohort of healthy and demented subjects from Latin America. The authors have provided responses to several more reviewer requests that address some but not all concerns. There remain vague statements and missing information on the procedures used that need clarity for future replication of their work, detailed below.

1. There are several instances where authors use terms like "hypoconnectivity" and "hyperconnectivity" that imply a comparison occurred such that x group connectivity measure was 'reduced' or 'increased' relative to y group measure, respectively. The authors should be more direct to avoid aid understanding of results, so readers understand the context of the comparison made in the text where it is stated. Examples can be found on pgs 11, 14, 17, 18.
2. Regarding degree centrality, which is now termed "number of connections", the concern was over the procedure described to calculate degree centrality. Right now, I assume the density threshold is based on only significant, positively weighted (not absolute value) FDR corrected edges (the Fisher's z correlation matrix or stat matrix t value where cell values are above 0 and contain information on roi to roi relations), that were binarized and set to a value of one, so each node's significant positively weighted edge could contribute a count of 1 to the degree centrality or "number of connections" for a single node. If this is accurate to what was done using the CONN toolbox, it should be stated as such so others can understand the procedure and replicate the work. Additional references that describe alternate methods of calculating degree centrality include: Guimerà & Nunes Amaral, 2005; Sporns, 2018). Again, the number of connections is not equivalent to a static functional connectivity measure or total connectivity strength (sum of weighted edges where edge is a continuous value) and should not be interpreted as such as authors suggest on pg 34.
3. I appreciate the responsiveness to more rigorous resting-state preprocessing (particularly the scrubbing threshold). Please provide additional details about frame retention so that readers can understand data quantities that go into analysis (range of resting-state frames remaining, threshold of minimum number of 'cleaned' frames allowed for analysis, etc).

Guimerà, R., & Nunes Amaral, L. A. (2005). Functional cartography of complex metabolic networks. *Nature*, 433(7028), 895–900. <https://doi.org/10.1038/nature03288>

Sporns, O. (2018). Graph theory methods: Applications in brain networks. *Dialogues in Clinical Neuroscience*, 20(2), 111–121. <https://doi.org/10.31887/DCNS.2018.20.2/osporns>

(Remarks on code availability)

Reviewer #4

(Remarks to the Author)

I greatly appreciate the authors' effort to align their responses to my points of criticism, but I have to insist that the responses miss the point.

1. Wrt "ecological fallacy" – the issue here is that we need a subject-level analysis that tests each individual against the neighbourhood within a spatial buffer (e.g. btw 100 and 1000 meters diameter).
2. The comment on the implausibility of MSE-brain structure correlations with fronto-temporal patterns in AD and insular regions in FTLD on the background of strong cerebellar associations was not addressed.

On a side note – given the inequalities in health care access, is there a possible bias in either the dementias sample that will drive the less socio-economically advantaged or the disadvantaged to seek diagnostic work-up? Similarly, in clinical neuropsychology testing we adjust the test results for the educational level, which is also recommended for screening tests like MMSE and MoCA in the form of reducing the maximum expected score of 30.

(Remarks on code availability)

n.a.

Version 3:

Reviewer comments:

Reviewer #2

(Remarks to the Author)

The authors have adequately addressed my comments. I appreciate their efforts in revising their manuscript.

(Remarks on code availability)

Reviewer #4

(Remarks to the Author)

Many thanks for the detailed responses on my queries and I appreciate the extensive work behind it. I still remain sceptical about the representativity of these results for "brain health outcomes of dementia across Latin America", but history will judge.

(Remarks on code availability)

Reviewer #5

(Remarks to the Author)

(Remarks on code availability)

RESPONSE TO REVIEWERS

The following responses have been prepared to address the Reviewer's comments and suggestions point-by-point. All additions and modifications are highlighted in yellow here and in the revised version of the manuscript. All new references have been incorporated into the revised 'References' section.

REVIEWERS COMMENTS

REVIEWER #1

Comment 1.1. Thanks for the opportunity to review this paper. This is an interesting study about the effect of social exposome on brain health outcomes of dementia in some countries in Latin America. The authors uses a series of variables measuring some risk factors for dementia and socioeconomic variables to predict cognition, functionality and neuropsychiatric outcomes in healthy individuals as well as patients with AD and FTD. However the study is quite interesting and methodological well executed, I have some comments and suggestions.

Response 1.1. We are grateful for your revision. Your thoughtful suggestions have been valuable, and we have addressed them as detailed below.

Comment 1.2. Fundamentally, the authors want to resume in a unique variable, called multidimensional social exposome (MSE), several factors that may influence the cognition in individuals with and without dementia. However, it is hard to understand this final MSE variable given that the variables from which it is formed as so different from each other and in fact represent different constructs. For example, education, traumatic events, childhood experiences and assets can be understandable as dementia risk factors; however, nutrition suffer for a obvious reverse causality in patients with dementia, that losses weight as dementia begin, or even before dementia as a effect of the incipient neurodegenerative process. Others, for instance, access to health care or subjective SES can represent a factor related to worsening of dementia symptoms, not just a risk factor. I think the authors should clarify this complexities.

Response 1.2. Thank you for this comment. We have provided further explanation of the MSE and the additional steps taken to avoid reverse causality or biased associations.

(a) We defined MSE explicitly and explain how different factors are included in this construct:

Introduction

[...] The cumulative and multidimensional impact of the social exposome—a subset of the exposome (the totality of environmental, social, and biological exposures over a lifetime)—captures socially driven exposures affecting health outcomes¹⁻⁴. The cumulative effects of the social exposome may exert a more substantial influence on aging and dementia than isolated risk factors⁵⁻⁷. For instance, individuals growing up in poverty face limited educational opportunities and food insecurity, relying on low-cost, nutrient-poor diets that impair brain development^{6,8}. In adulthood, chronic stress from traumatic events and financial insecurity add to this burden⁹, while in older age, accumulated adverse exposures throughout life contribute to physiological dysregulation, increasing vulnerability to neurodegeneration¹⁰. The cumulative burden of socially related factors (e.g., low educational attainment⁶, adverse childhood experiences¹¹, traumatic events¹²), aggravating factors (e.g., limited access to healthcare¹³), and affected domains (e.g., financial burden¹⁴, reduced social interaction¹⁵) may significantly exacerbate dementia phenotypes. These cumulative social burdens constitute the multidimensional social exposome (MSE)¹⁶. [...]

(b) In **response 2.6** we describe items exclusion to avoid reverse causality. We have carried out new analysis excluding dimensions that may bias the association between MSE and outcomes in **response 1.3** and **2.2**.

(c) Nutrition category refers to the access to food across the lifespan, including questions such as if the participant had to eat less or less healthy due to economic hardship, thus reverse causality does not apply. To be more precise, we have changed its name from "nutrition" to "food insecurity" across the text and figures.

Comment 1.3. The effect of education on cognition in healthy individuals can be explained simply because the effect of education on MMSE is quite high, reason why a different cutoff are commonly used to understand the effect of this scale. Maybe the authors can make a sensitivity analysis using the MMSE as a categorical variable, as a way to remove the effect of the education, which unsurprisingly has the biggest effect on this analysis.

Response 1.3. Thank you for this observation. We have clarified that our education measure goes beyond years of schooling, explaining also why standardizing MMSE by education can induce circularity (education is a component of MSE) and run new sensitivity analyses excluding education from MSE. This analysis showed that associations between MSE and cognition remain consistent, ruling out any biased effect of education.

(a) We have clarified that our measurement of education:

Discussion

[...] Most previous studies¹⁷⁻²² have used years of education, and we expanded these effects with a combined measure that incorporates multiple domains (school type and location, perceived education quality throughout life, the number of books at home during childhood, parental education levels, and years of formal schooling). [...]

(b) Standardizing MMSE by education is circular in our context, as education is a component of the MSE. The expected association between the MSE score and cognition in HC and the small number of subjects under the categorical cutoff of the MMSE score based on years of education adds limitations to a categorical conversion as detailed below:

1. Empirical testing confirmed an association between MSE and years of education, violating the principle of non-collinearity between predicted and predictor variables:

2. Applying cutoffs based on years of education²³ would comprise only 100 HCs (out of 1775) to be categorized below the cutoff. Black lines indicate the cutoff point for each educational level group:

(c) To address your observation, we conducted a sensitivity analysis excluding education from the MSE, showing that the effects remained consistent.

Sensitivity analyses

[...] Additional models were run to rule out potential bias from single factors linked to worse clinical-cognitive phenotypes (i.e., education and SES). Excluding education from the global MSE score confirmed that the association between adverse MSE and reduced cognition remained significant (**Supplementary Information 2**). [...]

Discussion

[...] The combined effects of MSE components demonstrated stronger associations even after excluding education [...]

Supplementary Information

Supplementary Information 2. Exclusion of education from the MSE global score

To eliminate potential bias in the association between MSE and cognition caused by education in HC, we created a model excluding this score. Results showed that the association between adverse MSE and reduced cognition persisted after excluding education ($\beta = 0.304$ [95% CI = 0.241- 0.367], $p < 0.001$). Similar indices were obtained as in the model that included education: CFI = 0.895, TLI = 0.721, RMSEA = 0.124, and SRMR = 0.083.

Comment 1.4. However the variables that authors evaluated in this work are interesting, several other risk factors that may have a higher impact on cognition and functionality are missed. For example, what are the effect of hypertension, diabetes, social interactions and all the well know dementia risk factors? The MSE variable would still be able to predict the outcomes if those variables would be included in the model. I suggest, to make the analysis still more robust, the authors included all risk factors available in a sensitivity analysis.

Response 1.4. Thank you for marking this point. We clarified the boundaries of the social exposome in **response 1.2**, explaining that other risk factors (e.g., hypertension, diabetes) fall outside the scope of the social exposome construct but have been further discussed.

Discussion

[...] Factors other than MSE, beyond the social domain, have cumulative effects on brain health. Potentially modifiable risk factors for dementia may be influenced by disparities⁶. Latin America has one of the highest population-attributable fractions for dementia risk factors in the world^{24,25}, associated with high levels of adversity, inequality²⁶⁻³⁰ and dementia prevalence^{27,31}. Complex interactions between social exposome, genetics, physical exposome (pollution, pesticides, heavy metals)⁵, and modifiable risk factors are beyond the scope of this work and require additional research. [...]

Comment 1.5. Specifically in AD patients, one of the weakness is the lack of AD biomarkers to make a biological diagnosis. Considering that around 30% of individuals can be misdiagnosed solely on clinical grounds, it is a weakness that deserve mention.

Response 1.5. We appreciate this point. We have discussed this issue as follows:

Discussion

[...] Like recent works in the region^{5,32,33}, a limitation of this study is the reliance on clinical criteria for diagnosis without biomarkers like amyloid- β and tau, assessed via PET or plasma. PET is costly, globally inaccessible, and lacks validation in diverse populations such as LA³⁴. Similarly, the feasibility and accuracy of blood-based biomarkers in LA have not yet been developed to validate their use combined with clinical criteria³⁵ [...]

Comment 1.6. About the neuroimaging analysis. It is interesting that cerebellar areas are so correlated with the MSE in this sample. The authors should discuss this.

Response 1.6. Thank you for your comment, which we have further addressed:

Discussion

[...] Cerebellar involvement is consistent with its vulnerability to dementia³⁶⁻³⁹ and environmental stressors such as low-SES^{18,40-43}, structural inequality³², and poverty⁴⁴. The cerebellum is part of the allostatic interoceptive network⁴⁵, whose dysregulation has been documented in individuals with dementia⁴⁶⁻⁴⁹ and those with socioeconomic disparities^{50,51}. Thus, allostatic interoception provides a framework for understanding the dual impact of dementia pathophysiology and social adversity. [...]

Comment 1.7. About the figures. However the objective seems to make them comprehensive, they sometimes can be hard to understand and are quite polluted. I suggest simplify them. Additionally, all the abbreviations should be explained in the figures.

Response 1.7. Thank you for this observation. We have clarified the figure legends to ensure they are easier to interpret. Additionally, we have removed redundant elements to improve the overall clarity and focus of the figures. The new versions of the figures are presented here:

[editorial note: figure redacted]

Figure 1. Study design and analysis pipeline. **a**, Healthy controls (HC) and participants with Alzheimer's disease (AD) and frontotemporal lobar degeneration (FTLD) were recruited from six Latin American countries: Argentina, Brazil, Chile, Colombia, Mexico, and Peru. **b**, The assessment protocol included a comprehensive multidimensional social exposome (MSE) evaluation alongside clinical, cognitive, and neuroimaging measures. For MSE quantification, all items underwent imputation for missing values (NaN) and were then min-max scaled between 0 and 1, with higher values indicating lower levels of social adversity. Composite scores for each participant were calculated by averaging variables within specific categories: education, food insecurity, financial status, assets, access to healthcare, childhood labor, subjective socioeconomic status (SES), childhood experiences, traumatic events, and relationships. **c**, These domain-specific scores, ranging from 0 to 1, served as inputs in structural equation models (SEM) to predict cognitive function, functional ability, neuropsychiatric symptoms. **d**, The global MSE score, extracted from the latent variable, was employed to predict brain structure and functional connectivity. The coverage of regions within the top clusters identified in whole-brain voxel-based morphometry analyses using magnetic resonance imaging (MRI) was examined. Specific regions may include the middle frontal gyrus (MFG), insula (INS), rolandic operculum (ROL), and anterior cingulate cortex (ACG). In resting-state functional magnetic resonance imaging (rsfMRI) analyses, ROI-to-ROI connectivity was assessed by identifying the top regions with the highest degree (i.e., the number of connections with distinct regions). For illustration purposes, critical regions such as the cerebellum (cerebellum crus: CCRU; cerebellum lobule: CER), and middle frontal gyrus (MFG) are highlighted. All analyses were controlled for age, sex, total intracranial volume (TIV), and scanner.

Figure 3. Association between multidimensional social exposome and brain structure. **a**, Whole-brain analysis for AD shows regions with significant associations between MSE and GMV, highlighted by threshold-free cluster enhancement (TFCE) values. Circular barplot charts are included for top cluster indicating the percentage coverage of top region within the cluster. Scatterplots illustrate the relationship between MSE scores and clusters GMV. **b**, Whole-brain analysis for FTLD highlights regions significantly associated with MSE, with TFCE values similarly visualized. Corresponding scatterplots depict the associations between MSE scores and GMV for each identified cluster. ACG = Anterior cingulate gyrus; CCRU = Cerebellum crus; CER = Cerebellum lobule; FFG = Fusiform gyrus; IFGoperc = Inferior frontal gyrus, opercular part; IFGtriang = Inferior frontal gyrus, triangular part; INS = Insula; LING = Lingual gyrus; MFG = Middle frontal gyrus; ORBinf = Inferior frontal gyrus, orbital part; ORBmed = Medial orbital gyrus; PUT = Putamen; REC = Rectus gyrus; ROL = Rolandic operculum; SFG = Superior frontal gyrus; SFGdor = Superior frontal gyrus, dorsolateral; SFGmed = Medial superior frontal gyrus; VER = Vermis lobule.

Figure 4. Association between multidimensional social exposure and brain connectivity. **a**, Functional connectivity in AD displayed through ROI-to-ROI connectivity maps, where color-coded edges represent significant t-values for decreased (red) and increased (blue) connectivity associated with MSE. The connectome ring provides a simplified visualization of interconnections between key regions. The degree percentage circular barplot chart highlights top regions with the highest degree (i.e., the number of connections with distinct regions) **b**, In FTLD, ROI-to-ROI connectivity maps similarly provide visualizations of significant connections associated with MSE, with connectome ring displaying interconnections between key regions and degree percentage circular barplot chart showing key regions with the highest degree. ACG = Cingulum anterior; AMYG = Amygdala; ANG = Angular gyrus; CAL = Calcarine cortex; CAU = Caudate nucleus; CCRU = Cerebellum crus; CER = Cerebellum lobule; CUN = Cuneus; FFG = Fusiform gyrus; HIP = Hippocampus; IFGtriang = Inferior frontal gyrus, triangular part; INS = Insula; IOG = Inferior occipital gyrus; IPL = Inferior parietal lobule; LING = Lingual gyrus; MOG = Middle occipital gyrus; ORBsup = Superior frontal gyrus, orbital part; ORBsupmed = Medial superior frontal gyrus, orbital part; PAL = Pallidum; PCUN = Precuneus; PHG = Parahippocampal gyrus; PUT = Putamen; PoCG = Postcentral gyrus; PreCG = Precentral gyrus; ROL = Rolandic operculum; SFGdor = Superior frontal gyrus, dorsolateral part; SFGmed = Superior frontal gyrus, medial part; SMA = Supplementary motor area; SMG = Supramarginal gyrus; SOG = Superior occipital gyrus; STG = Superior temporal gyrus; VER = Vermis lobule.

REVIEWER #2

Comment 2.1. The authors report on the relationship between multi-dimensional social exposure and cognitive and brain function, which they term "brain health," in a multi-site cohort of healthy and demented subjects from Latin America. The authors describe a new questionnaire called the "multidimensional social exposure" (MSE) assessment which they use to evaluate social and environmental inequities for cognitively intact and demented adults. Cross-sectional analyses were grouped into two themes, 1) MSE associations with cognitive, neuropsychiatric, and functional ability measures, and 2) MSE associations with structure and functional brain measures. This study is staged to address a potential gap in our understanding of the impact of the social exposure on cognitive and brain health in an older adult sample from Latin America.

Response 2.1. Thanks for your evaluation and feedback. We addressed your comments as outlined in the responses below.

Comment 2.2. However, the manuscript has several considerable conceptual and methodological limitations. Relatedly, the description lacks sufficient detail to allow replication. While the authors focus on their own prior work, several studies have described the effects of social factors on cognitive and structural and functional brain health in aging (Avila et al., *Alzheimer & Dementia* 2021; Binnewies et al., *Brain Research Bulletin* 2023; Chan et al., *Nature Aging*, 2021; Soh et al., *JAMA Neurology* 2023; Walhovd et al., *Cerebral Cortex* 2022). It is not clear how the MSE approach builds on the limitations of prior work which have used summary measures of exposures (e.g., income, education). I agree with the value of examining multi-dimensional exposures and appreciate the focus on participants from Latin America, which follows several related papers from this group of authors. However, the current work lacks in demonstrating information over and above traditional measures such as SES. As it rests, the paper also ultimately constitutes a large listing of MSE-brain/cognition relationships. It is not clear what exactly is learned beyond what has been demonstrated in work examining social determinants of brain health disparities in older adults, and how these specific sets of relationships should be considered.

Response 2.2. Thank you for these comments. In this revised version of the manuscript, we have focused all conceptual and methodological observations and incorporated previous literature to highlight the differences with this work. We have explicitly detailed the contribution of this study in comparison to traditional measures of SES and exposures. Additionally, we conducted new analyses demonstrating that the MSE surpasses traditional measures like SES and improving the results regarding the associations with cognitive and clinical outcomes through sensitivity analyses.

(a) We have addressed several methodological concerns and clarified our approach including the MSE definition, measurement, and relation with associations and other confounding (**responses 1.2, 1.3, 2.13, and 4.14**). In the manuscript, we also summarized these methodological considerations as follows:

Results

[...] Multiple sensitivity analyses were employed to confirm that the results were not biased by individual dimensions or traditional SES factors [...]

Sensitivity analyses

[...] In summary, the robustness of our findings was supported through multiple approaches, including validation of MSE dimensions, controlling for key covariates, and assessing cross-country consistency. MSE models showed better fit and predictive power compared to SES-only models, and the effects persisted even when education was excluded from the MSE or missing values were not imputed.

(b) We have clarified the novelty and relevance of our work compared to previous studies:

Discussion

[...] Our approach considered the combined influence of traditional (e.g., educational achievement and financial status^{20,21,52}) and non-traditional factors for each dimension, such as parental education^{53,54}, rurality⁵⁵, education quality^{22,56,57}, financial stress⁹, food insecurity⁵⁸, difficulties in accessing healthcare¹³, childhood experiences¹¹, exposure to traumatic events¹², and relationships⁵⁹⁻⁶¹. Each factor has been studied individually but not combined into a composite exposome measure to assess its association with cognition, functional ability, and brain health. Thus, our approach allows us to examine the cumulative effects of multiple dimensions via the MSE. [...]

(c) We have included the limitations of the employment of the SES measure:

Introduction

[...] Although valid, SES measures present multiple limitations⁶², such as inconsistent definitions (e.g., varying metrics for education, income, and profession), lack of standardization across contexts, and fragmentation of domains precluding the assessment of cumulative burdens. [...]

(d) We now statistically compared the predictive values of MSE vs (subjective and objective) SES:

Sensitivity analyses

[...] Furthermore, models considering only objective or subjective SES showed lower fit indices and weaker associations than MSE in predicting cognition, functional ability, and neuropsychiatric symptoms (Supplementary Information 3). [...]

Discussion

[...] The stronger model fit and greater magnitude of associations for MSE compared to SES alone underscored a cumulative multidimensional burden [...]

Supplementary Information

Supplementary Information 3. Comparison of MSE and SES models

We created a latent model of objective SES using educational attainment, occupation, assets, ability to cover basic needs, disposable income, proportion of income allocated to mortgage payments, and financial resilience. For a subjective SES model, we used self-evaluations of household socioeconomic position within their community based on income, education, and occupational status⁶³. These objective and subjective SES factors were then used to predict cognition, functional ability, and neuropsychiatric symptoms.

For the entire sample, lower objective SES was linked to lower cognitive function ($\beta = 0.185$, 95% CI [0.135–0.235], $p < 0.001$) but did not predict functional ability ($\beta = -0.009$, 95% CI [-0.060–0.043], $p = 0.744$) nor neuropsychiatric symptoms ($\beta = 0.040$, 95% CI [-0.011–0.091], $p = 0.126$). Regarding subjective SES, the latent factor was associated with decreased cognitive function ($\beta = 0.093$, 95% CI [0.048–0.138], $p < 0.001$), reduced functional ability ($\beta = 0.133$, 95% CI [0.088–0.177], $p < 0.001$), and increased neuropsychiatric symptoms ($\beta = 0.097$, 95% CI [0.052–0.142], $p < 0.001$).

Relative fit indices showed that the MSE model had the best fit⁶⁴ than SES models, with AIC = -41641.99, BIC = -41379.74, and SABIC = -41525.88. The objective SES model had AIC = -20051.62, BIC = -19920.49, and SABIC = -19993.57. The subjective SES model showed SRMR = 0.026, AIC = -10629.72, BIC = -10544.20, and SABIC = -10591.86.

Comment 2.3. Distribution of healthy and diseased older adult subjects pooled across all six sites for each analysis family is not described in detail. For an effort of this size, greater details about demographics and within sample distinctions should have been included.

Response 2.3. Thank you for this observation. We have added further details about the recruitment processes, harmonization, and characterization of the clinical samples, including the number of subjects per country for each type of analysis (tables omitted in this document for brevity). We also confirmed that differences between countries across LA did not influence the results.

(a) Details on sample size description:

Participants

The study included 2,211 participants with a mean age of 64.63 years (SD = 11.26), of whom 32.97% were women [...] Participants were recruited from the Multi-Partner Consortium to Expand Dementia

Research in Latin America (ReDLat)⁶⁵, with recruitment conducted across six LA countries: Argentina (n = 112, HC:AD/FTLD = 52:60), Brazil (n = 172, HC:AD/FTLD = 116:56), Chile (n = 200, HC:AD/FTLD = 78:122), Colombia (n = 730, HC:AD/FTLD = 250:480), Mexico (n = 356, HC:AD/FTLD = 227:129), Peru (n = 641, HC:AD/FTLD = 452:189) [...]

Neuroimaging acquisition and preprocessing

MRI preprocessing

3D T1-weighted images were collected for 875 individuals, including 310 with AD, 123 with FTLD, and 442 HC (Argentina [n = 84, HC:AD/FTLD = 38:46], Brazil [n = 92, HC:AD/FTLD = 69:23], Chile [n = 137, HC:AD/FTLD = 53:84], Colombia [n = 203, HC:AD/FTLD = 25:178], Mexico [n = 53, HC:AD/FTLD = 29:24], Peru [n = 306, HC:AD/FTLD = 228:78], **Supplementary Table 7**)

Resting-state fMRI preprocessing

Resting state sequences were collected for 500 individuals, including 232 with AD, 85 with FTLD, and 183 HC (Argentina [n = 51, HC:AD/FTLD = 27:24], Brazil [n = 86, HC:AD/FTLD = 65:21], Chile [n = 127, HC:AD/FTLD = 47:80], Colombia [n = 202, HC:AD/FTLD = 24:178], Mexico [n = 34, HC:AD/FTLD = 20:14], **Supplementary Table 8**).

(b) We have clarified the recruitment approach:

Participants

[...] Participants are recruited from extensive networks and include individuals from rural and urban settings, focusing on underrepresented groups, as demonstrated previously with our ReDLat cohort marked by socioeconomic inequality³² and educational disparities³³. Strategies to improve access and recruitment for these groups involve field screenings, community engagement efforts, and the use of mobile units. [...]

(c) Following the suggestion provided in **comment 3.2**, we have also included compliance with our recruitment strategy with current guidelines on increasing representativeness of samples enrolled in Alzheimer's disease research centers⁶⁶:

Participants

[...] ReDLat aligns with current guidelines to increase the representativeness of samples enrolled in Alzheimer's disease research centers⁶⁶ by implementing inclusive recruitment strategies and strengthened community engagement, geographic decentralization, and unified inclusion and exclusion criteria [...]

Materials and methods

Questionnaire

[...] Matching guidelines to increase sample representativeness in Alzheimer's disease research⁶⁶, and recommended practices for cross-national comparisons⁶⁷, validity testing was performed with a diverse sample of Latino Spanish-speaking participants, clinicians and researchers to iteratively improve the questionnaire and standardize the collection of sociodemographic factors. [...]

(c) Following **comment 3.3** on sample representativeness, we further discussed this issue and the challenges of its implementation within the broader Latin American context:

Discussion

[...] Sample sizes are relatively modest within each country. As with other cohorts, the ReDLat⁶⁸ participants are derived from clinical samples and are not representative of the general population. However, the ReDLat cohort uses a well-distributed LA clinical sample that includes diverse populations⁶⁸⁻⁷¹. While we acknowledge that LA is not a homogenous construct, this does not preclude

the use of harmonized datasets across diverse countries^{5,32,33,61,72,73}, as has been done in other large-scale initiatives such as ADNI and European cohorts, which also feature significant sociocultural differences. Future work should expand these findings by incorporating community-representative samples to enhance the generalizability of results. Although our recruitment was aligned with current guidelines⁶⁶, many of the recommended strategies are designed with U.S. research infrastructure in mind. They may not entirely apply to the general LA context, where limited resources and larger socioeconomic disparities often hinder research.

(d) We controlled for country effects on clinical-cognitive outcomes, and the results remained consistent:

Sensitivity analyses

The associations between adverse MSE with reduced cognition and functional ability, and increased neuropsychiatric symptoms were preserved (all $p < 0.01$) after controlling for country level (Supplementary Information 1) [...]

Discussion

[...] Variations in participants' country of origin, demographic, dementia-related factors, image acquisition methods, or signal quality did not account for the results. [...]

Supplementary Information

Supplementary Information 1. Sensitivity analysis for country effects

We examined potential bias from country-specific effects to ensure that the observed associations between MSE and clinical and cognitive outcomes were not influenced by any country. We ran a new model incorporated the country level as dummy predictor of cognition, functional ability and neuropsychiatric symptoms.

For the entire sample, more adverse MSE was linked to lower cognitive function ($\beta = 0.268$, 95% CI [0.227-0.308], $p < 0.001$), functional ability ($\beta = -0.085$, 95% CI [0.045-0.125], $p < 0.001$), and neuropsychiatric symptoms ($\beta = 0.100$, 95% CI [0.059-0.142], $p < 0.001$). The model showed good fit: CFI = 0.868, TLI = 0.767, RMSEA = 0.088, SRMR = 0.061.

Comment 2.4. The justification and procedure for calculating degree centrality and subsequent analyses of degree centrality with MSE to understand the results reported on p12-13 and Figure 4 are not provided. This seems like a very opaque way of examining functional correlations given the dependence on thresholding edges.

Response 2.4. Thank you for this insightful comment. We have clarified its importance in the text to ensure transparency in our analysis approach.

Statistical analyses

Resting-state functional connectivity associations with MSE

[...] The degree of each ROI was calculated to assess its contribution to the broader connectivity patterns associated with MSE. This descriptive metric⁷⁴ represents the number of significant connections each ROI exhibits in relation to MSE. Higher values indicate that a given ROI's connectivity demonstrates more widespread associations with MSE. [...]

Comment 2.5. The ten-factor structure of MSE used in this study is not clearly described. Two sources are cited, the first source - cited as 85, Pina-Escudero et al., 2023 is an abstract reporting only three latent MSE

factors, not ten, are valid with caregiver input. The second source - citation 86, is a manual about research practices and does not appear to be specific to MSE

Response 2.5. Thank you for this observation. We have clarified the procedure and now validated the MSE dimensions using confirmatory factor analysis.

- (a) In **response 3.2** we clarified the questionnaire dimensions were developed by a team of experts.
(b) We have conducted a confirmatory factor analysis to validate the dimensions:

Scoring

[...]

Confirmatory factor analyses showed a good model fit for the dimensions (**Supplementary Table 9**). These results indicate that the dimension composite scores accurately represent the latent construct and consistently capture the intended dimensions.

Supplementary tables

Dimension	CFI	TLI	RMSEA	SRMR
Education	0.995	0.965	0.053	0.013
Food insecurity	0.988	0.907	0.107	0.017
Financial status	0.985	0.975	0.035	0.026
Assets	0.943	0.883	0.081	0.044
Access to healthcare	0.993	0.985	0.023	0.018
Childhood labor	0.998	0.974	0.073	0.017
Subjective SES	0.99	0.99	0.001	0.001
Childhood experiences	0.883	0.805	0.086	0.087
Traumatic events	0.933	0.918	0.03	0.032
Relations	0.998	0.995	0.02	0.019

Supplementary Table 9. Fit indices of the validation analysis of dimensions. CFI = Comparative Fit Index; TLI = Tucker-Lewis Index; RMSEA = Root Mean Square Error of Approximation; SRMR = Standardized Root Mean Square Residual.

- (c) Following **comment 3.6**, we also incorporated the alignment with cultural harmonization guidelines:

Questionnaire

[...] Questionnaire construction aligns with current harmonization guidelines^{75,76}, as a) items were selected after careful inspection from clinicians and researchers from different cultures (at least one from each country) and comparability across countries; b) development of psychometric procedures to test if dimension composite score represents a coherent and consistent latent construct of interrelated variables and captures the intended dimension; and c) transparency and openness by providing the full version of the questionnaire in a repository (<https://osf.io/78ng6/>).

Comment 2.6. The procedure to discard (p22) some MSE items is not described

Response 2.6. Thank you. We have extended and clarified the procedure as follows:

Materials and Methods

Validation

[...] We reviewed the questionnaire to remove items that might create circular associations with dementia-related outcomes. We excluded dimensions that could be directly influenced by dementia symptoms, thereby risking the measurement of those symptoms instead of the intended MSE construct. Specifically, we removed items related to technology access (smartphone use), employment status (work engagement), and income sources (current salary) (Supplementary Information 5).

[...]

Multidimensional social exposome (MSE) assessment

Scoring

[...] After excluding items (Supplementary Information 5), each variable was min-max scaled between 0 and 1, with 0 and 1 indicating lower and higher levels of adversity of social exposome [...]

[...]

Supplementary Information 1

Certain items and options were excluded from the item grouping for MSE dimensions (see Scoring section) to prevent circular associations with dementia brain health outcomes [...]

Comment 2.7. Steps for data harmonization between sites mentioned on p28 and separately on p32 are not clear. For example, it is stated (p28) that further modeling of site was used in the LASSO regression for structural brain measures (not SEM?). Is this because site effects remained? How much variance did site explain in your models before and after adjustment? Along these lines it is not clear that re-scaling MRI measures using a min and max within scanner value (p31-32) procedure adequately achieves the goal of harmonizing data across sites.

Response 2.7. Thank you for these careful observations. Steps for data harmonization are addressed below.

(a) First, we clarified that Lasso regression was employed to compare the magnitude of associations between the global MSE score and individual dimensions, not as a means of data harmonization:

Materials and Methods

Lasso regression

Lasso regression was employed to examine the individual contributions of MSE dimensions while controlling multicollinearity. This method was chosen for its ability to perform variable selection and regularization, shrinking less relevant coefficients toward zero and effectively excluding them from the model. The standardized beta coefficients from the SEM models linking the global MSE score to cognition, functionality, and neuropsychiatric symptoms were compared to the R^2 values from the Lasso models that included individual MSE dimensions via meta-regressions. This comparison assessed the global MSE score's explanatory power versus individual dimensions, highlighting the benefits of using an aggregate social exposure metric for understanding multidimensional outcomes.

(b) In **response 2.3** we detailed the sensitivity analysis controlling for country effects, showing consistent results.

(c) On how much variance was explained by country and site, as shown in **response 2.3**, associations were robust against country differences. We have discussed these harmonization approaches as follows:

Discussion

[...] Cross-country and cross-site effects did not influence the link between MSE and clinical, cognitive, and neuropsychiatric outcomes. [...]

(d) Here we detailed how re-scaling MRI was performed and included references of similar works that have applied this approach:

Materials and Methods

Sensitivity analyses

[...]

To investigate whether MSE was influenced by data quality of structural and functional MRI, we conducted sensitivity analyses examining the relationships between MSE and spatial signal-to-noise ratio (SNR) for both MRI and fMRI, temporal SNR for fMRI, and head motion parameters for fMRI. Scanner scaling approaches were applied to the MRI and fMRI data to reduce inter scanner variability⁵ (Supplementary table 9-10). For MRI, the minimum and maximum voxel intensity values were extracted from all subjects belonging to each scanner type, which were then scaled between the minimum and maximum intensity values specific to each scanner type. Regarding fMRI, the Fisher z-transformed Pearson correlation ROI-to-ROI connectivity matrices were scaled between the minimum and maximum correlation values specific to each scanner type. Then, voxel-based morphometry and ROI-to-ROI connectivity analyses were replicated with the scaled images and connectivity matrices. This method accounts for differences in location and scale (e.g., mean and variance) between sites, reducing scanner-related variability. This process helps to minimize variability introduced by different scanners^{5,72,77}.

(e) Following **comment 3.2** on cross-national comparisons, we have clarified the sample representation in. We have reviewed the methodological considerations and recommended best practices for cross-national comparisons:

Discussion

[...] Further, our protocol adheres to specific methodological considerations and recommended best practices for cross-national comparisons⁶⁷: (a) The MSE questionnaire was developed and harmonized by a team of experts from various countries where it was subsequently applied, validated with a diverse sample of Latino Spanish-speaking participants⁷⁸, and tested to represent the same construct across countries via psychometric validation. (b) The covariables included in SEM analyses are available and collected under a harmonized protocol across all countries. (c) We used a pooled analysis approach, incorporating country-level effects in sensitivity analysis. [...]

Comment 2.8. T1w MRI processing is not clearly described on p28. For example, authors say " a 6 x 6 x 6 mm Gaussian kernel was used to smooth grey matter segmentations" are you referring to just grey matter or some kind of grey matter atlas?

Response 2.8. We have clarified this in the text to avoid any confusion.

MRI preprocessing

CAT12 also performed intra-subject harmonization by normalizing data to the mean global intensity for each subject, followed by smoothing gray matter **images** with a 6 x 6 x 6 mm Gaussian kernel

Comment 2.9. Rest-state nuisance regression is not described in sufficient detail for replication on p28. For example, were eight or nine regressors used in a single step? Is "motion scrubbing" referring to a spike regressor or the removal of frames that meet a certain threshold after nuisance regression?

Response 2.9. We appreciate your comment. We have clarified the resting-state fMRI preprocessing pipeline. This has been clarified in the manuscript.

(a) Regressors and steps:

Resting-state fMRI preprocessing

[...] These included smoothing with a 6 x 6 x 6 mm Gaussian kernel and denoising through linear regression with nine nuisance regressors applied in a single step: six motion parameters (translation and rotation), white matter, cerebrospinal fluid signals, and scrubbing regressors for high-motion time points. [...]

(b) We have explained what we mean by motion scrubbing and how it was implemented:

Resting-state fMRI preprocessing

[...] Motion scrubbing identified volumes with framewise displacement greater than 0.5 mm or global signal change exceeding 3 standard deviations. These volumes were flagged, and corresponding spike regressors were added to the nuisance regression model to account for their influence, as defined by the CONN denoising pipeline⁷⁹. This ensured that high-motion frames were adjusted for nuisance regression without directly excluding them from the time series. [...]

Comment 2.11. Why was the AAL atlas chosen for functional connectivity analysis (p28)? AAL do not correspond to functional areas and violates known biological principles of functional organization. This makes the resting-state analyses uninterpretable.

Response 2.11. Thank you for this observation and the opportunity to clarify our rationale for atlas selection as detailed below:

Resting-state fMRI preprocessing

[...] The AAL atlas has been widely utilized in brain network research on AD and FTL, where it has been applied in structural and functional imaging studies^{5,32,33,47,72}. As no single parcellation method consistently outperforms others across metrics⁸⁰, we selected the AAL atlas to ensure comparability with existing literature. This choice also allowed us to use the same atlas for structural and functional imaging analyses, aligning our study with previous works in dementia^{5,32,33,47,72}. [...]

Comment 2.13. The imputation procedure was not well described on p30-31:

a. Was it within disease state, within site, or across sample?

b. How was 20% imputation, above the 5-10% recommended threshold, justified (Jakobsen et al., BMC Medical Research Methodology 2017)?

Response 2.13. Thank you for highlighting this point. To address concerns regarding the imputation procedure, we conducted an additional analysis without imputing missing data, and the results remained consistent. This information has been added to the manuscript.

(a) First, we clarified the extension of the imputation procedure:

Structural equation model

[...] Missing data (maximum 19.7%), were imputed across the sample [...]

(b) Then we provided a new analysis without imputation of missing values:

Structural equation model

[...] Additional sensitivity analyses were performed to check for bias due to imputation. The main SEM models were rerun without imputation for the entire sample and each group separately. Results showed consistent associations across all models, indicating that imputation did not introduce bias. (Supplementary Table 12).

Supplementary Tables

[...]

Association	β	P value	Lower 5% CI	Upper 95% CI
All subjects model (CFI = 0.945, TLI = 0.905, RMSEA = 0.069, SRMR = 0.045)				
MSE → Cognition	0.239	< 0.001	0.197	0.282
MSE → Functional ability	0.056	0.012	0.012	0.101
MSE → Neuropsychiatric symptoms	0.065	0.004	0.021	0.109
HC model (CFI = 0.874, TLI = 0.781, RMSEA = 0.095, SRMR = 0.081)				
MSE → Cognition	0.564	< 0.001	0.518	0.61
MSE → Functional ability	0.029	0.316	-0.028	0.087
MSE → Neuropsychiatric symptoms	-0.045	0.121	-0.103	0.012
AD model (CFI = 0.964, TLI = 0.937, RMSEA = 0.051, SRMR = 0.041)				
MSE → Cognition	0.216	< 0.001	0.149	0.282
MSE → Functional ability	0.069	0.048	0.001	0.137
MSE → Neuropsychiatric symptoms	0.069	0.047	0.001	0.137
FTLD model (CFI = 0.981, TLI = 0.968, RMSEA = 0.038, SRMR = 0.043)				
MSE → Cognition	0.149	0.025	0.019	0.279
MSE → Functional ability	0.152	0.022	0.021	0.282
MSE → Neuropsychiatric symptoms	0.153	0.021	0.023	0.283

Supplementary Table 12. Models results without applying imputation strategies for missing values

Comment 2.14. Multiple comparison correction for resting state MRI functional connectivity is described with threshold free cluster estimated (TFCE) voxel-based correction p32. However, TFCE relies on spatial information not found in a functional connectivity matrix.

Response 2.14. We are thankful for your perceptive observations. We have clarified the analyses as follows:

Resting-state functional connectivity associations with MSE

[...] In functional connectivity analysis, TFCE is employed to identify clusters of statistically similar ROI-to-ROI connections without arbitrary thresholds⁸¹. These clusters are based on anatomical proximity and functional similarity, using T-values from the general linear model with MSE scores predicting whole-brain ROI-to-ROI connectivity⁸¹ [...]

Comment 2.15. Many of the tables and figures lack clarity in their representation of methodology, analysis, and results. For example:

a. Table 2 results cover a SEM path analysis and LASSO regression for "Global Score" and "individual predictors." Yet, there is neither a table legend or explicit in-text description of what "Global Score" and "individual predictors" are. Moreover, it is not clear why "individual predictors" are a single value in Table 2 when p9-10 describe results for different individual predictors that differ between analysis groups (i.e., all subjects, healthy, and demented).

Response 2.15. We appreciate your observation. We have revised all tables and figures to enhance clarity, ensure consistency in reporting, and align with the terminology and structure used throughout the manuscript (see **response 1.7**). These updates include standard labels, refining figure legends to improve interpretability, and ensuring that the presentation of data matches the descriptions provided in the main text. Specifically, we have revised Table 2 to improve clarity and ensure consistency in reporting. The table now explicitly refers to the global MSE score and its individual dimensions, aligning with measures described in the manuscript.

Tables

[...]

Variable	Model type	Common effects size	Models difference	
			χ^2	p-value
All subjects				
Cognition	Global MSE score (SEM)	0.23 (0.23 - 0.24)	3498.68	< 0.001
	Individual MSE dimensions (LASSO)	0.12 (0.12 - 0.12)		
Functional ability	Global MSE score (SEM)	0.06 (0.06 - 0.06)	130.98	< 0.01
	Individual MSE dimensions (LASSO)	0.05 (0.05 - 0.05)		
Neuropsychiatric symp.	Global MSE score (SEM)	0.08 (0.08 - 0.08)	803.71	< 0.01
	Individual MSE dimensions (LASSO)	0.03 (0.03 - 0.03)		
HC				
Cognition	Global MSE score (SEM)	0.60 (0.60 - 0.60)	7133.11	< 0.001
	Individual MSE dimensions (LASSO)	0.43 (0.43 - 0.44)		
AD				
Cognition	Global MSE score (SEM)	0.23 (0.23 - 0.24)	3078.06	< 0.001
	Individual MSE dimensions (LASSO)	0.08 (0.08 - 0.08)		
Functional ability	Global MSE score (SEM)	0.08 (0.08 - 0.09)	369.65	< 0.01
	Individual MSE dimensions (LASSO)	0.04 (0.03 - 0.04)		
Neuropsychiatric symp.	Global MSE score (SEM)	0.11 (0.10 - 0.11)	502.61	< 0.01
	Individual MSE dimensions (LASSO)	0.05 (0.04 - 0.05)		
FTLD				
Cognition	Global MSE score (SEM)	0.17 (0.16 - 0.18)	375.32	< 0.01
	Individual MSE dimensions (LASSO)	0.08 (0.07 - 0.08)		
Functional ability	Global MSE score (SEM)	0.15 (0.14 - 0.16)	174.28	< 0.01
	Individual MSE dimensions (LASSO)	0.08 (0.08 - 0.09)		
Neuropsychiatric symp.	Global MSE score (SEM)	0.15 (0.15 - 0.16)	144.27	< 0.01
	Individual MSE dimensions (LASSO)	0.10 (0.09 - 0.10)		

Table 2. Meta-regression comparison between global MSE score and the effect of individual MSE dimensions.

Comment 2.16. Figure 3 does not provide x and y-axis labels for regression plots so authors must assume x-axis is some individual brain structure's volume regressed overall MSE composite. However, a toy representative plot in the bottom of Figure 3 may be a legend for regression plots except they say "MSDH" on y-axis relating to Cluster GMV on x-axis.

Response 2.16. Thank you for noting this typographical error. We have corrected this figure as shown in response 1.7.

REVIEWER #3

Comment 3.1. This study investigates the influence of the social exposome – an aggregate of exposures/experiences across the lifespan, like nutrition, education, finances, and healthcare access – on cognition and brain health among healthy controls and people diagnosed with Alzheimer's dementia or frontotemporal lobar degeneration residing in various countries in Latin America. While we commend the authors for these large cross-national efforts and the richness of the data collected (i.e., social factors, diagnostic, neuroimaging), there are significant limitations in their approach that obscure any potential meaningful contribution this study can offer to improving our understanding of global cognitive and brain health.

Response 3.1. Thank you for reviewing our manuscript. Your suggestions have helped us refine our work. We have provided detailed responses to each point below.

Comment 3.2. First, the authors group various diverse Latin American countries together to represent Latin America as a whole, when Latin America is far from a homogenous construct. How do the findings reported in the manuscript provide any additional help to further our understanding of cognitive health among these Latin American countries? All countries listed here have very diverse, historical and current, sociopolitical and sociocultural contexts that likely would influence unique and shared pathways that impact late-life cognitive health. A more appropriate and informative approach would be to leverage this large cohort for cross-national comparisons. For instance, are the results reported here the same across each of the countries? For instance, educational and healthcare policies are different between these countries, would that not suggest differences in the strength of the associations reported? Would that not provide more valuable information to the aging communities in the individual Latin American countries and their respective policy makers to effect change for the benefit of their communities? I recommend the current article in guidance on best approaches for cross-national analyses <https://doi.org/10.1002/alz.13694>

Response 3.2. We are thankful for your comprehensive evaluation. We have further explained LA's heterogeneity by highlighting its sociocultural diversity. Also, we have explained the alignment of our protocol with best practices for cross-national comparisons (**responses 2.7, 3.4**), including harmonization and sensitivity analyses (**responses 2.3, 2.7**) and controlled for country effects (**response 2.3**).

(a) In addition, we addressed the point regarding LA heterogeneity:

Introduction

[...] Heterogeneous and distributed factors associated with social and health disparities impact cognition and functional ability across LA⁶¹. These factors are linked to historical, sociopolitical, and sociocultural diversity²⁷, as well as political polarization and its potential impact on health⁸². [...]

(b) Novelty of our work regarding brain health: We have clarified the novelty of our approach in **responses 1.2, 1.3, and 2.2**. Also, new analyses shown that MSE association with clinical and cognitive outcomes are robust against country differences (**response 2.3**) and are stronger than traditional measures (SES) (**response 2.2**).

(c) Methodological considerations and recommended best practices for cross-national comparisons has been addressed in **response 2.7**

Comment 3.3. Second, sample sizes are relatively modest within each country, yet they are described as if they represent the given country and thus represents Latin America. What efforts have been made to demonstrate that these samples are representative of their country? If they are not meant to be representative, that is fine, but then interpretation and messaging of findings should be modified slightly as to make sure to not indicate that these findings represent Latin America as their introduction and discussion states. I recommend the following article regarding issues related to representativeness <https://doi.org/10.1002/dad2.12450>.

Response 3.3. We appreciate your comments and the opportunity to clarify this point. As detailed here and in other ReDLat studies (e.g.^{5,32}), our samples are clinical and are not intended to be representative of the general population. We have revised the text throughout the manuscript to avoid any misinterpretation (**response 2.3**). Additionally, in the same response we have assessed the adherence to best approaches for representativeness regarding ReDLat recruitments.

Comment 3.4. Third, to further enhance the contribution and differences between each country, information regarding recruitment approaches by country should be described and provide information whether the distribution of the MSE scores were similar across country or not. For instance, authors describe that they ran sensitivity analyses to address recruitment bias, yet they do not define what is meant by recruitment bias and

no such analysis is reported. Similarly, it would be helpful to know whether participants are recruited from one particular city in each country or multiple cities? Primarily urban settings or rural settings as well? What are the sociodemographic characteristics of the different cohorts? Along those same lines, it would be informative to know about differences in the MSE characteristics by country.

Response 3.4. We appreciate your detailed comments and have addressed each point as detailed below.

(a) Recruitment by country: In **response 2.3**, we have further detailed the recruitment, including an expanded description on how participants are recruited from both rural and urban settings, and included additional information on the distribution of participants per country.

(b) Recruitment bias:

Discussion

[...] Recruitment bias—defined as factors that may influence the distribution of MSE scores during participant recruitment—was controlled for in our analyses. These controls include: (a) accounting for variations in disease severity, age at diagnosis, and time elapsed since diagnosis to mitigate potential overrepresentation of participants from disadvantaged groups in more advanced disease stages; (b) eliminating circularity in the assessment of MSE by excluding variables that may confound socioeconomic disparities with dementia outcomes (e.g., access to specific services such as cell phones could indicate either socioeconomic disparities or the consequences of cognitive decline); and (c) conducting within-group analyses to identify MSE associations through a dimensional approach rather than presuming uniform patterns across groups. [...]

(c) Control for country-level effects: In **response 2.3** we detail how country-level data does not impact the general results.

Comment 3.5. Fourth, the construct of the social exposome is not clearly defined nor supported. In the introduction, the authors do not define the social exposome. It would be helpful to provide support for why this composite of various social factors is more informative than evaluating each factor jointly? To gain better understanding on how and where to intervene, we need to understand the contribution of individual social factors on health. In fact, the authors do provide in their results information on which MSE indicators are driving their findings, then, why use the MSE composite score at all? For instance, a top indicator was education, but education comprised of several different aspects from years of schooling, self-reported quality of education, parental years of education, so what about education might be driving this result? How can one intervene in terms of education then? And would the intervention be the same across all of Latin America?

Response 3.5. We value your observations. We have now included a detailed definition of the social exposome in **response 1.2**, and explained how accumulated adversity and disparity may impact more significantly than isolated factors.

(a) Relevance of the composite score of various social exposome dimensions has also been assessed in **response 2.2**. In addition, we have included an additional clarification:

Discussion

[...] Results demonstrate that the MSE, as a combined measure, exhibits stronger effects than individual factors and composite SES measures, providing a low-dimensional representation of the cumulative social exposome. The findings highlight how diverse data and tailored modeling can capture precise brain health outcomes of aging and dementia, crucial for efficient prevention and multicomponent interventions addressing both individual- and societal-level determinants⁸³.

Comment 3.6. Fifth, the development and validity of the MSE is unclear. The authors reported that they created composite factors, but that wording suggests they were derived through a factor analysis when they were not.

These composite scores were derived through consensus agreement and summing of each item. Why not employ psychometric techniques to determine the structure of the proposed measure such as using EFA and CFA techniques to determine that the theoretically proposed groupings do hang together psychometrically? And are these factors measuring truly the same thing between countries? Greater information regarding their statistical harmonization approaches is warranted. I recommend the following readings as helpful guidelines <https://doi.org/10.1037/neu0000816> and [10.1016/S2666-7568\(23\)00170-8](https://doi.org/10.1016/S2666-7568(23)00170-8)

Response 3.6. Thank you for this valuable comment. We have now integrated these harmonization guidelines^{75,76} in the text and highlighted their alignment with our approach. Furthermore, in **response 2.5** we conducted a confirmatory analysis to validate the dimensions of the MSE.

(a) Wording regarding the calculation of composite scores: We use now "dimension scores" throughout the text for clarity and consistency:

Results

The MSE was operationalized by grouping items measuring facets of similar dimensions based on expert consensus agreement. Then, we calculated the weighted averages on these items to create dimension scores for education, food insecurity, financial status, assets, healthcare access, childhood labor, subjective SES, childhood experiences, traumatic events, and relationships. These scores were obtained by scaling between the minimum and maximum values for a total of 319 items and then averaging these to obtain a value for each dimension (Figure 1B). We validated these dimensions via confirmatory analysis. [...]

(b) Validation and addressing of suggested references and alignment with cultural harmonization guidelines: Detailed in **response 2.5**.

Comment 3.7. In Figure 1 for their Outcomes predictions figure shouldn't the labels be flipped and MSE factor score be on the X axis? It is my understanding that they are using MSE to predict the outcomes of cognitive status, functional ability, and neuropsychiatric symptoms, right? If not, then please clarify.

Response 3.7. Thank you for this observation. We clarified the figure to show that MSE predicts cognitive status, functional ability, and neuropsychiatric symptoms. The changes are shown in **response 1.7**.

Comment 3.8. Across their assessments of lifecourse experiences there is a huge age gap between age 10 and 35. Unclear why they do not assess for experiences during that large and formative age period.

Response 3.8. Thank you for this comment. We have incorporated text to acknowledge this limitations and further research:

Discussion

[...] While our MSE assessment considers different life stages, items were limited to early, middle, and late life (0 to 10, 35 to 45 years old, and recently), leaving out other age periods. This approach was taken to ensure a more detailed assessment than traditional SES approaches while avoiding excessive items that could increase participant burden and compromise data quality. However, future studies should incorporate additional life stages to enable a more comprehensive analysis across the lifespan [...]

Comment 3.9. There are several sentences that require additional information to help clarify what was being meant: Authors state in their introduction that LA is one of the most unequal regions in the world, but unequal in what way? And are those inequalities the same across the various countries in Latin America? These are very large overgeneralizing sweeping comments that obscure the heterogeneity within Latin America.

Response 3.9. Thank you for noting this point. We have extensively revised the introduction and other parts of the manuscript (**responses 1.7, 2.15, 3.7, 4.15**). We also clarified the aspect of inequality as follows:

Introduction

[...]

The region has the second-highest estimated global prevalence of dementia³¹ and is one of the most unequal regions in the world, with significant disparities in both material and non-material social resources. These include financial assets, healthcare access, education, social support, and infrastructure²⁷. Such inequality is reflected by Gini scores frequently surpassing 0.47, indicating considerable wealth inequality^{32,84}.

Comment 3.10. “Other reports suggest stronger effects of SDH than ancestry for dementia risk in LA” authors should specify that this study measured genetic ancestry.

Response 3.10. We have corrected the text (“genetic ancestry”).

Comment 3.11. Sentence: “Thus, incorporating the multiple dimensions of social exposome and assessing their potential impact on brain health outcomes of aging and dementia in underrepresented populations is critically needed to develop robust tailored models.” Tailored models of what?

Response 3.11. We have clarified this as follows:

Introduction

[...] There is a need to explore the various aspects of the social exposome and their impact on brain health outcomes in aging and dementia for underrepresented populations. This may help to develop tailored biopsychosocial models of dementia.

REVIEWER #4

Comment 4.1. In this multi-centre study Migeot and coauthors collect in six Latin American countries a rich set of “social exposome” variables using a fit-for-purpose questionnaire alongside demographic, clinical and brain imaging data to address the question about their association with two types of dementia – Alzheimer’s disease (AD) and frontotemporal lobar degeneration (FTLD). Using structural equation modeling, the obtained “multidimensional social exposome (MSE)” construct showed association with poorer cognition in healthy aging, mainly modulated by variables related to different measures of education and SES, while in AD and FTLD it also demonstrated associations with functional abilities and neuropsychiatric symptoms. The brain anatomical and functional associations were confined to frontal and cerebellar regions in AD, and fronto-temporo-limbic and cerebellar regions in FTLD.

Response 4.1. Thanks for your evaluation and the opportunity to improve our manuscript based on your review. We have addressed all your comments in the following responses.

Comment 4.2. This is a potentially interesting study, shedding light on parts of the world that are clearly underrepresented in current epidemiological research. In my view, a “head-to-head” comparison with similar open access data (e.g. UK Biobank, ADNI) would help convincing the readership on some of the critical aspects outlined below. The study strongly relies on subjective data with a major part of events dating back many decades before the current assessment. Given that the focus is on dementia – Alzheimer’s and FTLD, I remain skeptical about the validity of the obtained information given a strong disease-specific bias. Similarly, the diagnosis of dementia is denoted with “probable”. The absence of fluid biomarkers results, together with the missing information about the presence of small vessel disease and the marked age differences between cohorts raise questions about the validity of the obtained results.

Response 4.2. Thank you for these insightful comments. Below we have discussed challenges in harmonizing our protocol with open datasets, addressed reliance on subjective data as a field-wide limitation, clarified our use of clinical criteria for dementia diagnosis, and demonstrated robustness of results despite age and sex differences through sensitivity analyses.

(a) “Head-to-head” comparison with similar open access data:

Discussion

[...] Our study addresses underrepresented populations from LA which present unique patterns of brain health^{5,61} in comparison with more homogenous populations (e.g. UK Biobank). The assessments of the dimensions related to social exposome are designed with different objectives and utilize distinct methodologies⁶². Future studies should develop more global and inclusive participation and assessment across the globe. [...]

(b) Reliance on subjective data:

Discussion

[...] While our MSE assessment considers different life stages, it relies on participant self-report, which could introduce bias. This is a limitation of the broader field of dementia assessment through self-report methods⁸⁵. We partially addressed this issue by omitting items from the assessment that can be biased by dementia symptomatology (e.g., smartphone use, occupational situation, **Supplementary Information 5**), and corroborating participant self-reports with information provided by caregivers. This is consistent with standard practices for questionnaire-based assessments in individuals with dementia, such as evaluating depressive symptoms⁸⁶, quality of life⁸⁷, and activities of daily living⁸⁸. Additionally, we controlled the impact of disease severity on these predictors to minimize confounding effects. [...]

(c) Dementia diagnosis clinical criteria: Addressed in **response 1.5**.

(d) Age differences: we have clarified our approach to control for these differences as follows:

Sensitivity analyses

[...] Despite an unbalanced distribution of age and sex across groups (Table 1), all effects remained consistent after controlling for these covariables [...]

[...]

[...] The observed effects in whole-brain gray matter volume and ROI-to-ROI functional connectivity analyses were robust to imbalances in age and sex distribution across groups. Consistent results were observed while controlling for these covariates in the regression models (Tables 3-6). [...]

Comment 4.3. There is a significant age difference between the cohorts of healthy controls, FTLD and AD, which mounts up to a decade.

Response 4.3. Age differences between cohorts were accounted for in sensitivity analyses (**response 4.2**), and here we provide further considerations on age and sex effects for within-group models:

Materials and methods

Structural equation model

[...]

[...] In addition to testing group-specific effects, this approach allows us to isolate the effects within each group. This approach abolishes the confounding factors observed in group comparisons due to age and sex differences. [...]

Comment 4.4. The expected higher ratio of women in the AD cohort established in epidemiological studies on AD does not substantiate here. Could you please expand on this?

Response 4.4. Thank you for this observation and apologize for the oversight. This was a typographical error where the reported ratio appeared as M:F instead of F:M. The corrected ratio aligns with the established epidemiological pattern of a higher proportion of women in the AD cohort. This error has been fixed in the table, as noted in **response 2.3**.

Comment 4.5. In patients with dementia, one would expect difficulties correctly recalling items from the distant and near past. Similarly, individuals with FTLN would tend to differentially recall traumatic events from the past given predominant emotional regulation deficits compared to the other two cohorts. In my view, this represents a significant bias. What measures were taken to mitigate these effects?

Response 4.5. Thank you for your detailed examination. As noted above, we mitigated these biases by excluding questions prone to recruitment bias and ensuring consistency in responses across caregivers and patients (see **response 2.6**). These steps were designed to reduce recall variability. Additionally, we have explicitly discussed these limitations in the manuscript (see **response 4.2**).

Comment 4.6. Could the authors provide a matrix with correlations between the individual variables of the social exposome. Given that the observed effects of nutrition, financial status, subjective SES, and access to healthcare are tightly linked to socio-economic status, I wonder about the amount of shared variance between them.

Response 4.6. We appreciate your meaningful suggestion. We have now demonstrated that SES, whether objective or subjective, is less effective in explaining the observed effects compared to MSE (**response 2.2**).

In response to your request, we have provided a correlation matrix. The analysis shows moderate correlations among financial status, assets, and education—variables traditionally used to measure SES—while weaker correlations were observed between these and the remaining variables. This suggests, consistent with our new SES results reported in **response 2.2**, that the SES and non SES components of the social exposome display distinct patterns, with some shared but non-uniform variance.

Comment 4.7. “To enhance behavioral variance and statistical power, we analyzed 720 each patient group (AD and FTLN) in tandem with HCs.” Could you please clarify the design matrix for VBM analysis? Did it include

all 875 individuals categorized in three different groups according to their disease label or did you create two identical designs with healthy controls and either AD or FTLD?

Response 4.7. Thanks for your evaluation. We have clarified the design matrix for VBM.

Grey matter associations with MSE

[...] To enhance behavioral variance and statistical power^{89,90}, HCs were merged with AD patients to run VBM models for the AD group, and the process was repeated by pairing HCs with FTLD patients for the FTLD model [...]

Comment 4.8. The brain anatomical patterns associated with MSE in both clinical cohorts are unexpected. If MSE in both AD and FTLD is closely associated with cognitive function, functional ability, and neuropsychiatric symptoms, what might be the explanation for the differential anatomical pattern?

Response 4.8. Thank you for this question. We have expanded on this idea in the Discussion section, highlighting how the cumulative influence of the social exposome and allostatic overload may be associated with brain burden and accelerate pathophysiological processes in dementia. Additionally, as noted in **response 1.6**, we expanded on novel models that might explain how stress and adversity affect brain regions not typically associated with dementia and the distinctive pattern between AD and FTLD. These points have been clarified in the manuscript.

Discussion

[...] Environmental stressors can impact distributed brain regions while accelerating specific pathophysiological processes related to dementia^{5,32,33,47}. In AD, progressive amyloid and tau pathology affect frontal and parietal structures, potentially amplifying the effects of MSE disrupting executive and integrative processes related to cognitive, functional, and neuropsychiatric impairments⁹¹⁻⁹³. In FTLD, early degeneration targets fronto-temporo-limbic networks, leading to connectivity disruptions underlying socioemotional dysregulation, behavioral changes, and similar clinical outcomes⁹⁴. MSE may primarily exacerbate AD structural vulnerability through cumulative stress and resource depletion, whereas in FTLD, they predominantly could aggravate functional connectivity dysregulation. These neurobiological pathways may suggest that MSE can be linked with partially comparable clinical outcomes despite differing structural and functional correlates. [...]

Comment 4.9. What is the rationale behind the use of TFCE correction in a well-powered cohort of more than 870 individuals?

Response 4.9. Correction procedures are relevant regardless of sample size, to ensure the robustness of findings⁷⁹. Without corrections, there is a risk of overinflating the results. In **response 2.14** we provided a more detailed explanation of the TFCE correction method.

Comment 4.10. Were the results of the sensitivity analyses also corrected for multiple comparisons? If yes, please state at which threshold and if FWE, FDR or other method.

Response 4.10. Thanks for the opportunity to clarify this point. The results of the sensitivity analyses were corrected for multiple comparisons using the TFCE method, with a pFDR < 0.05 threshold:

Sensitivity analyses

[...] All analyses were controlled by age, sex, and total intracranial volume (for voxel-based morphometry analyses). Multiple comparisons were corrected with the threshold-free cluster (TFCE) method with a pFDR < 0.05

Comment 4.11. How was “disease severity” (line 340) measured? If based on MMSE for AD, what was the metric used in patients with FTLD?

Response 4.11. Thank you. We have clarified the use of a severity scale (CDR) in the manuscript.

Sensitivity analysis

To control for disease-related variables, we ran a model with AD and FTLD participants that included disease severity (Clinical Dementia Rating (CDR) total score⁹⁵ and CDR-FTLD scale-modified⁹⁶) [...]

Comment 4.12. Could you please elaborate how exactly «scanner type» was included as a covariate of no interest in the brain imaging analyses?

Response 4.12. Thanks for your question. This has been clarified in the new version of the manuscript as follows:

Sensitivity analyses

To control the effects of different scanners, scanner type was included as a dummy covariate of no interest among the predictors in whole-brain gray matter volume and ROI-to-ROI functional connectivity analyses³² [...]

Comment 4.13. What criteria were applied for Quality Assessment of structural and functional MRI data?

Response 4.13. Thank you for the opportunity to clarify this point. Now we have clarified the quality assessment measures as follows:

Materials and methods

Sensitivity analyses

[...] To evaluate the quality of our MRI and fMRI data, we employed the ODQ metric⁹⁷⁻⁹⁹. For MRI data, the assessment focused on the signal-to-noise ratio (SNR) across each slice. The SNR was determined by calculating the mean signal of brain voxels and dividing it by the standard deviation of the signal^{100,101}. For fMRI data, the quality evaluation involved segmenting each time series into 20 repetition times (TR) intervals to compute the temporal signal-to-noise ratio (tSNR)¹⁰². The tSNR was defined as the mean fMRI signal within each segment divided by its standard deviation. [...]

Comment 4.14. I wonder about the added value of subjective SES compared to the “objective” SES. Could you please elaborate on this?

Response 4.14. Thanks for this insightful question. We have addressed this point in **response 2.2**, where we discuss the limitations of traditional SES measures, and the complementary insights provided by subjective SES. We have also addressed the importance of MSE with statistical comparisons with both subjective and subjective SES (**response 2.2**)

Comment 4.15. The manuscript needs serious language editing. There are also numerous missed prepositions, tautologies and orthographic errors in both text and figures

Response 4.15. The new version of the manuscript has been re-reviewed by our native speakers' co-authors.

References

- 1 Ibanez, A. *et al.* Neuroecological links of the exposome and One Health. *Neuron* **112**, 1905-1910, doi:<https://doi.org/10.1016/j.neuron.2024.04.016> (2024).
- 2 Vermeulen, R., Schymanski, E. L., Barabási, A. L. & Miller, G. W. The exposome and health: Where chemistry meets biology. *Science* **367**, 392-396, doi:10.1126/science.aay3164 (2020).
- 3 Mendenhall, E., Kohrt, B. A., Logie, C. H. & Tsai, A. C. Syndemics and clinical science. *Nature Medicine* **28**, 1359-1362, doi:10.1038/s41591-022-01888-y (2022).
- 4 Vineis, P. & Barouki, R. The exposome as the science of social-to-biological transitions. *Environment International* **165**, 107312, doi:<https://doi.org/10.1016/j.envint.2022.107312> (2022).
- 5 Moguilner, S. *et al.* Brain clocks capture diversity and disparities in aging and dementia across geographically diverse populations. *Nat Med*, doi:10.1038/s41591-024-03209-x (2024).
- 6 Livingston, G. *et al.* Dementia prevention, intervention, and care: 2024 report of the Lancet standing Commission. *The Lancet*, doi:10.1016/S0140-6736(24)01296-0 (2024).
- 7 Sakowski, S. A., Koubek, E. J., Chen, K. S., Goutman, S. A. & Feldman, E. L. Role of the Exposome in Neurodegenerative Disease: Recent Insights and Future Directions. *Annals of Neurology* **95**, 635-652, doi:<https://doi.org/10.1002/ana.26897> (2024).
- 8 Kirolos, A. *et al.* Neurodevelopmental, cognitive, behavioural and mental health impairments following childhood malnutrition: a systematic review. *BMJ Global Health* **7**, e009330, doi:10.1136/bmjgh-2022-009330 (2022).
- 9 Mani, A., Mullainathan, S., Shafir, E. & Zhao, J. Poverty Impedes Cognitive Function. *Science* **341**, 976-980, doi:10.1126/science.1238041 (2013).
- 10 Lupien, S. J., McEwen, B. S., Gunnar, M. R. & Heim, C. Effects of stress throughout the lifespan on the brain, behaviour and cognition. *Nature Reviews Neuroscience* **10**, 434-445, doi:10.1038/nrn2639 (2009).
- 11 Bhutta, Z. A., Bhavnani, S., Betancourt, T. S., Tomlinson, M. & Patel, V. Adverse childhood experiences and lifelong health. *Nat Med* **29**, 1639-1648, doi:10.1038/s41591-023-02426-0 (2023).
- 12 Severs, E. *et al.* Traumatic life events and risk for dementia: a systematic review and meta-analysis. *BMC geriatrics* **23**, 587, doi:10.1186/s12877-023-04287-1 (2023).
- 13 Mullins, M. A., Bynum, J. P. W., Judd, S. E. & Clarke, P. J. Access to primary care and cognitive impairment: results from a national community study of aging Americans. *BMC geriatrics* **21**, 580, doi:10.1186/s12877-021-02545-8 (2021).
- 14 Landeiro, F. *et al.* The economic burden of cancer, coronary heart disease, dementia, and stroke in England in 2018, with projection to 2050: an evaluation of two cohort studies. *The Lancet Healthy Longevity* **5**, e514-e523, doi:[https://doi.org/10.1016/S2666-7568\(24\)00108-9](https://doi.org/10.1016/S2666-7568(24)00108-9) (2024).
- 15 Gebhard, D., Lang, L., Maier, M. J. & Dichter, M. N. Social interaction of people living with dementia in residential long-term care: an ecological momentary assessment study. *BMC Health Services Research* **24**, 1640, doi:10.1186/s12913-024-12056-y (2024).
- 16 Ibanez, A. *et al.* Neuroecological links of the exposome and One Health. *Neuron*, doi:10.1016/j.neuron.2024.04.016 (2024).
- 17 Farah, M. J. The Neuroscience of Socioeconomic Status: Correlates, Causes, and Consequences. *Neuron* **96**, 56-71, doi:10.1016/j.neuron.2017.08.034 (2017).
- 18 Thanaraju, A. *et al.* Structural and functional brain correlates of socioeconomic status across the life span: A systematic review. *Neuroscience & Biobehavioral Reviews* **162**, 105716, doi:<https://doi.org/10.1016/j.neubiorev.2024.105716> (2024).
- 19 Hatzenbuehler, M. L., McLaughlin, K. A., Weissman, D. G. & Cikara, M. A research agenda for understanding how social inequality is linked to brain structure and function. *Nature Human Behaviour* **8**, 20-31, doi:10.1038/s41562-023-01774-8 (2024).
- 20 Avila, J. F. *et al.* Education differentially contributes to cognitive reserve across racial/ethnic groups. *Alzheimers Dement* **17**, 70-80, doi:10.1002/alz.12176 (2021).
- 21 Chan, M. Y. *et al.* Long-term prognosis and educational determinants of brain network decline in older adult individuals. *Nature Aging* **1**, 1053-1067, doi:10.1038/s43587-021-00125-4 (2021).
- 22 Soh, Y. *et al.* State-Level Indicators of Childhood Educational Quality and Incident Dementia in Older Black and White Adults. *JAMA Neurology* **80**, 352-359, doi:10.1001/jamaneurol.2022.5337 (2023).

- 23 Kochhann, R., Varela, J. S., Lisboa, C. S. M. & Chaves, M. L. F. The Mini Mental State Examination: Review of cutoff points adjusted for schooling in a large Southern Brazilian sample. *Dement Neuropsychol* **4**, 35-41, doi:10.1590/s1980-57642010dn40100006 (2010).
- 24 Mukadam, N., Sommerlad, A., Huntley, J. & Livingston, G. Population attributable fractions for risk factors for dementia in low-income and middle-income countries: an analysis using cross-sectional survey data. *The Lancet Global Health* **7**, e596-e603, doi:[https://doi.org/10.1016/S2214-109X\(19\)30074-9](https://doi.org/10.1016/S2214-109X(19)30074-9) (2019).
- 25 Paradela, R. S. *et al.* Population attributable fractions for risk factors for dementia in seven Latin American countries: an analysis using cross-sectional survey data. *The Lancet Global Health* **12**, e1600-e1610, doi:[https://doi.org/10.1016/S2214-109X\(24\)00275-4](https://doi.org/10.1016/S2214-109X(24)00275-4) (2024).
- 26 Miranda, J. J. *et al.* Understanding the rise of cardiometabolic diseases in low- and middle-income countries. *Nat Med* **25**, 1667-1679, doi:10.1038/s41591-019-0644-7 (2019).
- 27 United Nations Development Programme. in *Trapped: High Inequality and Low Growth in Latin America and the Caribbean* (2021).
- 28 Busso, M. & Messina, J. The inequality crisis: Latin America and the Caribbean at the Crossroads. *Inter-American Development Bank* **32**, 0002629 (2020).
- 29 Migeot, J. *et al.* Allostasis, health, and development in Latin America. *Neuroscience & Biobehavioral Reviews* **162**, 105697, doi:<https://doi.org/10.1016/j.neubiorev.2024.105697> (2024).
- 30 Borelli, W. V. *et al.* Race-related population attributable fraction of preventable risk factors of dementia: A Latino population-based study. *Alzheimer's & dementia (Amsterdam, Netherlands)* **15**, e12408, doi:10.1002/dad2.12408 (2023).
- 31 Parra, M. A. *et al.* Dementia in Latin America: Assessing the present and envisioning the future. *Neurology* **90**, 222-231, doi:10.1212/WNL.0000000000004897 (2018).
- 32 Legaz, A. *et al.* Structural inequality linked to brain volume and network dynamics in aging and dementia across the Americas. *Nature Aging*, doi:10.1038/s43587-024-00781-2 (2024).
- 33 Gonzalez-Gomez, R. *et al.* Educational disparities in brain health and dementia across Latin America and the United States. *Alzheimer's & Dementia: The Journal of the Alzheimer's Association* (2024).
- 34 Parra, M. A. *et al.* Biomarkers for dementia in Latin American countries: Gaps and opportunities. *Alzheimers Dement* **19**, 721-735, doi:10.1002/alz.12757 (2023).
- 35 Dubois, B. *et al.* Alzheimer Disease as a Clinical-Biological Construct—An International Working Group Recommendation. *JAMA Neurology*, doi:10.1001/jamaneurol.2024.3770 (2024).
- 36 Singh-Bains, M. K. *et al.* Altered microglia and neurovasculature in the Alzheimer's disease cerebellum. *Neurobiology of Disease* **132**, 104589, doi:<https://doi.org/10.1016/j.nbd.2019.104589> (2019).
- 37 Jacobs, H. I. L. *et al.* The cerebellum in Alzheimer's disease: evaluating its role in cognitive decline. *Brain* **141**, 37-47, doi:10.1093/brain/awx194 (2018).
- 38 Chen, Y. *et al.* Cerebellar atrophy and its contribution to cognition in frontotemporal dementias. *Annals of Neurology* **84**, 98-109, doi:<https://doi.org/10.1002/ana.25271> (2018).
- 39 Schmahmann, J. D. Cerebellum in Alzheimer's disease and frontotemporal dementia: not a silent bystander. *Brain* **139**, 1314-1318, doi:10.1093/brain/aww064 (2016).
- 40 Cavanagh, J. *et al.* Socioeconomic Status and the Cerebellar Grey Matter Volume. Data from a Well-Characterised Population Sample. *The Cerebellum* **12**, 882-891, doi:10.1007/s12311-013-0497-4 (2013).
- 41 Loued-Khenissi, L. *et al.* Signatures of life course socioeconomic conditions in brain anatomy. *Human Brain Mapping* **43**, 2582-2606, doi:<https://doi.org/10.1002/hbm.25807> (2022).
- 42 Kweon, H. *et al.* Human brain anatomy reflects separable genetic and environmental components of socioeconomic status. *Science advances* **8**, eabm2923 (2022).
- 43 Michael, C. *et al.* Socioeconomic resources in youth are linked to divergent patterns of network integration/segregation across the brain's transmodal axis. *PNAS Nexus* **3**, pgae412, doi:10.1093/pnasnexus/pgae412 (2024).
- 44 Qiu, S. *et al.* The ecology of poverty and children's brain development: A systematic review and quantitative meta-analysis of brain imaging studies. *Neuroscience & Biobehavioral Reviews* **169**, 105970, doi:<https://doi.org/10.1016/j.neubiorev.2024.105970> (2025).

- 45 Kleckner, I. R. *et al.* Evidence for a Large-Scale Brain System Supporting Allostasis and Interoception in Humans. *Nat Hum Behav* **1**, doi:10.1038/s41562-017-0069 (2017).
- 46 Migeot, J. A., Duran-Aniotz, C. A., Signorelli, C. M., Piguet, O. & Ibanez, A. A predictive coding framework of allostatic-interoceptive overload in frontotemporal dementia. *Trends Neurosci* **45**, 838-853, doi:10.1016/j.tins.2022.08.005 (2022).
- 47 Birba, A. *et al.* Allostatic-Interoceptive Overload in Frontotemporal Dementia. *Biological Psychiatry* **92**, 54-67, doi:<https://doi.org/10.1016/j.biopsych.2022.02.955> (2022).
- 48 Santamaría-García, H. *et al.* Allostatic Interoceptive Overload Across Psychiatric and Neurological Conditions. *Biological Psychiatry* **97**, 28-40, doi:10.1016/j.biopsych.2024.06.024 (2025).
- 49 Franco-O'Byrne, D., Santamaría-García, H., Migeot, J. & Ibáñez, A. Emerging Theories of Allostatic-Interoceptive Overload in Neurodegeneration. *Current topics in behavioral neurosciences*, doi:10.1007/7854_2024_471 (2024).
- 50 Migeot, J. & Ibáñez, A. in *Reference Module in Neuroscience and Biobehavioral Psychology* (Elsevier, 2023).
- 51 Alvarez, G. M., Rudolph, M. D., Cohen, J. R. & Muscatell, K. A. Lower Socioeconomic Position Is Associated with Greater Activity in and Integration within an Allostatic-Interoceptive Brain Network in Response to Affective Stimuli. *Journal of cognitive neuroscience* **34**, 1906-1927, doi:10.1162/jocn_a_01830 (2022).
- 52 Walhovd, K. B. *et al.* Education and Income Show Heterogeneous Relationships to Lifespan Brain and Cognitive Differences Across European and US Cohorts. *Cerebral Cortex* **32**, 839-854, doi:10.1093/cercor/bhab248 (2022).
- 53 Jeong, J., McCoy, D. C. & Fink, G. Pathways between paternal and maternal education, caregivers' support for learning, and early child development in 44 low- and middle-income countries. *Early Childhood Research Quarterly* **41**, 136-148, doi:<https://doi.org/10.1016/j.ecresq.2017.07.001> (2017).
- 54 Schady, N. Parents' Education, Mothers' Vocabulary, and Cognitive Development in Early Childhood: Longitudinal Evidence From Ecuador. *American Journal of Public Health* **101**, 2299-2307, doi:10.2105/AJPH.2011.300253 (2011).
- 55 Zheng, L., Qi, X. & Zhang, C. Can improvements in teacher quality reduce the cognitive gap between urban and rural students in China? *International Journal of Educational Development* **100**, 102781, doi:<https://doi.org/10.1016/j.ijedudev.2023.102781> (2023).
- 56 Mantri, S., Nwadiogbu, C., Fitts, W. & Dahodwala, N. Quality of education impacts late-life cognition. *International Journal of Geriatric Psychiatry* **34**, 855-862, doi:<https://doi.org/10.1002/gps.5075> (2019).
- 57 Barba, C. *et al.* Quality of Education and Late-Life Cognitive Function in a Population-Based Sample From Puerto Rico. *Innovation in Aging* **5**, igab016, doi:10.1093/geroni/igab016 (2021).
- 58 Townsend, R. *et al.* Nutrition for dementia prevention: a state of the art update for clinicians. *Age and Ageing* **53**, ii30-ii38, doi:10.1093/ageing/afae030 (2024).
- 59 Cao, W. *et al.* Complex association of self-rated health, depression, functional ability with loneliness in rural community-dwelling older people. *BMC geriatrics* **23**, 267, doi:10.1186/s12877-023-03965-4 (2023).
- 60 Pan, Z., Liu, Y., Liu, Y., Huo, Z. & Han, W. Age-friendly neighbourhood environment, functional abilities and life satisfaction: A longitudinal analysis of older adults in urban China. *Social Science & Medicine* **340**, 116403, doi:<https://doi.org/10.1016/j.socscimed.2023.116403> (2024).
- 61 Santamaria-Garcia, H. *et al.* Factors associated with healthy aging in Latin American populations. *Nat Med* **29**, 2248-2258, doi:10.1038/s41591-023-02495-1 (2023).
- 62 Ibáñez, A., Legaz, A. & Ruiz-Adame, M. Addressing the gaps between socioeconomic disparities and biological models of dementia. *Brain* **146**, 3561-3564, doi:10.1093/brain/awad236 (2023).
- 63 Sekher, T. V., Pai, M. & Muhammad, T. Subjective social status and socio-demographic correlates of perceived discrimination among older adults in India. *BMC geriatrics* **24**, 617, doi:10.1186/s12877-024-05114-x (2024).
- 64 Nylund, K. L., Asparouhov, T. & Muthén, B. O. Deciding on the Number of Classes in Latent Class Analysis and Growth Mixture Modeling: A Monte Carlo Simulation Study. *Structural Equation Modeling: A Multidisciplinary Journal* **14**, 535-569, doi:10.1080/10705510701575396 (2007).

- 65 Ibanez, A. *et al.* The Multi-Partner Consortium to Expand Dementia Research in Latin America (ReDLat): Driving Multicentric Research and Implementation Science. *Front Neurol* **12**, 631722, doi:10.3389/fneur.2021.631722 (2021).
- 66 Arce Rentería, M. *et al.* Representativeness of samples enrolled in Alzheimer's disease research centers. *Alzheimer's & Dementia: Diagnosis, Assessment & Disease Monitoring* **15**, e12450, doi:<https://doi.org/10.1002/dad2.12450> (2023).
- 67 Kobayashi, L. C. *et al.* Cross-national comparisons of later-life cognitive function using data from the Harmonized Cognitive Assessment Protocol (HCAP): Considerations and recommended best practices. *Alzheimer's & Dementia* **20**, 2273-2281, doi:<https://doi.org/10.1002/alz.13694> (2024).
- 68 Ibáñez, A. *et al.* The Multi-Partner Consortium to Expand Dementia Research in Latin America (ReDLat): Driving Multicentric Research and Implementation Science. *Frontiers in neurology* **12**, 303 (2021).
- 69 Maito, M. A. *et al.* Classification of Alzheimer's disease and frontotemporal dementia using routine clinical and cognitive measures across multicentric underrepresented samples: A cross sectional observational study. *The Lancet Regional Health-Americas* **17**, 100387 (2023).
- 70 Santamaria-Garcia, N. S.-B., A; Hernandez, H; Moguilner, S; Maito, M; Ochoa-Rosales, C; Corley, M; Valcour, V; Miranda, J; Lawlor, B; Ibanez, A. Heterogeneous risk factors impact aging in Latin American populations. *Nature Medicine* **Accepted**. (2023).
- 71 Moguilner, S. *et al.* Visual deep learning of unprocessed neuroimaging characterises dementia subtypes and generalises across non-stereotypic samples. *EBioMedicine* **90** (2023).
- 72 Fittipaldi, S. *et al.* Heterogeneous factors influence social cognition across diverse settings in brain health and age-related diseases. *Nature Mental Health* **2**, 63-75, doi:10.1038/s44220-023-00164-3 (2024).
- 73 Maito, M. A. *et al.* Classification of Alzheimer's disease and frontotemporal dementia using routine clinical and cognitive measures across multicentric underrepresented samples: A cross sectional observational study. *Lancet Reg Health Am* **17**, doi:10.1016/j.lana.2022.100387 (2023).
- 74 Rubinov, M. & Sporns, O. Complex network measures of brain connectivity: Uses and interpretations. *NeuroImage* **52**, 1059-1069, doi:<https://doi.org/10.1016/j.neuroimage.2009.10.003> (2010).
- 75 Briceño, E. M. *et al.* A cultural neuropsychological approach to harmonization of cognitive data across culturally and linguistically diverse older adult populations. *Neuropsychology* **37**, 247-257, doi:10.1037/neu0000816 (2023).
- 76 Gross, A. L. *et al.* Harmonisation of later-life cognitive function across national contexts: results from the Harmonized Cognitive Assessment Protocols. *The lancet. Healthy longevity* **4**, e573-e583, doi:10.1016/s2666-7568(23)00170-8 (2023).
- 77 Chung, J. *et al.* Normalization of cortical thickness measurements across different T1 magnetic resonance imaging protocols by novel W-Score standardization. *NeuroImage* **159**, 224-235, doi:10.1016/j.neuroimage.2017.07.053 (2017).
- 78 Piña-Escudero, S. D. *et al.* Social determinants of health questionnaire development for patients with Alzheimer's disease and frontotemporal dementia. *Alzheimer's & Dementia* **19**, e065204, doi:10.1002/alz.065204 (2023).
- 79 Nieto-Castanon, A. *Handbook of functional connectivity Magnetic Resonance Imaging methods in CONN*. (2020).
- 80 Arslan, S. *et al.* Human brain mapping: A systematic comparison of parcellation methods for the human cerebral cortex. *NeuroImage* **170**, 5-30, doi:10.1016/j.neuroimage.2017.04.014 (2018).
- 81 Nieto-Castanon, A. in *Handbook of functional connectivity Magnetic Resonance Imaging methods in CONN* 83-104 (Hilbert Press Boston, MA, 2020).
- 82 Van Bavel, J. J., Gadarian, S. K., Knowles, E. & Ruggeri, K. Political polarization and health. *Nature Medicine* **30**, 3085-3093, doi:10.1038/s41591-024-03307-w (2024).
- 83 Chater, N. & Loewenstein, G. The i-frame and the s-frame: How focusing on individual-level solutions has led behavioral public policy astray. *Behavioral and Brain Sciences* **46**, e147, doi:10.1017/S0140525X22002023 (2023).
- 84 World Bank. *Poverty and inequality platform*, <<https://pip.worldbank.org/home>> (

- 85 Gridley, K., Baxter, K. & Birks, Y. How do quantitative studies involving people with dementia
report experiences of standardised data collection? A narrative synthesis of NIHR published
studies. *BMC Medical Research Methodology* **24**, 43, doi:10.1186/s12874-024-02148-y (2024).
- 86 Park, S.-H. & Cho, Y. S. Predictive validity of the Cornell Scale for depression in dementia
among older adults with and without dementia: A systematic review and meta-analysis.
Psychiatry Research **310**, 114445, doi:<https://doi.org/10.1016/j.psychres.2022.114445> (2022).
- 87 Kahle-Wroblewski, K. *et al.* Assessing quality of life in Alzheimer's disease: Implications for
clinical trials. *Alzheimer's & dementia (Amsterdam, Netherlands)* **6**, 82-90,
doi:10.1016/j.dadm.2016.11.004 (2017).
- 88 Fish, J. in *Encyclopedia of Clinical Neuropsychology* (eds Jeffrey S. Kreutzer, John DeLuca,
& Bruce Caplan) 111-112 (Springer New York, 2011).
- 89 O'Callaghan, C. *et al.* Fair play: social norm compliance failures in behavioural variant
frontotemporal dementia. *Brain* **139**, 204-216, doi:10.1093/brain/awv315 (2016).
- 90 Garcia-Cordero, I. *et al.* Metacognition of emotion recognition across neurodegenerative
diseases. *Cortex; a journal devoted to the study of the nervous system and behavior* **137**, 93-
107, doi:10.1016/j.cortex.2020.12.023 (2021).
- 91 Palpatzis, E. *et al.* Lifetime Stressful Events Associated with Alzheimer's Pathologies,
Neuroinflammation and Brain Structure in a Risk Enriched Cohort. *Annals of Neurology* **95**,
1058-1068, doi:<https://doi.org/10.1002/ana.26881> (2024).
- 92 Pietrzak, R. H. *et al.* Plasma Cortisol, Brain Amyloid- β , and Cognitive Decline in Preclinical
Alzheimer's Disease: A 6-Year Prospective Cohort Study. *Biological Psychiatry: Cognitive
Neuroscience and Neuroimaging* **2**, 45-52, doi:<https://doi.org/10.1016/j.bpsc.2016.08.006>
(2017).
- 93 Mosconi, L. *et al.* Sex-specific associations of serum cortisol with brain biomarkers of
Alzheimer's risk. *Scientific Reports* **14**, 5519, doi:10.1038/s41598-024-56071-9 (2024).
- 94 Piguet, O., Hornberger, M., Mioshi, E. & Hodges, J. R. Behavioural-variant frontotemporal
dementia: diagnosis, clinical staging, and management. *The Lancet Neurology* **10**, 162-172,
doi:[https://doi.org/10.1016/S1474-4422\(10\)70299-4](https://doi.org/10.1016/S1474-4422(10)70299-4) (2011).
- 95 Morris, J. C. *et al.* Clinical Dementia Rating training and reliability in multicenter studies.
Neurology **48**, 1508-1510, doi:10.1212/WNL.48.6.1508 (1997).
- 96 Mioshi, E., Flanagan, E. & Knopman, D. Detecting clinical change with the CDR-FTLD:
differences between FTLD and AD dementia. *Int J Geriatr Psychiatry* **32**, 977-982,
doi:10.1002/gps.4556 (2017).
- 97 Taylor, P. A. *et al.* Demonstrating quality control (QC) procedures in fMRI. *Frontiers in
neuroscience* **17**, 1205928 (2023).
- 98 Firbank, M., Harrison, R., Williams, E. & Coulthard, A. Quality assurance for MRI: practical
experience. *The British journal of radiology* **73**, 376-383 (2000).
- 99 Liu, T. T. Noise contributions to the fMRI signal: An overview. *NeuroImage* **143**, 141-151 (2016).
- 100 Reeder, S. B. *et al.* Iterative decomposition of water and fat with echo asymmetry and least-
squares estimation (IDEAL): application with fast spin-echo imaging. *Magnetic resonance in
medicine* **54**, 636-644, doi:<https://doi.org/10.1002/mrm.20624> (2005).
- 101 Dietrich, O., Raya, J. G., Reeder, S. B., Reiser, M. F. & Schoenberg, S. O. Measurement of
signal-to-noise ratios in MR images: influence of multichannel coils, parallel imaging, and
reconstruction filters. *Journal of magnetic resonance imaging : JMRI* **26**, 375-385,
doi:<https://doi.org/10.1002/jmri.20969> (2007).
- 102 Murphy, K., Bodurka, J. & Bandettini, P. A. How long to scan? The relationship between fMRI
temporal signal to noise ratio and necessary scan duration. *NeuroImage* **34**, 565-574 (2007).

RESPONSE TO REVIEWERS

The following responses have been prepared to address the Reviewer's comments and suggestions point-by-point. All additions and modifications are highlighted in yellow here and in the revised version of the manuscript. All new references have been incorporated into the revised 'References' section.

REVIEWERS COMMENTS

REVIEWER #1

Comment 1.1. The authors responded appropriately to the reviewers' comments and in my view, the manuscript improved substantially. I don't have any more questions or suggestions to make.

Response 1.1. We appreciate your positive evaluation and constructive suggestions.

REVIEWER #2

Comment 2.1. The authors present a revised manuscript detailing an examination of the relationship between the multi-dimensional social exposome and cognitive function and brain structure/function, which they term "brain health," in a multi-site cohort of healthy and demented subjects from Latin America. The authors have provided responses to several reviewer requests that address some of the identified concerns. However, they have decided to not address previously identified issues related to parcel definition, which is critical for interpreting the brain's functional connectivity measures. In addition, justification for several data processing decisions reveal that the procedures are not appropriate. Based on this I do not accept this manuscript's conclusions; I primarily focus on these issues in my comments below.

Response 2.1. Thank you for your evaluation. We addressed your comments as outlined in the responses below.

Comment 2.2. I agree with authors that no single field standard on atlases exist. However, their rationale does not preclude ensuring the biological relevance of the atlas being used to examine correlations in activity among brain areas. It is known that brain areas do not often follow morphometric boundaries (as used in AAL) which predicated the need for the development of a new generation of atlases for purposes of localization and interrogation in neuroimaging, roughly 15 years ago. The authors respond that "no single parcellation method consistently outperforms others across metrics". However, while prior work has convincingly demonstrated that functional atlases can vary across metrics of validation, they still outperform the AAL atlas which is what the current conclusions rest on (e.g., Arslan, Neuroimage, 2018). The authors go on to indicate that "[they] selected the AAL atlas to ensure comparability with existing literature. This choice also allowed us to use the same atlas for structural and functional imaging analyses, aligning our study with previous works in dementia." The vast majority of the literature has moved away from this outdated atlas; further, the alternate atlases do not preclude examining the same regions for structural and functional imaging analyses as the authors contend. In summary, the functional connectivity results contain a known problem in their estimation, due to the use of inappropriate nodes. This issue is compounded by additional methodological issues which are described further below.

Response 2.2. Thank you for these comments. Now we have re-run functional connectivity analysis and replaced Automated Anatomical Labeling (AAL) results by the Brainnetome atlas¹ (adding AAL cerebellar regions), a connectivity-based parcellation atlas that captures both cortical and subcortical regions, better suited to functional connectivity analysis. The new results are even more robust than the previous ones, and we appreciate the Reviewer's suggestion.

Results, Materials and Methods, Tables, and the Figure associated with these results were updated, and the interpretation of results was refined based on new findings. We retained the previous AAL results in the Supplementary Material for comparison with earlier functional connectivity studies on dementia in Latin America, which typically utilize the AAL atlas²⁻¹³. The following changes have been performed:

(a) Results (tables omitted in this document for brevity):

Adverse MSE and altered brain connectivity in dementia

In individuals with AD, more adverse MSE was associated with reduced frontotemporal connectivity between the anterior cingulate cortex, hippocampus, parahippocampal gyrus, striatum, insula, and cerebellum (lobule III and vermis lobule III). Conversely, hyperconnectivity was observed between frontal (superior, middle inferior), temporal (middle), occipital (middle, cuneus), and cingulate regions (Figure 4A, Table 5). The middle temporal, cingulate, and superior frontal gyrus had the most connections (Figure 4A).

In FTLD, more adverse MSE was associated with decreased frontotemporal connectivity primarily between frontal (superior, middle, and inferior, precentral gyrus), temporal (superior, parahippocampal gyrus), limbic (amygdala, hippocampus, and cingulate cortex), and cerebellar regions (crus I, lobule III, IV-V, IX, and X). Hyperconnectivity in frontal (superior, middle inferior), temporal (superior, middle, inferior, and parahippocampal gyrus), hippocampus, and cerebellar regions (crus I, lobule IV-V, VI, and X) (Figure 4B, Table 6) was observed. The regions with the highest number of connections were the hippocampus, inferior and superior frontal gyrus, and cingulate gyrus (Figure 4B).

Figure 4. Associations between multidimensional social exposome and brain connectivity. **a**, Functional connectivity in AD displayed through ROI-to-ROI connectivity maps, where color-coded edges represent significant t-values for decreased (red) and increased (blue) connectivity associated with MSE. The connectome ring provides a simplified visualization of interconnections between key regions. The circular barplot chart highlights top regions with the highest number of connections. **b**, In FTLD, ROI-to-ROI connectivity maps similarly provide visualizations of significant connections associated with MSE, with the connectome ring displaying interconnections between key regions and circular barplot chart showing key regions with the highest number of connections. AG = Angular Gyrus; Cer = Cerebellum; CG = Cingulate Gyrus; Cun = Cuneus; FuG = Fusiform Gyrus; Hipp = Hippocampus; IFG = Inferior Frontal Gyrus; INS = Insula; ITG = Inferior Temporal Gyrus; MFG = Middle Frontal Gyrus; MTG = Middle Temporal Gyrus; OcG = Occipital Gyrus; PCL = Paracentral Lobule; Pcun = Precuneus; PhG = Parahippocampal Gyrus; PoCG = Postcentral Gyrus; PrG = Precentral Gyrus; SFG = Superior

Frontal Gyrus; SmG = Supramarginal Gyrus; SPL = Superior Parietal Lobule; STG = Superior Temporal Gyrus; Str = Striatum; Thal = Thalamus.

(b) Refined interpretation of findings:

Discussion

[...] Potential compensatory functional hyperconnectivity was observed primarily in regions with reduced volume that were associated with more adverse MSE. [...]

[...] Conversely, hyperconnectivity is aligned with evidence on compensatory effects in AD¹⁴ and FTLN¹⁵, as well as adversity-related factors such as socioeconomic status¹⁶, financial stress¹⁷, and discrimination¹⁸. This may reflect a compensatory process triggered by brain burdens associated with (a) neurodegenerative processes^{14,15}, (b) disease-specific environmental factors contributing to pathophysiological mechanism¹⁹⁻²¹, and (c) their interactions. In line with the prolonged accumulation of amyloid and tau pathology and with environmental exposures linked to the later onset of AD²²⁻²⁵, stronger compensatory hyperconnectivity was observed in this group compared to FTLN. This may reflect the longer-lasting impact of degenerative and environmental burdens on brain structural signatures (atrophy) in AD, whereas in FTLN, more transient compensatory changes (functional hyperconnectivity) may emerge earlier in the disease course due to its typically younger onset. [...]

(d) Materials and methods

Resting-state fMRI preprocessing

[...] Pearson correlation coefficients were computed between the average BOLD time series of each pair of regions of interest (ROIs) from the Brainnetome atlas¹, a structural and functional connectivity-based parcellation atlas that capture both cortical and subcortical regions, better suited to functional connectivity analysis. Automated Anatomical Labeling (AAL) atlas²⁶ cerebellar regions were added, generating a total of 272 x 272 ROIs correlation matrix for each participant. [...]

Here, we also outline the changes associated with functional connectivity within the AAL for replication purposes.

(c) Sensitivity analysis (tables omitted in this document for brevity):

Results

Sensitivity analyses

[...] To provide a framework for replication with previous evidence on functional connectivity in LA populations²⁻¹³, we ran additional models employing the Automated Anatomical Labeling (AAL) atlas. Results matched the connectivity patterns described in the main results (**Supplementary Information 4, Supplementary Tables 7-8**)

Supplementary Information

Supplementary Information 4. Functional connectivity analysis with AAL atlas

We conducted regression analyses via parametric tests with the MSE score as a predictor of the whole-brain ROI-to-ROI connectivity employing the AAL atlas. Like the main functional connectivity analysis described in the manuscript, age, sex, and scanner effects were included as covariates of no interest. Each patient group was analyzed concurrently with healthy controls (HCs). All results were P_{FDR}

corrected at the connection level, accounting for the size of the ROI-to-ROI matrix for each seed ROI, as implemented in the CONN toolbox²⁷.

Results replicated the main functional connectivity analysis, showing fronto-temporo-cerebellar associations and greater compensatory hyperconnectivity in AD than FTLD (**Supplementary Tables 7-8**). Cerebellar associations were more pronounced across groups, probably because the AAL atlas natively includes cerebellar regions²⁶, unlike the Brainnetome¹, which required an external addition of cerebellum.

Comment 2.3. The authors have provided some data comparing MSE to individual metrics alongside information on how the MSE improves on prior approaches that incorporate summary measures of SES. However, the authors do not elaborate on what prior work has told us about the exposome and brain structure, function, cognition, and dementia relations, and how their work fits in with these findings, which is a critical part of scholarship.

Response 2.3. Thank you for this observation. We have now elaborated on prior work examining the relationship between the exposome and brain health, including cognition, functional ability, brain and dementia outcomes.

Discussion

Different exposomes impact clinical, cognitive, and brain outcomes in healthy aging and dementia. Physical exposome factors—such as lead exposure²⁸, microplastics²⁹, heatwaves³⁰, and air pollution³¹—affect cognitive performance and brain structure. Lifestyle exposomes, including physical activity, smoking, alcohol consumption, and cognitive engagement in late life, are associated with dementia risk³¹. Specific components related to social exposome—including education, social isolation, socioeconomic status (SES), and structural inequality—have been linked to cognition, functional ability³², and alterations in brain structure and functional connectivity³⁻⁵. The complex social exposome in AD³³ and FTLD³⁴ suggests a lifespan cumulative influence^{19-21,31,35}.

[...] Components of adverse social exposome in AD are associated with dysregulated stress responses, linked with reduced cognition^{36,37}, impaired functional ability³⁸, increased neuropsychiatric symptoms³⁹, and brain burden²³.

[...] Emerging evidence in FTLD links altered stress responses with neurodegenerative processes via multisystemic dysregulations^{19,21}. Overall, our MSE approach offers a combined and granular characterization of the social exposome in dementia. This work supports current efforts to define the broader dementia exposome^{33,34}, enabling more precise risk profiling.

Comment 2.4. a) Authors chose to point to the CONN toolbox manual for reviewers to learn the TFCE procedure instead of providing an explicit explanation. CONN TFCE expects a group level clustered T or F statistical matrix with a user selected h parameter to smooth over matrix data for later permutation testing at a given FDR threshold. Authors chose to not describe their cluster arrangement, analysis performed to generate each statistical matrix, or the necessary h parameter selected. The lack of detail prevents readers from replicating their atypical correction for multiple comparisons procedure.

b) The procedure for calculating degree centrality for later testing with MSE is still poorly justified and described. For instance, did authors use their TFCE corrected test statistic matrix as a threshold, such that only significant "clusters" within AAL atlas lobes were summed in degree centrality calculation (assuming the matrix was spatially organized somehow, which seems impossible to achieve)? Similarly, was degree centrality calculated with a weighted (i.e., t-value) or unweighted (i.e., binary) matrix? Too few details are given to support future replication and the absence of adequate explanation of these critical aspects of analysis diminishes confidence in the results.

c) Authors clarify that their "motion scrubbing" procedure refers to spike regression of frames flagged at .05mm FD or 3 standard deviations away from mean global signal - which is a vary lenient motion scrubbing (i.e., removal of frames) procedure. Given that older adults express higher rates of motion, the threshold is far too liberal to accept the results as reported (see Ciric et al., 2017). As a result, it is likely that the dataset contains many erroneous correlations that are being attributed to biological signal. Future efforts should work to ensure that resting-state data are adequately cleaned using appropriate data processing parameters.

Response 2.4. Thank you for this comment. In response to your concerns, and to facilitate comparisons between the two functional atlases (Brainnetome and AAL atlas, see **response 2.2**), we have implemented a False Discovery Rate (FDR) correction at the ROI level. For gray matter volume analysis, we have described the extent and height of cluster parameters for TFCE. We have also replaced 'degree' with 'number of connections' throughout the text and figure and preprocessed functional connectivity data with more stringent motion scrubbing parameters. The following changes have been implemented in the new version of the manuscript:

(a) Multiple comparisons correction for functional connectivity:

Results

Section 2.2 details the changes and new results in the functional connectivity section.

Materials and methods

Statistical analyses

Resting-state functional connectivity associations with MSE

[...] All results were P_{FDR} corrected at the connection level for the size of the ROI-to-ROI matrix for each ROI, as implemented in CONN toolbox²⁷ [...]

(b) Description of TFCE cluster parameters for gray matter volume:

Materials and methods

Statistical analyses

Grey matter associations with MSE

[...] Extent and height cluster parameters were set to 0.5 and 2, respectively, following standard approaches⁴⁰. [...]

(c) Replacement of 'degree' by 'number of connections':

Materials and methods

Statistical analyses

Resting-state functional connectivity associations with MSE

[...] The **number of connections** of each ROI was calculated to assess its contribution to the broader connectivity patterns associated with MSE. [...]

We have verified that the term 'degree' has been replaced by 'number of connections' in the figure legends and other sections of the manuscript. Modifications Figure 3 were addressed in **Response 2.2**.

(d) Description of updated motion scrubbing parameters:

Materials and Methods

Neuroimaging acquisition and preprocessing

Resting-state fMRI preprocessing

[...] A motion correction technique was applied by rigidly aligning fMRI volumes to T1-weighted images, ensuring that the impact of motion artifacts was minimized⁴¹. Motion scrubbing was then applied using framewise displacement > 0.2 mm and temporal derivative variance across space > 5%⁴², which are stricter than conventional thresholds, to flag and remove high-motion frames. This was implemented using the artifact detection tools within the CONN toolbox⁴³, using a conservative setting (FD = 0.2 mm, global signal Z = 5) to remove motion artefacts while preserving the biological signal [...]

REVIEWER #3

Comment 3.1. Dear authors,

I greatly appreciate the time and effort you all put in to address my comments as well as those of other reviewers. Excellent job.

I just have very minor lingering comments.

When you described the model fit for your sensitivity analysis for country effects in your supplementary information, you described that "The model showed good fit: CFI = 0.868, TLI = 0.767, RMSEA = 0.088, SRMR = 0.061". However, CFI/TLI are in the poor range and RMSEA/SRMR are bordering acceptable.

Response 3.1. Thank you for your evaluation and feedback. As part of the sensitivity analysis, country effects were now evaluated using a regression-based approach⁴⁴ to prevent overfitting and potential artefactual misfit in the SEM model's fit indices⁴⁵.

Supplementary information

Supplementary Information 1. Sensitivity analysis for country effects

[...] The inclusion of multiple dummy variables⁴⁶⁻⁴⁸ for factors such as country can be considered suboptimal, as it may introduce unnecessary complexity into SEM models⁴⁵. Although such models may be well-specified and theoretically justified, they can still yield maladjusted fit indices⁴⁵. Consequently, we conducted multiple linear regression analyses using MSE scores as predictors of cognition, functional ability, and neuropsychiatric symptoms, with country included as dummy covariates⁴⁴. All models were statistically significant ($p < 0.001$) and MSE effects were replicated for cognition ($\beta = 0.91$, $p < 0.001$), functional ability ($\beta = 0.41$, $p < 0.001$), and neuropsychiatric symptoms ($\beta = 0.43$, $p < 0.001$).

Comment 3.2. Lastly, when describing the recruitment efforts, authors mentioned that participants were recruited "from extensive networks". A bit more information regarding what is meant by extensive networks, like what kind of networks? would be helpful for future investigators seeking to work in LA

Response 3.2. Thank you for this observation. We have provided more information on the extensive networks employed for participants' recruitment.

Materials and methods

Participants

[...] Participants were recruited from extensive networks including (a) clinical networks, involving memory clinics, neurology departments, and affiliated hospitals; (b) academic collaborations, leveraging partnerships with universities and research institutions; (c) community outreach programs, engaging with local communities through informational sessions, and culturally tailored materials to encourage participation from rural and urban populations with diverse socioeconomic backgrounds; and (d) public health initiatives and local organizations, integrating recruitment efforts with public health campaigns and community groups to raise awareness and facilitate participation. These efforts allowed us to include individuals from rural and urban settings, focusing on underrepresented groups, as demonstrated previously with our ReDLat cohort marked by socioeconomic inequality⁴ and educational disparities³ [...]

REVIEWER #4

Comment 4.1. This is the revised version of the manuscript that I had the chance to review previously. While seemingly exhaustive, the answers to my queries and points of criticism are mostly vague. Most importantly, the way addressing the socio-demographical aspect of the study with focus on Latin America drawing up individual-level conclusions from group level geo-spatial data falls under the description "ecological fallacy". If the goal is to address risk exposures adequately, then we need a true geo-spatially referenced analysis where within- and across-countries differences and similarities will be explicitly tested on a geo-coded reference basis.

Response 4.1. We appreciate your comment. We have now clarified that all our analyses are conducted at the individual level, not the aggregate level, including the derivation of the MSE and the association with clinical and neuroimaging outcomes. In addition, we have stated that future research should also incorporate spatially referenced methods and geocoded data referencing regional disparities.

Discussion

[...] Similar to previous studies from our group and others^{3,49}, all primary analyses are based on individual measurements across countries. Our results do not comprise aggregate-level data. Future studies should incorporate nested designs with aggregated-level analyses, spatially referenced methods, and geocoded data relevant to brain health in LA, such as structural inequality⁴, air pollution⁵, and access to green spaces⁵⁰ [...]

Comment 4.2. One of my suggestions to compare the data at hand with other large-scale open access epidemiological data - e.g. the UK Biobank, was not followed suit. Similarly, there are still many orthographic mistakes - see "finanTial status" in Fig.2 and counterintuitive brain morphometry results (Fig. 3) that show implausible association patterns of the multi-dimensional social exposome with brain structure in patients with AD and FTLTLD.

Response 4.2. Thank you for these careful observations. We believe that comparing our data with datasets like the UK Biobank is beyond the scope and context of our study. As detailed in the previous submission, several reasons were considered. The main reason is that our paper—clearly stated in the title, abstract, objectives, methods, and analyses—focuses specifically on Latin American populations. As such, comparisons to UK cohorts, such as the UK Biobank, are beyond the scope of our research. Additional reasons include challenges in harmonizing our protocol with open datasets and the lack of comparable assessments. Unlike datasets such as the UK Biobank, which primarily represent high-income, well-resourced populations with distinct social and environmental exposures, our study centers on underrepresented, socioeconomically diverse, and more heterogeneous Latin American populations^{7,49}. Our questionnaire includes hundreds of items that were validated across each participating country and offer a level of granularity not available in the

UK Biobank or ADNI. Given these characteristics, direct comparisons would not only be methodologically inappropriate but could also misrepresent the socio-cultural and structural determinants specific to the region.

For these reasons, we requested permission from the Editors to focus solely on the current datasets. The Editors approved this decision, confirming that no additional databases are required.

We have now reviewed the manuscript in depth to ensure that there are no typographical errors in the text, tables, figures, and supplementary material. Additional explanations and references were included to support our gray matter volume associations. The following text has been added to the manuscript:

Discussion

The brain correlates of adverse MSE in dementia align with disease-specific environmental hypotheses regarding the pathophysiological mechanisms in AD. These suggest lower resistance to p-tau and A β burden across frontotemporal regions associated with lifetime stressful events²³, stress-mediated impaired insulin signaling and cerebellar vulnerability²⁰, and inflammatory models of clinical progression^{23,37}

[...] However, FTLN brain health outcomes were linked with social exposome in this study. Evidence of pathophysiological mechanisms suggests that the brain burden observed in FTLN may be driven by the accumulation of physiological stressors over time. Cerebellar gray matter patterns with posterior involvement are associated with educational disparities³ and structural inequality⁴. Evidence on pathophysiological mechanisms links the brain burden to the accumulation of physiological stressors (i.e., allostatic overload) in FTLN^{19,21} and also relates it to SDH³⁻⁵, targeting both disease-specific and non-specific regions.

References

- 1 Fan, L. *et al.* The Human Brainnetome Atlas: A New Brain Atlas Based on Connectional Architecture. *Cerebral Cortex* **26**, 3508-3526, doi:10.1093/cercor/bhw157 (2016).
- 2 Fittipaldi, S. *et al.* Heterogeneous factors influence social cognition across diverse settings in brain health and age-related diseases. *Nature Mental Health* **2**, 63-75, doi:10.1038/s44220-023-00164-3 (2024).
- 3 Gonzalez-Gomez, R. *et al.* Educational disparities in brain health and dementia across Latin America and the United States. *Alzheimer's & Dementia: The Journal of the Alzheimer's Association* (2024).
- 4 Legaz, A. *et al.* Structural inequality linked to brain volume and network dynamics in aging and dementia across the Americas. *Nature Aging*, doi:10.1038/s43587-024-00781-2 (2024).
- 5 Moguilner, S. *et al.* Brain clocks capture diversity and disparities in aging and dementia across geographically diverse populations. *Nat Med*, doi:10.1038/s41591-024-03209-x (2024).
- 6 Legaz, A. *et al.* Multimodal mechanisms of human socially reinforced learning across neurodegenerative diseases. *Brain* **145**, 1052-1068, doi:10.1093/brain/awab345 (2022).
- 7 Birba, A. *et al.* Allostatic-Interoceptive Overload in Frontotemporal Dementia. *Biological Psychiatry* **92**, 54-67, doi:<https://doi.org/10.1016/j.biopsych.2022.02.955> (2022).
- 8 Prado, P. *et al.* Source space connectomics of neurodegeneration: One-metric approach does not fit all. *Neurobiology of Disease* **179**, 106047, doi:<https://doi.org/10.1016/j.nbd.2023.106047> (2023).
- 9 Legaz, A. *et al.* Social and non-social working memory in neurodegeneration. *Neurobiology of Disease* **183**, 106171, doi:<https://doi.org/10.1016/j.nbd.2023.106171> (2023).
- 10 Prado, P. *et al.* Harmonized multi-metric and multi-centric assessment of EEG source space connectivity for dementia characterization. *Alzheimer's & Dementia: Diagnosis, Assessment & Disease Monitoring* **15**, e12455, doi:<https://doi.org/10.1002/dad2.12455> (2023).
- 11 Coronel-Oliveros, C. *et al.* Viscous dynamics associated with hypoexcitation and structural disintegration in neurodegeneration via generative whole-brain modeling. *Alzheimer's & Dementia* **20**, 3228-3250, doi:<https://doi.org/10.1002/alz.13788> (2024).

- 12 Moguilner, S. *et al.* Biophysical models applied to dementia patients reveal links between geographical origin, gender, disease duration, and loss of neural inhibition. *Alzheimer's Research & Therapy* **16**, 79, doi:10.1186/s13195-024-01449-0 (2024).
- 13 Hazelton, J. L. *et al.* Altered spatiotemporal brain dynamics of interoception in behavioural-variant frontotemporal dementia. *eBioMedicine* **113**, 105614, doi:<https://doi.org/10.1016/j.ebiom.2025.105614> (2025).
- 14 Roemer-Cassiano, S. N. *et al.* Amyloid-associated hyperconnectivity drives tau spread across connected brain regions in Alzheimer's disease. *Science Translational Medicine* **17**, eadp2564, doi:10.1126/scitranslmed.adp2564.
- 15 Agosta, F. *et al.* Functional Connectivity From Disease Epicenters in Frontotemporal Dementia. *Neurology* **100**, e2290-e2303, doi:10.1212/WNL.0000000000207277 (2023).
- 16 Alvarez, G. M., Rudolph, M. D., Cohen, J. R. & Muscatell, K. A. Lower Socioeconomic Position Is Associated with Greater Activity in and Integration within an Allostatic-Interoceptive Brain Network in Response to Affective Stimuli. *Journal of cognitive neuroscience* **34**, 1906-1927, doi:10.1162/jocn_a_01830 (2022).
- 17 Suárez-Pellicioni, M. & McDonough, I. M. Separating neurocognitive mechanisms of maintenance and compensation to support financial ability in middle-aged and older adults: The role of language and the inferior frontal gyrus. *Archives of Gerontology and Geriatrics* **130**, 105705, doi:<https://doi.org/10.1016/j.archger.2024.105705> (2025).
- 18 Han, S. D. *et al.* Self-reported experiences of discrimination in older black adults are associated with insula functional connectivity. *Brain imaging and behavior* **15**, 1718-1727, doi:10.1007/s11682-020-00365-9 (2021).
- 19 Migeot, J. A., Duran-Aniotz, C. A., Signorelli, C. M., Piguet, O. & Ibanez, A. A predictive coding framework of allostatic-interoceptive overload in frontotemporal dementia. *Trends Neurosci* **45**, 838-853, doi:10.1016/j.tins.2022.08.005 (2022).
- 20 De Felice, F. G., Goncalves, R. A. & Ferreira, S. T. Impaired insulin signalling and allostatic load in Alzheimer disease. *Nat Rev Neurosci* **23**, 215-230, doi:10.1038/s41583-022-00558-9 (2022).
- 21 Migeot, J. & Ibáñez, A. in *Reference Module in Neuroscience and Biobehavioral Psychology* (Elsevier, 2023).
- 22 Antonioni, A. *et al.* Frontotemporal dementia, where do we stand? A narrative review. *International Journal of Molecular Sciences* **24**, 11732 (2023).
- 23 Palpatzis, E. *et al.* Lifetime Stressful Events Associated with Alzheimer's Pathologies, Neuroinflammation and Brain Structure in a Risk Enriched Cohort. *Annals of Neurology* **95**, 1058-1068, doi:<https://doi.org/10.1002/ana.26881> (2024).
- 24 Pietrzak, R. H. *et al.* Plasma Cortisol, Brain Amyloid- β , and Cognitive Decline in Preclinical Alzheimer's Disease: A 6-Year Prospective Cohort Study. *Biological Psychiatry: Cognitive Neuroscience and Neuroimaging* **2**, 45-52, doi:<https://doi.org/10.1016/j.bpsc.2016.08.006> (2017).
- 25 Mosconi, L. *et al.* Sex-specific associations of serum cortisol with brain biomarkers of Alzheimer's risk. *Scientific Reports* **14**, 5519, doi:10.1038/s41598-024-56071-9 (2024).
- 26 Tzourio-Mazoyer, N. *et al.* Automated anatomical labeling of activations in SPM using a macroscopic anatomical parcellation of the MNI MRI single-subject brain. *Neuroimage* **15**, 273-289, doi:10.1006/nimg.2001.0978 (2002).
- 27 Nieto-Castanon, A. *Handbook of functional connectivity Magnetic Resonance Imaging methods in CONN*. (2020).
- 28 Shao, K., Yu, Y., Ritz, B. & Paul, K. C. DNA methylation biomarkers for cumulative lead exposures and cognitive impairment. *Environmental Research* **264**, 120304, doi:<https://doi.org/10.1016/j.envres.2024.120304> (2025).
- 29 Nihart, A. J. *et al.* Bioaccumulation of microplastics in decedent human brains. *Nature Medicine*, doi:10.1038/s41591-024-03453-1 (2025).
- 30 Gong, J., Part, C. & Hajat, S. Current and future burdens of heat-related dementia hospital admissions in England. *Environ Int* **159**, 107027, doi:10.1016/j.envint.2021.107027 (2022).
- 31 Livingston, G. *et al.* Dementia prevention, intervention, and care: 2024 report of the Lancet standing Commission. *The Lancet*, doi:10.1016/S0140-6736(24)01296-0 (2024).
- 32 Santamaria-Garcia, H. *et al.* Factors associated with healthy aging in Latin American populations. *Nat Med* **29**, 2248-2258, doi:10.1038/s41591-023-02495-1 (2023).

- 33 Finch, C. E. & Kulminski, A. M. The Alzheimer's Disease Exposome. *Alzheimer's & Dementia* **15**, 1123-1132, doi:<https://doi.org/10.1016/j.jalz.2019.06.3914> (2019).
- 34 Sakowski, S. A., Koubek, E. J., Chen, K. S., Goutman, S. A. & Feldman, E. L. Role of the Exposome in Neurodegenerative Disease: Recent Insights and Future Directions. *Annals of Neurology* **95**, 635-652, doi:<https://doi.org/10.1002/ana.26897> (2024).
- 35 Lupien, S. J., McEwen, B. S., Gunnar, M. R. & Heim, C. Effects of stress throughout the lifespan on the brain, behaviour and cognition. *Nature Reviews Neuroscience* **10**, 434-445, doi:10.1038/nrn2639 (2009).
- 36 Sotiropoulos, I. *et al.* Stress Acts Cumulatively To Precipitate Alzheimer's Disease-Like Tau Pathology and Cognitive Deficits. *The Journal of Neuroscience* **31**, 7840, doi:10.1523/JNEUROSCI.0730-11.2011 (2011).
- 37 Csernansky, J. G. *et al.* Plasma Cortisol and Progression of Dementia in Subjects With Alzheimer-Type Dementia. *American Journal of Psychiatry* **163**, 2164-2169, doi:10.1176/ajp.2006.163.12.2164 (2006).
- 38 Perry, B. L. *et al.* Why the cognitive "fountain of youth" may be upstream: Pathways to dementia risk and resilience through social connectedness. *Alzheimer's & Dementia* **18**, 934-941, doi:<https://doi.org/10.1002/alz.12443> (2022).
- 39 Ouanes, S., Rabl, M., Clark, C., Kirschbaum, C. & Popp, J. Persisting neuropsychiatric symptoms, Alzheimer's disease, and cerebrospinal fluid cortisol and dehydroepiandrosterone sulfate. *Alzheimer's Research & Therapy* **14**, 190, doi:10.1186/s13195-022-01139-9 (2022).
- 40 Smith, S. M. & Nichols, T. E. Threshold-free cluster enhancement: addressing problems of smoothing, threshold dependence and localisation in cluster inference. *Neuroimage* **44**, 83-98, doi:10.1016/j.neuroimage.2008.03.061 (2009).
- 41 Jenkinson, M., Bannister, P., Brady, M. & Smith, S. Improved Optimization for the Robust and Accurate Linear Registration and Motion Correction of Brain Images. *NeuroImage* **17**, 825-841, doi:<https://doi.org/10.1006/nimg.2002.1132> (2002).
- 42 Ciric, R. *et al.* Benchmarking of participant-level confound regression strategies for the control of motion artifact in studies of functional connectivity. *Neuroimage* **154**, 174-187, doi:10.1016/j.neuroimage.2017.03.020 (2017).
- 43 Nieto-Castanon, A. Handbook of functional connectivity Magnetic Resonance Imaging methods in CONN. 108 (2020).
- 44 Aknin, L. B. *et al.* Policy stringency and mental health during the COVID-19 pandemic: a longitudinal analysis of data from 15 countries. *The Lancet Public Health* **7**, e417-e426, doi:10.1016/S2468-2667(22)00060-3 (2022).
- 45 Shi, D., Lee, T. & Maydeu-Olivares, A. Understanding the Model Size Effect on SEM Fit Indices. *Educational and psychological measurement* **79**, 310-334, doi:10.1177/0013164418783530 (2019).
- 46 Shipley, B. *Cause and Correlation in Biology: A User's Guide to Path Analysis, Structural Equations and Causal Inference with R*. 2 edn, (Cambridge University Press, 2016).
- 47 Byrne, B. M. *Structural Equation Modeling with EQS and EQS-Windows: Basic Concepts, Applications, and Programming*. (Sage Publications, Inc., 1994).
- 48 Kline, R. B. *Principles and practice of structural equation modeling, 4th ed.* (Guilford Press, 2016).
- 49 Llibre-Guerra, J. J. *et al.* Social determinants of health but not global genetic ancestry predict dementia prevalence in Latin America. *Alzheimer's & Dementia*, doi:<https://doi.org/10.1002/alz.14041> (2024).
- 50 Rojas-Rueda, D., Vaught, E. & Buss, D. Why a New Research Agenda on Green Spaces and Health Is Needed in Latin America: Results of a Systematic Review. *Int J Environ Res Public Health* **18**, doi:10.3390/ijerph18115839 (2021).

RESPONSE TO REVIEWERS

The following responses have been prepared to address the Reviewer's comments and suggestions point-by-point. All additions and modifications are highlighted in yellow here and in the revised version of the manuscript. All new references have been incorporated into the revised 'References' section.

REVIEWERS COMMENTS

REVIEWER #2

Comment 2.1. The authors present a second revised manuscript detailing an examination of the relationship between the multi-dimensional social exposome and cognitive function and brain structure/function, which they term "brain health," in a multi-site cohort of healthy and demented subjects from Latin America. The authors have provided responses to several more reviewer requests that address some but not all concerns. There remain vague statements and missing information on the procedures used that need clarity for future replication of their work, detailed below.

Response 2.1. Thank you for your evaluation. We have addressed your comments below.

Comment 2.2. There are several instances where authors use terms like "hypoconnectivity" and "hyperconnectivity" that imply a comparison occurred such that x group connectivity measure was 'reduced' or 'increased' relative to y group measure, respectively. The authors should be more direct to avoid aid understanding of results, so readers understand the context of the comparison made in the text where it is stated. Examples can be found on pgs 11, 14, 17, 18.

Response 2.2. We replaced terms like "hyperconnectivity" and "reduced" with phrasing that conveys the intended meaning of higher connectivity associated with higher MSE adversity.

Results

Adverse MSE and altered brain connectivity in dementia

[...] more adverse MSE was associated with lower frontotemporal connectivity [...] Conversely, greater MSE adversity was associated with higher connectivity [...].

[...] more adverse MSE was associated with lower frontotemporal connectivity [...] Greater MSE adversity was associated with higher connectivity [...].

Discussion

[...] Potential compensatory functional connectivity, characterized by higher functional connectivity associated with higher MSE adversity, [...]

[...] Conversely, greater functional connectivity is associated with greater MSE adversity [...]

[...] stronger compensatory functional connectivity associated with greater MSE adversity was observed in this group compared to FTLD [...]

Comment 2.3. Regarding degree centrality, which is now termed "number of connections", the concern was over the procedure described to calculate degree centrality. Right now, I assume the density threshold is based on only significant, positively weighted (not absolute value) FDR corrected edges (the Fisher's z correlation matrix or stat matrix t value where cell values are above 0 and contain information on roi to roi relations), that were binarized and set to a value of one, so each node's significant positively weighted edge could contribute a count of 1 to the degree centrality or "number of connections" for a single node. If this is accurate to what was done using the CONN toolbox, it should be stated as such so others can understand the procedure and

replicate the work. Additional references that describe alternate methods of calculating degree centrality include: Guimerà & Nunes Amaral, 2005; Sporns, 2018). Again, the number of connections is not equivalent to a static functional connectivity measure or total connectivity strength (sum of weighted edges where edge is a continuous value) and should not be interpreted as such as authors suggest on pg 34.

Response 2.3. We have provided a more detailed description of the degree/number of connections calculation in the revised Methods section and removed the misinterpretation of its value.

Materials and methods

Statistical analyses

Resting-state functional connectivity associations with MSE

[...] Following alternate methods of calculating degree centrality¹⁻³, we considered edges that were positively or negatively weighted and statistically significant (FDR-corrected). These correspond to values different from zero in the *t*-statistic matrix, reflecting meaningful ROI-to-ROI relationships. These edges were binarized—assigned a value of one—so that each significant, weighted edge contributes a count of 1 to a node's degree centrality, representing the number of connections each ROI exhibits in relation to MSE. [...]

Comment 2.4. I appreciate the responsiveness to more rigorous resting-state preprocessing (particularly the scrubbing threshold). Please provide additional details about frame retention so that readers can understand data quantities that go into analysis (range of resting-state frames remaining, threshold of minimum number of 'cleaned' frames allowed for analysis, etc).

Response 2.4. Thanks for your observation. We have added more details about frame retention and new sensitivity analysis excluding subjects with less than 70% of artifact-free frames to avoid systematic sampling bias in clinical and aging populations⁴⁻⁸, as follows:

Results

Sensitivity analysis

[...] Further, we tested separate models excluding participants with less than 70% of artifact-free frames⁴. Results replicated the main functional connectivity effects (**Supplementary Tables 5-6**) [...]

Materials and methods

Neuroimaging acquisition and preprocessing

Resting-state fMRI preprocessing

[...] The mean proportion of artifact-free to rejected frames was 92% (SD = 1.26), with values ranging from 30% to 100% [...]

Statistical analyses

Sensitivity analysis

[...] To further test the effect of motion correction, we applied two procedures: rigid realignment of fMRI volumes to T1-weighted images and scrubbing of high-motion frames. Scrubbing was based on framewise displacement > 0.2 mm and global signal change > 5% (**Materials and Methods, Neuroimaging Acquisition and Preprocessing**). Participants with less than 70% of artifact-free frames were excluded (**Supplementary Tables 5-6**).

REVIEWER #4

Comment 4.1. I greatly appreciate the authors' effort to align their responses to my points of criticism, but I have to insist that the responses miss the point.

Response 4.1. Thanks for your feedback. We have addressed your new comments below.

Comment 4.2. Wrt "ecological fallacy" – the issue here is that we need a subject-level analysis that tests each individual against the neighbourhood within a spatial buffer (e.g. btw 100 and 1000 meters diameter).

Response 4.2. We have now clarified this point as follows:

Discussion

[...] Similar to previous studies employing social determinants measures^{9,10}, our data was measured at individual-level. [...] Future studies should incorporate nested designs with aggregated-level analyses and geocoded data relevant to brain health in LA, such as structural inequality¹¹, air pollution¹², and access to green spaces¹³. A further step in future research may involve adjusting individual metrics by nested analysis of spatial buffers at different scales (i.e., block [0–100 m], neighborhood [100–500 m], district/commune [500–5,000 m], city [5–50 km]). Despite being unavailable in the current neuroscience literature, this would improve the ecological significance of the approach. [...]

Comment 4.3. The comment on the implausibility of MSE-brain structure correlations with fronto-temporal patterns in AD and insular regions in FTLD on the background of strong cerebellar associations was not addressed.

Response 4.3. We had previously provided evidence supporting the plausibility of this result based on the cerebellum's involvement in dementia¹⁴⁻¹⁷ and social determinants¹⁸⁻²². We have now clarified this more explicitly in the revised manuscript:

Discussion

[...] In addition to the disease-sensitive associations in each condition, we found additional cerebellar involvement in structural correlations. Although sometimes neglected in the literature^{17,23}, the cerebellum is a core structure in atrophy related to dementia¹⁴⁻¹⁷ and environmental stressors such as low-SES¹⁸⁻²², structural inequality¹¹, and poverty²⁴. This may also explain why the cerebellum is part of the allostatic interoceptive network²⁵, which is impaired in individuals with dementia²⁶⁻²⁹ and those with social disparities^{30,31} [...]

Comment 4.4. On a side note – given the inequalities in health care access, is there a possible bias in either the dementias sample that will drive the less socio-economically advantaged or the disadvantaged to seek diagnostic work-up? Similarly, in clinical neuropsychology testing we adjust the test results for the educational level, which is also recommended for screening tests like MMSE and MoCA in the form of reducing the maximum expected score of 30.

Response 4.4. We addressed the points about recruitment bias and education in the previous responses to reviewers #1 (**response 1.3**) and #3 (**response 3.4**). We have now made them more explicit in the revised manuscript:

(a) Recruitment bias: This point was already addressed in response to another reviewer during the first round of reviews (**response 3.4**). We have now made the point more explicit as follows:

Discussion

[...] Recruitment bias—factors that may influence the distribution of MSE scores during participant recruitment—could influence the results. However, several reasons suggest this is not the case, as we performed multiple control analyses. These included: (a) accounting for variations in disease severity, age at diagnosis, and time elapsed since diagnosis to mitigate potential overrepresentation of participants from disadvantaged groups in more advanced disease stages; (b) eliminating circularity in the assessment of MSE by excluding variables that may confound socioeconomic disparities with dementia outcomes (e.g., access to specific services such as cell phones could indicate either socioeconomic disparities or the consequences of cognitive decline); and (c) conducting within-group analyses to identify MSE associations through a dimensional approach rather than presuming uniform patterns across groups. However, future studies should incorporate other probabilistic or stratified random sampling strategies in the designs. [...]

(b) Adjustment by education: This point was also addressed in the previous response to another reviewer (**response 1.3**). Adjusting MMSE by education, as is common in clinical settings, is not applicable in our context as education is a core component of our MSE score. Adjusting the outcome variable (MMSE) by the predictor (education) would involve statistical circularity. In consequence, the model will have an artificial increased statistical effect. Instead, we conducted sensitivity analyses excluding education from the MSE score. We found that the association between lower MSE and poorer cognitive performance remained robust, confirming that educational differences between groups did not drive the results.

First, we confirmed that education is associated with the MSE, which violates assumptions of independence between predictor and outcome in regression analyses:

We additionally showed that applying cutoffs based on years of education³² is not instrumental, as it would include only 100 HCs (out of 1775) below the cutoff:

Black lines indicate the cutoff point for each educational level group:

We conducted a sensitivity analysis excluding the education component from the MSE composite score. This analysis demonstrated that the association between lower MSE and reduced cognitive performance is beyond education alone.

Sensitivity analyses

[...] Additional models were run to rule out potential bias from single factors linked to worse clinical-cognitive phenotypes (i.e., education and SES). Excluding education from the global MSE score confirmed that the association between adverse MSE and reduced cognition remained significant (**Supplementary Information 2**). [...]

Discussion

[...] The combined effects of MSE components demonstrated stronger associations even after excluding education [...]

Supplementary Information

Supplementary Information 2. Exclusion of education from the MSE global score

We created a model excluding this education to eliminate potential bias in the association between MSE and cognition caused by education in HC. Results showed that the association between adverse MSE and reduced cognition persisted after excluding education ($\beta = 0.304$ [95% CI = 0.241- 0.367], $p < 0.001$). Similar indices were obtained in the model that included education: CFI = 0.895, TLI = 0.721, RMSEA = 0.124, and SRMR = 0.083.

References

- 1 Guimerà, R. & Nunes Amaral, L. A. Functional cartography of complex metabolic networks. *Nature* **433**, 895-900, doi:10.1038/nature03288 (2005).
- 2 Sporns, O. Graph theory methods: applications in brain networks. *Dialogues in Clinical Neuroscience* **20**, 111-121, doi:10.31887/DCNS.2018.20.2/osporns (2018).
- 3 Rubinov, M. & Sporns, O. Complex network measures of brain connectivity: Uses and interpretations. *NeuroImage* **52**, 1059-1069, doi:<https://doi.org/10.1016/j.neuroimage.2009.10.003> (2010).
- 4 Suarez-Jimenez, B. et al. Intrusive Traumatic Re-Experiencing Domain: Functional Connectivity Feature Classification by the ENIGMA PTSD Consortium. *Biological Psychiatry Global Open Science* **4**, 299-307, doi:<https://doi.org/10.1016/j.bpsgos.2023.05.006> (2024).
- 5 Hausman, H. K. et al. The association between head motion during functional magnetic resonance imaging and executive functioning in older adults. *Neuroimage. Reports* **2**, doi:10.1016/j.nirp.2022.100085 (2022).
- 6 Haller, S. et al. Head Motion Parameters in fMRI Differ Between Patients with Mild Cognitive Impairment and Alzheimer Disease Versus Elderly Control Subjects. *Brain Topography* **27**, 801-807, doi:10.1007/s10548-014-0358-6 (2014).
- 7 Wylie, G. R., Genova, H., DeLuca, J., Chiaravalloti, N. & Sumowski, J. F. Functional magnetic resonance imaging movers and shakers: Does subject-movement cause sampling bias? *Human Brain Mapping* **35**, 1-13, doi:<https://doi.org/10.1002/hbm.22150> (2014).
- 8 Seto, E. et al. Quantifying Head Motion Associated with Motor Tasks Used in fMRI. *NeuroImage* **14**, 284-297, doi:<https://doi.org/10.1006/nimg.2001.0829> (2001).

- 9 Gonzalez-Gomez, R. *et al.* Educational disparities in brain health and dementia across Latin America and the United States. *Alzheimer's & Dementia: The Journal of the Alzheimer's Association* (2024).
- 10 Llibre-Guerra, J. J. *et al.* Social determinants of health but not global genetic ancestry predict dementia prevalence in Latin America. *Alzheimer's & Dementia*, doi:<https://doi.org/10.1002/alz.14041> (2024).
- 11 Legaz, A. *et al.* Structural inequality linked to brain volume and network dynamics in aging and dementia across the Americas. *Nature Aging*, doi:10.1038/s43587-024-00781-2 (2024).
- 12 Moguilner, S. *et al.* Brain clocks capture diversity and disparities in aging and dementia across geographically diverse populations. *Nat Med*, doi:10.1038/s41591-024-03209-x (2024).
- 13 Rojas-Rueda, D., Vaught, E. & Buss, D. Why a New Research Agenda on Green Spaces and Health Is Needed in Latin America: Results of a Systematic Review. *Int J Environ Res Public Health* **18**, doi:10.3390/ijerph18115839 (2021).
- 14 Singh-Bains, M. K. *et al.* Altered microglia and neurovasculature in the Alzheimer's disease cerebellum. *Neurobiology of Disease* **132**, 104589, doi:<https://doi.org/10.1016/j.nbd.2019.104589> (2019).
- 15 Jacobs, H. I. L. *et al.* The cerebellum in Alzheimer's disease: evaluating its role in cognitive decline. *Brain* **141**, 37-47, doi:10.1093/brain/awx194 (2018).
- 16 Chen, Y. *et al.* Cerebellar atrophy and its contribution to cognition in frontotemporal dementias. *Annals of Neurology* **84**, 98-109, doi:<https://doi.org/10.1002/ana.25271> (2018).
- 17 Schmahmann, J. D. Cerebellum in Alzheimer's disease and frontotemporal dementia: not a silent bystander. *Brain* **139**, 1314-1318, doi:10.1093/brain/aww064 (2016).
- 18 Cavanagh, J. *et al.* Socioeconomic Status and the Cerebellar Grey Matter Volume. Data from a Well-Characterised Population Sample. *The Cerebellum* **12**, 882-891, doi:10.1007/s12311-013-0497-4 (2013).
- 19 Loued-Khenissi, L. *et al.* Signatures of life course socioeconomic conditions in brain anatomy. *Human Brain Mapping* **43**, 2582-2606, doi:<https://doi.org/10.1002/hbm.25807> (2022).
- 20 Thanaraju, A. *et al.* Structural and functional brain correlates of socioeconomic status across the life span: A systematic review. *Neuroscience & Biobehavioral Reviews* **162**, 105716, doi:<https://doi.org/10.1016/j.neubiorev.2024.105716> (2024).
- 21 Kweon, H. *et al.* Human brain anatomy reflects separable genetic and environmental components of socioeconomic status. *Science advances* **8**, eabm2923 (2022).
- 22 Michael, C. *et al.* Socioeconomic resources in youth are linked to divergent patterns of network integration/segregation across the brain's transmodal axis. *PNAS Nexus* **3**, pgae412, doi:10.1093/pnasnexus/pgae412 (2024).
- 23 Guo, C. C. *et al.* Network-selective vulnerability of the human cerebellum to Alzheimer's disease and frontotemporal dementia. *Brain* **139**, 1527-1538, doi:10.1093/brain/aww003 (2016).
- 24 Qiu, S. *et al.* The ecology of poverty and children's brain development: A systematic review and quantitative meta-analysis of brain imaging studies. *Neuroscience & Biobehavioral Reviews* **169**, 105970, doi:<https://doi.org/10.1016/j.neubiorev.2024.105970> (2025).
- 25 Kleckner, I. R. *et al.* Evidence for a Large-Scale Brain System Supporting Allostasis and Interoception in Humans. *Nat Hum Behav* **1**, doi:10.1038/s41562-017-0069 (2017).

- 26 Migeot, J. A., Duran-Aniotz, C. A., Signorelli, C. M., Piguet, O. & Ibanez, A. A predictive coding framework of allostatic-interoceptive overload in frontotemporal dementia. *Trends Neurosci* **45**, 838-853, doi:10.1016/j.tins.2022.08.005 (2022).
- 27 Birba, A. *et al.* Allostatic-Interoceptive Overload in Frontotemporal Dementia. *Biological Psychiatry* **92**, 54-67, doi:<https://doi.org/10.1016/j.biopsych.2022.02.955> (2022).
- 28 Santamaría-García, H. *et al.* Allostatic Interoceptive Overload Across Psychiatric and Neurological Conditions. *Biological Psychiatry* **97**, 28-40, doi:10.1016/j.biopsych.2024.06.024 (2025).
- 29 Franco-O'Byrne, D., Santamaría-García, H., Migeot, J. & Ibáñez, A. Emerging Theories of Allostatic-Interoceptive Overload in Neurodegeneration. *Current topics in behavioral neurosciences*, doi:10.1007/7854_2024_471 (2024).
- 30 Migeot, J. & Ibáñez, A. in *Reference Module in Neuroscience and Biobehavioral Psychology* (Elsevier, 2023).
- 31 Alvarez, G. M., Rudolph, M. D., Cohen, J. R. & Muscatell, K. A. Lower Socioeconomic Position Is Associated with Greater Activity in and Integration within an Allostatic-Interoceptive Brain Network in Response to Affective Stimuli. *Journal of cognitive neuroscience* **34**, 1906-1927, doi:10.1162/jocn_a_01830 (2022).
- 32 Kochhann, R., Varela, J. S., Lisboa, C. S. M. & Chaves, M. L. F. The Mini Mental State Examination: Review of cutoff points adjusted for schooling in a large Southern Brazilian sample. *Dement Neuropsychol* **4**, 35-41, doi:10.1590/s1980-57642010dn40100006 (2010).